# Transcriptomic architecture of nuclei in the marmoset CNS

Jing-Ping Lin ®[1], Hannah M. Kelly[1], Yeajin Song[1], Riki Kawaguchi[2], Daniel H. Geschwind ®[2,3], Steven Jacobson[4] & Daniel S. Reich ®[1] ✉

To understand the cellular composition and region-specific specialization of white matter − a disease-relevant, glia-rich tissue highly expanded in primates relative to rodents − we profiled transcriptomes of ~500,000 nuclei from 19 tissue types of the central nervous system of healthy common marmoset and mapped 87 subclusters spatially onto a 3D MRI atlas. We performed cross-species comparison, explored regulatory pathways, modeled regional inter-cellular communication, and surveyed cellular determinants of neurological disorders. Here, we analyze this resource and find strong spatial segregation of microglia, oligodendrocyte progenitor cells, and astrocytes. White matter glia are diverse, enriched with genes involved in stimulus-response and biomole-cule modification, and predicted to interact with other resident cells more extensively than their gray matter counterparts. Conversely, gray matter glia preserve the expression of neural tube patterning genes into adulthood and share six transcription factors that restrict transcriptome complexity. A companion *Callithrix jacchus* Primate Cell Atlas (CjPCA) is available through https://cjpca.ninds.nih.gov.

An understanding of microenvironmental heterogeneity and its broad impact on biological processes is necessary to interpret experimental perturbations, especially in the central nervous system (CNS). Recent advances in genetic profiling tools have uncovered regional cellular diversity in the brain's gray matter (GM) far beyond what had traditionally been appreciated[1]. However, the characterization of cellular profiles in white matter (WM) is limited due to its modest representation in mouse. The common marmoset (*Callithrix jacchus*) is an emerging animal model that bridges mouse and higher primates genetically, immunologically, and behaviorally. Importantly, there is a massively greater (>5-fold) subcortical WM to cortical GM volumetric ratio in marmoset compared to mouse[2], raising the possibility that primates evolved novel but as-yet-undescribed glial heterogeneity to support this expansion.

Moreover, beyond obvious differences in the density of neurons and oligodendrocytes between GM and WM, the extent of structural and functional heterogeneity of other resident cells remains unclear. Motivating a deep investigation of such heterogeneity are prior observations that WM-astrocytes are primed to be more advanced in their response to pathological challenges[3,4]. For example, compared to astrocytes in GM, astrocytes in WM have a higher capacity for gluta-mate clearance to handle excitotoxic insults and disproportionally higher senescence-induced expression of GFAP (a reactive gliosis indicator)[5]. Similarly, more microglia are found in WM than in GM of normal human brain[6], and microglia in WM are primed to be more active and respond to injury faster than their GM counterparts[4,7,8]. Additionally, it has been shown that the timing and efficiency of remyelination mediated by oligodendrocyte progenitor cell (OPC) differentiation varies significantly between GM and WM[9].

To determine whether location-specific regulatory programs broadly influence resident cells, and whether these microenviron-mental cues lead to transcriptomic segregation that further defines

[1]Translational Neuroradiology Section, National Institute of Neurological Disorders and Stroke, National Institutes of Health, Bethesda, MD, USA. [2]Psychiatry, Semel Institute for Neuroscience and Human Behavior, David Geffen School of Medicine, University of California, Los Angeles, Los Angeles, CA, USA. [3]Departments of Neurology and Human Genetics, University of California, Los Angeles, Los Angeles, CA, USA. [4]Viral Immunology Section, National Institute of Neurological Disorders and Stroke, National Institutes of Health, Bethesda, MD, USA. ✉e-mail: daniel.reich@nih.gov

cell identities, we here describe a detailed map of the cellular composition across 19 CNS regions, including many WM areas, created by profiling all cell types without preselection. We extensively investigate the regional diversity of cells, especially with respect to GM-WM segregation, by comparing transcriptome similarity across species and datasets. We demonstrate ways of using this resource to classify unknown cell types, query intercellular communication, and discover associations with disease. Our carefully annotated marmoset brain cell atlas resource, "CjPCA," is designed to inform future studies in evolutionary, developmental, and pathological neurobiology.

## Results

### *Callithrix jacchus* primate cell atlas (CjPCA) analysis pipeline

To build an atlas with sampling reproducibility, a low bias in cell-type recovery, and good compatibility with clinical studies, we performed single-nucleus RNA sequencing (snRNA-seq) of nuclei extracted from uniformly sized tissue punches without preselection or sorting. snRNA-seq is widely applied in human tissue[10], able to identify cell types similarly to single-cell (sc) RNA-seq[11], and the only proven method to analyze tissue that cannot be readily dissociated into single-cell suspensions without introducing additional artifacts[10,12]. We performed in vivo magnetic resonance imaging (MRI) of the 2 marmoset brains, cross-referenced the imaging data to 3D MRI atlases to guide tissue sampling, surveyed cells without preselection, and grouped them into three different categories to facilitate downstream comparison (Fig. 1a–d and Supplementary Fig. 1). As indexed in Fig. 1d, we use "WM," "GM," and "other" (in quote marks) to indicate sampling sites as specifically defined in our paper; WM/GM (without quote marks) is used for general descriptive purposes, including when mentioning published works.

To ensure that common artifacts were properly addressed in droplet-based transcriptome analysis, we applied SoupX[13] to subtract ambient RNA background and DoubletFinder[14] to remove doublets for individual samples before data integration with Harmony[15] (Methods, Table 1, Supplementary Figs. 2–4, and Supplementary Data 1). We confirmed that the segregation between major cell classes is stable and that paired samples from different animals are comparable, without much variation even before data alignment with Harmony (Supplementary Fig. 3a). After data integration, a total of 534,575 nuclei were recovered in Level 1 analysis, and six cell classes were determined by the expression of canonical markers (NEU, *CNTN5*⁺ neurons; OLI, *MOG*⁺ oligodendrocytes; AST, *ALDH1L1*⁺ astrocytes; OPC, *PDGFRA*⁺ oligodendrocyte progenitor cells; MIC, *PTPRC*⁺ microglia/immune cells; VAS, *LEPR*⁺ vascular cells/*CEMIP*⁺ meningeal cells / *TMEM232*⁺ ventricular cells; Fig. 1e). We found that only a modest number of neurons (~11% median abundance of total cells; Fig. 1f) were present in "WM," showcasing the precision of image-guided tissue sampling. We compared the profile of ambient RNA, frequently detected genes, and high-ranked variable genes in different cell types across brain regions and found no evidence of systematic tissue type-specific contamination from background RNA or doublets after quality control (Supplementary Figs. 5, 6).

The abundance of oligodendrocytes and neurons was correlated across tissue types (Supplementary Fig. 4e), such that more oligodendrocytes were found in "WM" and more neurons in "GM," as expected. By contrast, "other" had cellular composition intermediate between "WM" and "GM" (Fig. 1f). The relative composition of marmoset major cell types (47% neurons, 35% oligodendrocyte-lineage cells, 12% astrocytes, and 4% immune cells) across 19 selected regions corresponds well to morphological counting of cell types in the human neocortex (42% neurons, 43% oligodendrocyte-lineage cells, 11% astrocytes, and 3% immune cells) across age (18–93 years) and sex[16,17]. Interestingly, there was a positive correlation between the abundance of microglia and oligodendrocyte progenitor cells (OPC) across tissue types (Supplementary Fig. 4e), with about three-fold higher microglia and two-fold higher OPC density in "WM" than "GM" (Fig. 1f).

Additional rounds of quality control and manifold learning constituted Level 2 analysis (Methods), in which six major cell classes were further grouped into 87 subclusters (comprising 50 NEU, 6 OLI, 5 OPC, 7 MIC, 8 AST, and 11 VAS subclusters). The 87 subclusters were then colored by sampling site to highlight regionally enriched subtypes (Fig. 1g). We mapped the general landscape of the dataset (Supplementary Figs. 6–8) and summarized the analysis workflow in a diagram to elucidate which type of cross-subcluster/cross-species comparison was performed in which cell class (Fig. 2). Unless indicated otherwise, all available nuclei collected from "WM," "GM," and "other" were included in each analysis.

With respect to neurons, it was not our primary focus to define new subtypes or quantify region and layer specificity, but we performed some basic analyses to anchor the resolution of our atlas with published datasets collected primarily from cortical regions[18]. In the current atlas, we profiled five different cortical areas and employed MRI-guided tissue collection to ensure consistency across animals. We note that a 2-mm-diameter tissue punch is sufficient to cover nearly the full thickness of marmoset cortex. Furthermore, the purity of cortical sampling can be estimated by the number of oligodendrocytes present in "GM" (~8% median abundance; Fig. 1f).

For neurons, a total of 248,091 nuclei yielded 50 NEU subclusters. We intentionally subclustered neurons at relatively low resolution to facilitate tracking of spatial origin. We highlight well-studied markers, cluster annotations, and sampling sites to facilitate cross-database comparison (Supplementary Figs. 9, 10). Nuclei in the UMAP were first colored by the expression of vesicular glutamate transporters (VGLUT; *SLC17A7*, *SLC17A6*, *SLC17A8*) (Supplementary Fig. 9b) and dot plot colored by neurotransmitter module scores (Supplementary Fig. 9b), dropping subclusters with less than 10% detection rate of VGLUT transcripts. We also colored nuclei by sampling site, which demonstrated, as expected based on our sampling, that cortical "GM" was the major contributor of neurons, with relatively high consistency in neuronal composition across "GM" (Supplementary Fig. 10c–g). In addition, we observed that cortical excitatory neurons (primarily VGLUT1⁺) were arranged onto a continuous path in the UMAP plot (lamination layer L2–L6, NEU32–45), which indicates similarity in the transcriptomes of neurons that reside in adjacent laminae. As previously reported in mouse and human[19], the expression of *STAB2* (L2–6), *LAMP5* (L2/3), *RORB* (L4), and *THEMIS* (L5/6) anchor the transition of the graded pattern along this path. Given that the establishment of lamination is completed prenatally[20], we cross-referenced our findings in the adult with an available in situ hybridization (ISH) database (Marmoset Gene Atlas) from P0 marmoset[21,22]. We found that the expression of lamina-enriched genes agreed with what has been examined spatially in the database (Supplementary Fig. 11).

Overall, the major features related to neurons (Supplementary Figs. 9–13) closely agree with several previous reports[1,19]. Therefore, we focused on the strong GM-WM spatial segregation of microglia, OPC, and astrocytes (Fig. 1g) to assess glial heterogeneity across 19 brain regions in detail. We further used GM-glia and WM-glia (i.e., WM-microglia, WM-OPC, WM-astrocytes, written here without quote marks) to indicate regionally enriched glial subtypes, as opposed to glia sampled from "GM" or "WM," which include all glia collected from the indicated area regardless of subtype. The design of our study does not enable us to address directly if GM- and WM-glia diverge autonomously (such as from different progenitors). However, we first explored the possibility that, regardless of developmental origin, glia might be specialized within each microenvironment in response to different functional demands. As gene expression often falls along a spectrum, we considered the transcriptomic landscape in its entirety instead of using one or a few genes to define each subpopulation. In the following analysis, we approach glial heterogeneity by comparing the biological programs we identified in marmoset with 11 published single-cell or single-nucleus studies in other species.

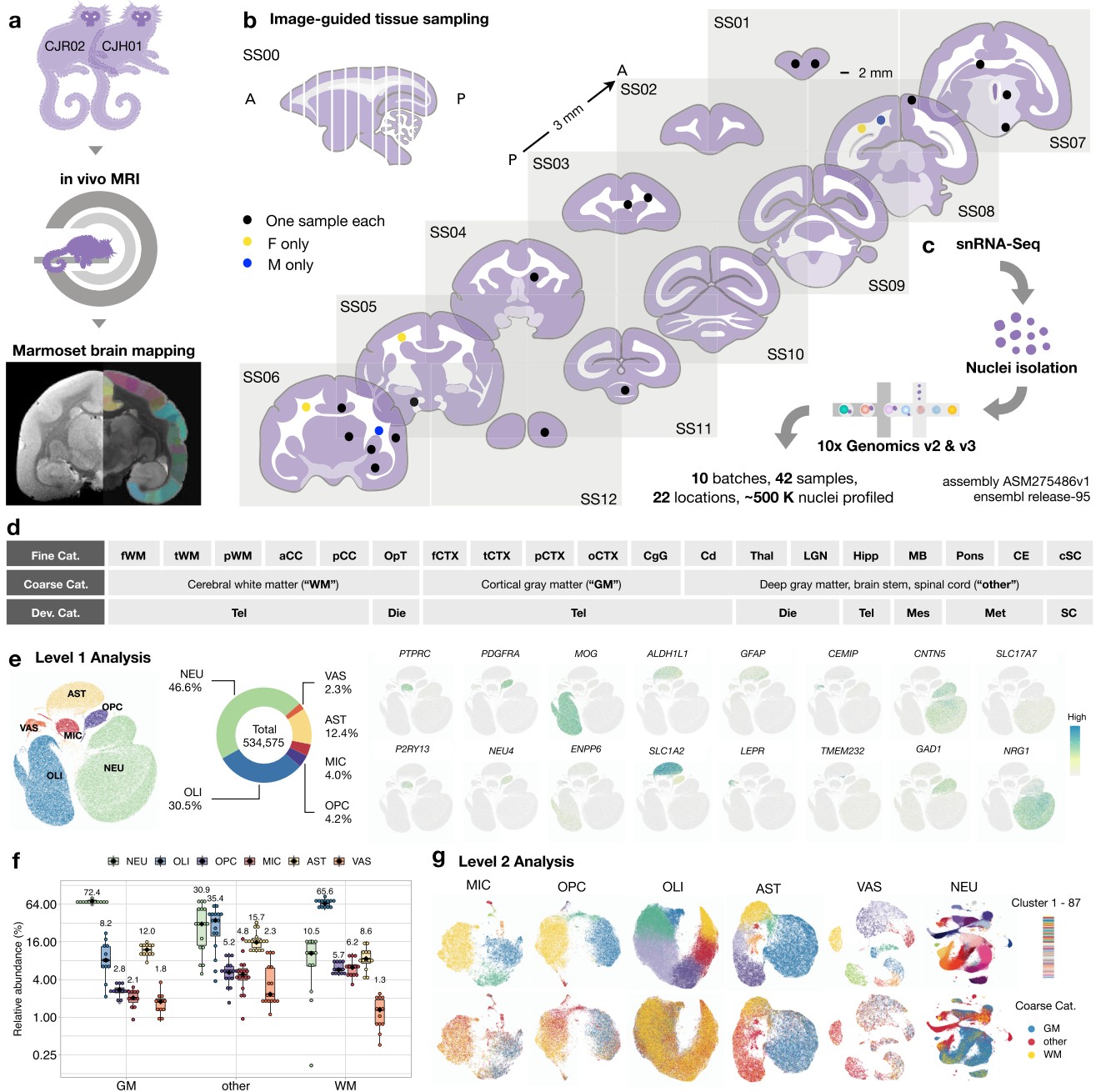

**Fig. 1 | Glial transcriptome reflects differential residence in gray and white matter. a** Experimental workflow to scan and map images to marmoset MRI atlases. **b** Location of samples collected as cylinders of 2 mm diameter and 3 mm thickness on the standard slab (SS) index. A anterior, P posterior. **c** Nuclei were isolated to prepare cDNA libraries and sequenced. **d** Total sampled areas are labeled by three types of tissue categories (Cat.): fine, coarse, and developmental (Dev). f frontal, t temporal, p parietal, WM white matter, a anterior, p posterior, CC corpus callosum, OpT optic tract, CTX cortex, o occipital, CgG cingulate gyrus, Cd caudate, Thal thalamus, LGN lateral geniculate nucleus, Hipp hippocampus, MB midbrain, CE cerebellum, cSC cervical spinal cord, Tel telencephalon, Die diencephalon, Mes mesencephalon, Met metencephalon, SC spinal cord. **e** The Level 1 analysis

identified six cell classes, rendered as a uniform manifold approximation and projection (UMAP) scatter plot annotated by expression of canonical marker genes: neurons (NEU), oligodendrocytes (OLI), oligodendrocyte progenitor cells (OPC), microglia/immune cells (MIC), astrocytes (AST), and vascular/meningeal/ventricular cells (VAS). **f** Box plot showing the abundance of Level 1 clusters as a function of tissue type; $n = 42$ independent samples; the median is annotated (black diamond shape) and listed. The lower and upper hinges of the box plot correspond to the 25th and 75th percentiles; whiskers extend from the hinges to maxima or minima at most 1.5 times inter-quartile range. **g** Top, each level 1 cell class was further subclustered in level 2 analysis. Bottom, the UMAP plots from level 2 analysis are colored by coarse tissue category.

## Microglia vary in density, morphology, and identity among gray and white matter

A total of 18,279 nuclei were included in the Level 2 analysis of microglia/immune cell (MIC class; Fig. 3a). We found seven distinct subclusters, of which four were circulating peripheral immune populations (PBMC1–4) and three were brain-resident immune cells (MIC1–3). In

addition to canonical markers (*P2RY13* and *ITGAX*), the expression of *FLT1* (vascular endothelial growth factor receptor 1) differentiates circulating from resident immune cells, such that microglia were *FLT1*+ and PBMC were *FLT1*− (Fig. 3b and Supplementary Figs. 14, 15).

We identified regionally enriched subtypes across 19 tissue types. We denoted two subtypes (MIC1 and MIC2) as GM-microglia, for they

**Table 1 | Key resources**

| Deposited Data | Identifier |
|---|---|
| Hammond et. al. 2019 | GEO: GSE121654 |
| Marisca et. al. 2020 | GEO: GSE132166 |
| Marques et. al. 2016 | GEO: GSE75330 |
| Zeisel et. al. 2018 | http://mousebrain.org/ |
| Zhang et. al. 2014 | GEO: GSE52564 |
| Polioudakis et. al. 2019 | dbGaP: phs001836 |
| Lake et. al. 2018 | GEO: GSE97930 |
| Habib et. al. 2017 | GEO: GSE104525 |
| Zhang et. al. 2016 | GEO: GSE73721 |
| Jäkel et. al. 2019 | GEO: GSE118257 |
| Absinta et. al. 2021 | GEO: GSE180759 |
| Marmoset Gene Atlas | https://gene-atlas.brainminds.riken.jp/ |
| Marmoset Brain Mapping | https://marmosetbrainmapping.org/ |
| **Software and Algorithms** | **Identifier** |
| R (v3.6.1 2019-07-05) | https://cran.r-project.org/bin/macosx/ |
| Cellranger (v3.0.2) | https://www.10xgenomics.com/ |
| seurat (v3.1.5) | https://github.com/satijalab/seurat |
| DoubletFinder (v2.0.2) | https://github.com/chris-mcginnis-ucsf/DoubletFinder |
| clustree (v0.4.3) | https://github.com/lazappi/clustree |
| SoupX (v1.4.5) | https://github.com/constantAmateur/SoupX |
| harmony (v1.0) | https://github.com/immunogenomics/harmony |
| monocle3 (v0.2.0) | https://github.com/cole-trapnell-lab/monocle3 |
| gprofiler2 (v0.1.9) | https://cran.r-project.org/web/packages/gprofiler2/index.html |
| nichenetr (v0.1.0) | https://github.com/saeyslab/nichenetr |
| EWCE (v0.99.2) | https://github.com/NathanSkene/EWCE |
| Ingenuity Pathway Analysis (v01-16) | https://digitalinsights.qiagen.com/product-login/ |
| Fiji (v2.1.0/1.53c) | https://imagej.net/Fiji/Downloads |
| **Other** | **Identifier** |
| CjPCA website | https://cjpca.ninds.nih.gov |

were found to be most abundant in "GM." We then named the other major cluster (MIC3) WM-microglia for its absence in "GM" and enrichment in "WM." All three subtypes of microglia present with various proportions in "other," which had cellular composition intermediate between relative pure WM and GM (Fig. 3c and Supplementary Fig. 14a, b). This GM-WM segregation of microglia was so strong that the abundance of WM-microglia (MIC3) was positively and negatively correlated with the number of oligodendrocytes and neurons, respectively. In contrast, GM-microglia (MIC1 and MIC2) had similar densities across brain regions (Supplementary Fig. 14c). We found that the expression of *SLC15A1*, an oligopeptide transporter, is selectively enriched in WM-microglia (Fig. 3c) and validated that anti-SLC15A1 preferentially labels IBA1+ cells in WM (Fig. 3d). Next, we performed particle and morphological analysis on IBA1 labeling to compare the density and the shape of microglia in GM and adjacent WM. We found two to three times more IBA1+ cells present in WM compared to GM, which agrees with the relative abundance of microglia profiled from "GM" and "WM" with snRNA-seq (Fig. 3c). Moreover, the shape of microglia in WM was more elongated, indicated by a larger value of reciprocal circularity, compared to GM (Fig. 3e–h).

To understand the functional implication of this segregation, we identified pathways that are differentially weighted in each microglia subtype using gene module[23] and gene ontology (GO) analysis and explored the similarity of these programs compared to published work

(Supplementary Figs. 14–17 and Supplementary Data 2). It has been shown that normal aging impacts GM and WM asynchronously[24–26]. Therefore, we sought to compare these regionally enriched modules in microglia against a dataset with temporal resolution. We linked marmoset gene names to their mouse orthologs, then cross-referenced the expression pattern of the defined modules in microglia extracted from the whole mouse brain (ages E14.5 to P540)[27]. After splitting mouse microglia into 3 age groups (embryo, neonate, and adult; Fig. 3i and Supplementary Fig. 16), we found that gene modules enriched in marmoset GM-microglia were highly expressed in microglia of young mice, whereas gene modules enriched in marmoset WM-microglia were also highly expressed in microglia of adult mice (Fig. 3j). These findings suggest that the transcriptomic profile of WM-microglia appears further aged than that of GM-microglia. GM-WM segregation of the microglial transcriptome is observed as early as P7 (during myelinogenesis) in mouse[8] and persists with normal aging in both human and mouse[7,28,29]. Understanding whether environmental cues in myelin-rich regions drive microglia specialization requires further study.

Next, we performed a GO analysis to summarize the regulatory programs enriched in each gene module. As a positive control, we found the expected sharing across microglia subtypes of gene modules involved in synapse pruning, complement system, and major histocompatibility complex (MHC) (Supplementary Fig. 17a). Next, we focused on comparing GO terms that are specifically enriched in each subtype. Terms related to synaptic plasticity, neurotransmitter secretion, and neuron survival are enriched in GM-microglia (Knn.m3; Supplementary Fig. 17b), whereas terms related to biomolecule metabolism, cell movement, and response to stimulus are enriched in WM-microglia (PG.m1/4; Supplementary Fig. 17d, e). Therefore, GM-microglia appear younger and more involved in modulating neuronal activity, while WM-microglia appear older and are primed to a more reactive state even in homeostasis.

## WM-OPC form a unique population with regional density closely associated with oligodendrocytes

Our analysis demonstrates that although it is challenging to find OPC-specific gene markers, OPC nonetheless comprises a distinct and varied population and express more genes that are enriched in neurons (e.g., *CADPS*, *RIMS2*, *DLGAP1*, *NRXN3*, and *STBP5L*) than do other glia and vasculature-associated cells (Supplementary Figs. 5, 6). As with microglia, GM-WM segregation is prominent in OPC, which we grouped into 5 subclusters (OPC1–5) from a total of 20,306 nuclei (Fig. 4a and Supplementary Figs. 18–20). The number of WM-OPC (OPC3) was positively correlated with the abundance of oligodendrocytes and negatively with the abundance of neurons, whereas GM-OPC (OPC1) were similar in density regardless of sampling site (Supplementary Fig. 18c).

Interestingly, several top-enriched genes related to general nervous system functioning were shared between GM-OPC (OPC1) and GM-microglia (MIC1), and both populations had fewer detected genes compared to their WM counterparts (Supplementary Figs. 14b, 18b). In gene module analysis (Supplementary Figs. 18e–19 and Supplementary Data 2), we found that WM-OPC were enriched with GO processes related to component organization, molecule modification, and stress granules (Knn.m6; Supplementary Fig. 19d), whereas GM-OPC enriched pathways are involved in neuronal support (PG.m2; Supplementary Fig. 19c) similar to those enriched in GM-microglia. Markers enriched in WM-OPC are known to regulate OPC dispersal (*SLIT2*)[30] and inhibit CNS angiogenesis (*SEMA3E*)[31] (Supplementary Figs. 18d, 20). This analysis suggests that WM-OPC, in homeostasis, are a population tuned to a more reactive state, whereas GM-OPC are more involved in supporting neuronal functions.

In line with our finding that marmoset WM-microglia appear transcriptionally more advanced in normal aging than their GM

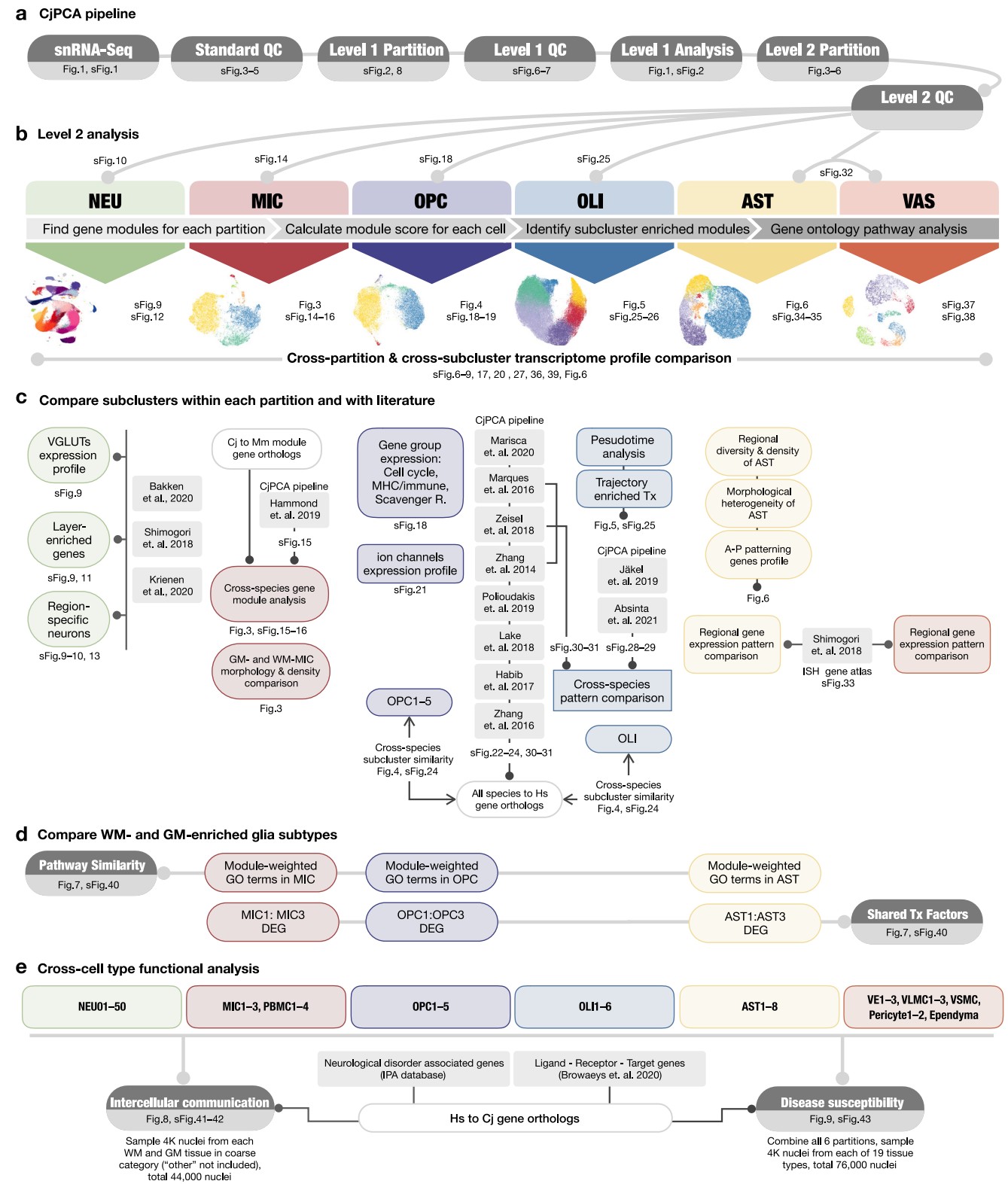

**Fig. 2 | Schematic summary of analysis workflow with figure index. a** Data preprocessing steps with quality check and Level 1 analysis. **b** Level 2 analysis done on all cell classes, including subclustering, gene module analysis, and gene ontology pathway analysis. **c** Level 2 analysis done on individual partitions, focusing on cross-dataset and cross-cluster comparisons. Mm *Mus musculus*, Hs *Homo* *sapiens*, Cj *Callithrix jacchus*, Tx transcription factors, A anterior, P posterior. **d** Finding regional regulatory programs that are shared across glia enriched in WM and GM. **e** Exploring functional implications for the transcriptomes enriched in WM- and GM-glia by assessing intercellular communication and disease susceptibility. Hs *Homo sapiens,* Cj *Callithrix jacchus*.

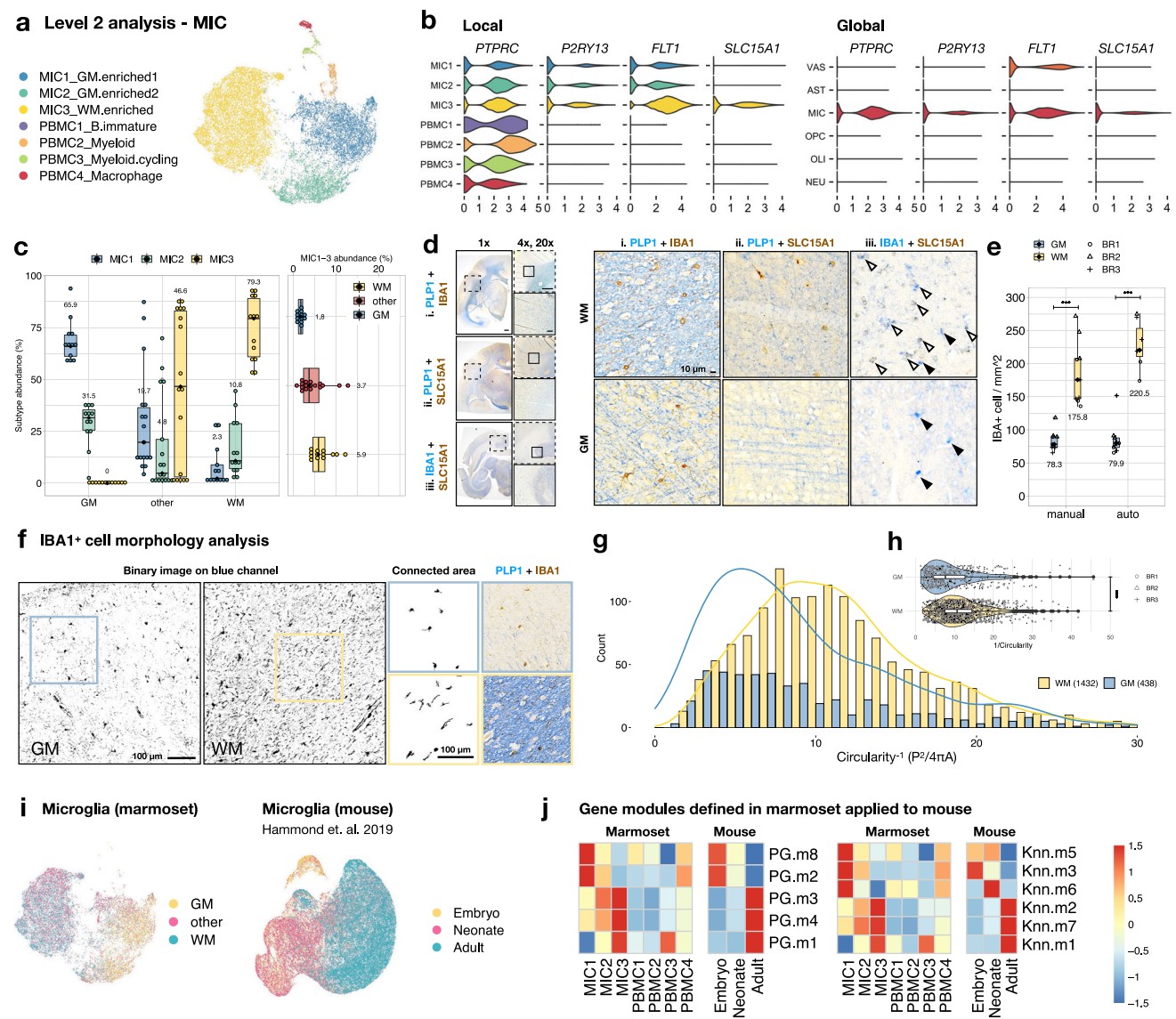

**Fig. 3 | WM-microglia appear more advanced in age than GM-microglia, are elongated, and selectively express SLC15A1. a** UMAP plot of microglia/immune cells colored by Level 2 subclustering. PBMC peripheral blood mononuclear cells. **b** Violin plot showing the expression of genes in each subcluster (local) and in each cell class (global). **c** Relative abundance of microglia by coarse tissue category; $n = 42$ independent samples; the median is annotated (black diamond shape) and listed. **d** Expression of SLC15A1 in adult marmoset brain sections. Solid arrowheads, IBA1⁺/SLC15A1⁻ microglia; open arrowheads, IBA1⁺/SLC15A⁺ microglia. IBA1 is a pan microglia marker, and intense PLP1 labeling demarcates the WM area. Scale bar, 1 mm (1x and 4x), 100 μm (20x). **e** Box-and-whisker plot showing the number of IBA1⁺ cells from three ROI per tissue type per biological repeat (BR) that were quantified manually or by automatic image processing; $n = 9$ ROI from three biologically independent animals; the median is annotated (black diamond shape) and listed. ***$p < 0.001$, t-test, two-sided, $p = 2.5E-04$ (group manual), $p = 5.8E-08$ (group auto). **f** The morphology of IBA⁺ cells in GM and WM was extracted by processing IBA1/PLP1 labeled images. Experiment was repeated independently three times with similar results as quantified in **g, h**. **g** The distribution of shape factor by

measuring perimeter (P) and area (A) of IBA1⁺ cells to calculate the reciprocal of circularity (P²/4 πA) with a step size of 1 grouped by tissue type. Circularity⁻¹ ranges from 1 (perfect circle) to infinity. **h** Violin plot summarizing the shape factor of IBA1⁺ cells. The reciprocal of circularity measured from WM cells is significantly higher than that measured from GM cells, ***$p < 0.001$, t-test, two-sided, $p = 6.8E-07$; $n = 1432$ cells examined over three WM areas from biologically independent animals; $n = 438$ cells examined over three GM areas from biologically independent animals. **i** UMAP plots of marmoset microglia colored by tissue type and mouse microglia from Hammond et al. 2019 colored by animal age. **j** Heatmap showing the expression of gene modules in seven MIC subclusters from marmoset and mouse microglia grouped by age. Gene modules enriched in GM-microglia (MIC1) of marmoset are enriched in microglia of younger mice (PG.m2/8, Knn.m6/3/5), and gene modules enriched in WM-microglia (MIC3) of marmoset are enriched in microglia of older mice (PG.m1/4/3, Knn.m1/7/2). The lower and upper hinges of the box plot in **a, d, e** correspond to the 25th and 75th percentiles, whiskers extend from the hinges to maxima or minima at most 1.5 times inter-quartile range.

counterparts (Fig. 3i, j), it has been reported that rat OPC in WM are more mature than those in GM[32], and that they differentiate into mature oligodendrocytes more efficiently than OPC in GM[33]. Electrophysiological properties of OPC vary between WM and GM and with age, and they correlate with differentiation potentiality[34,35]. We examined the OPC expression of ion channel genes as a surrogate of

electrophysiological function and examined the tissue origin of differentiating OPC (OPC5). We found different profiles of ion channels in GM-OPC and WM-OPC (Supplementary Fig. 21) but a similar abundance (<1.5%) in OPC5 across brain regions (Fig. 4a).

Taken together, these findings lead us to hypothesize that divergent CNS environments might influence the molecular profile of their

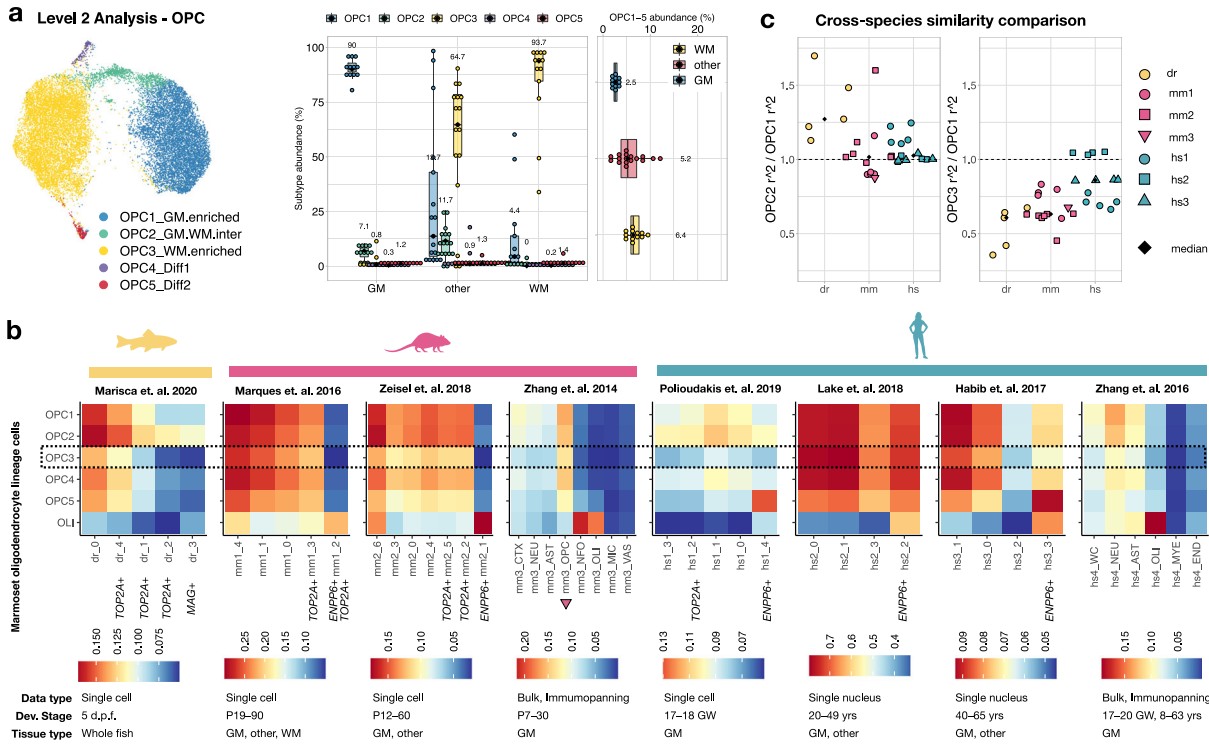

**Fig. 4 | Marmoset WM-OPC form a transcriptionally disparate population that diverges from previously reported OPC in other species. a** Left, UMAP plot of OPC colored by Level 2 subclustering. Right, relative abundance of OPC subclusters in coarse tissue category; $n = 42$ independent samples; the median is annotated (black diamond shape) and listed. The lower and upper hinges of the box plot correspond to the 25th and 75th percentiles, whiskers extend from the hinges to maxima or minima at most 1.5 times inter-quartile range. **b** Heatmap showing the Pearson's correlation coefficient $r$) between 6 oligodendrocyte-lineage cell clusters

(OPC1–5 and all oligodendrocytes, see Fig. 5) from marmoset and multiple sub-clusters of oligodendrocyte-lineage cells in zebrafish (dr), mouse (mm), and human (hs). d.p.f. days post fertilization, P postnatal day, GW gestational weeks. **c** Scatter plot showing the ratio of $r^2$ between OPC2 and OPC1 (left) and OPC3 and OPC1 (right). OPC1 and OPC2 are similar to one another and to OPC found in other species. WM-OPC (OPC3) is a distinct subcluster, in general showing lower similarity with previously defined clusters compared to GM-OPC (OPC1), though it is relatively more similar to human than mouse or zebrafish OPC.

resident cells in primates, and specifically that WM-OPC acquired additional features in response to their intercellular microenvironment. Testing this hypothesis and determining whether our observations translate to actual differences in stimulus responses in health and disease requires further experimental study.

To understand how marmoset OPC subclusters compare to those from other species, we reanalyzed data from prior studies[36–43] and performed a Pearson's correlation analysis (Fig. 2c and Supplementary Figs. 22–24). Consistent with what has been reported for OPC derived from the adult human brain, we did not observe a separate cycling OPC cluster ($TOP2A^+$) in adult marmoset brain, as has been reported for OPC derived from zebrafish, adult mouse, and developing human cortex (Supplementary Figs. 22, 23). Instead, cells expressing S and G2/M phase genes were dispersed across the OPC1–3 subclusters. OPC4, however, was enriched with G0G1 genes (Supplementary Fig. 18d), i.e., they are quiescent cells[44].

Prior to this comparison, we humanized gene names of each species with one-to-one orthologs and only included genes that are detected in all datasets, which might limit the depth of the comparison. However, we found agreement in oligodendrocyte-lineage differentiation features across species (Fig. 4b): marmoset differentiating OPC (OPC5) and oligodendrocytes (OLI) correlate with $ENPP6^+/MAG^+$ oligodendrocyte-lineage cells in mouse (mm1_2, mm2_1, mm3_NFO, mm3_OLI) and human (hs1_4, hs2_2, hs3_3, hs4_OLI), but less so in zebrafish (dr_3). Also, we observed consistently larger differences between marmoset WM-OPC (OPC3) and OPC from all other species analyzed. We quantified this observation by comparing the fold-change of similarity between OPC sub-clusters, measured as the ratio of $r^2$ values across clusters (Fig. 4c).

Although the underrepresentation of a marmoset WM-OPC-like population in other datasets may partially be due to technical differences, such as sampling site, it is also possible that OPC were broadly undersampled in other datasets. As clustering resolution is sensitive to cell counts in single-cell studies, low recovery number of OPC (particularly those derived from humans) and/or lack of inclusion of enough equivalent WM regions (especially in mice, where there is little WM) might contribute to this observation.

## The graded expression of *ENPP6*/*MUSK* and the succession of transcription factors delineate the transcriptome trajectory of oligodendrocytes

A total of 128,710 nuclei were included in the marmoset OLI class, from which six subclusters (OLI1–6) were identified (Fig. 5 and Supplementary Figs. 25–27). Different from other cell classes, marmoset oligodendrocytes were arranged into a continuous path in 2D dimension-reduced space (Fig. 5a), in which nuclei with similar transcriptomes are arranged as neighbors[45]. We found this to be similar to the patterns in human[46,47] and mouse[38,40]. In the following sections, we describe how this trajectory cannot be parsimoniously explained by a unidirectional path in oligodendrocyte-lineage development.

Based on mouse studies, differentiation-committed oligodendrocyte precursors are $Pdgfra^-/Tns3^+$[38], and the expression of *Enpp6* is a marker of newly forming oligodendrocytes[36,48]. Therefore, we denoted as OLI1 the subcluster that is $PDGFRA^-/TNS3^+/ENPP6^{high}$ and named the other OLI clusters (OLI2–6) consecutively (Supplementary Fig. 25e). Instead of a clear GM-WM segregation, we found proportional differences along the intermingled OLI subtypes across brain regions. OLI1 was lowest in "GM" (median abundance ~0.5%),

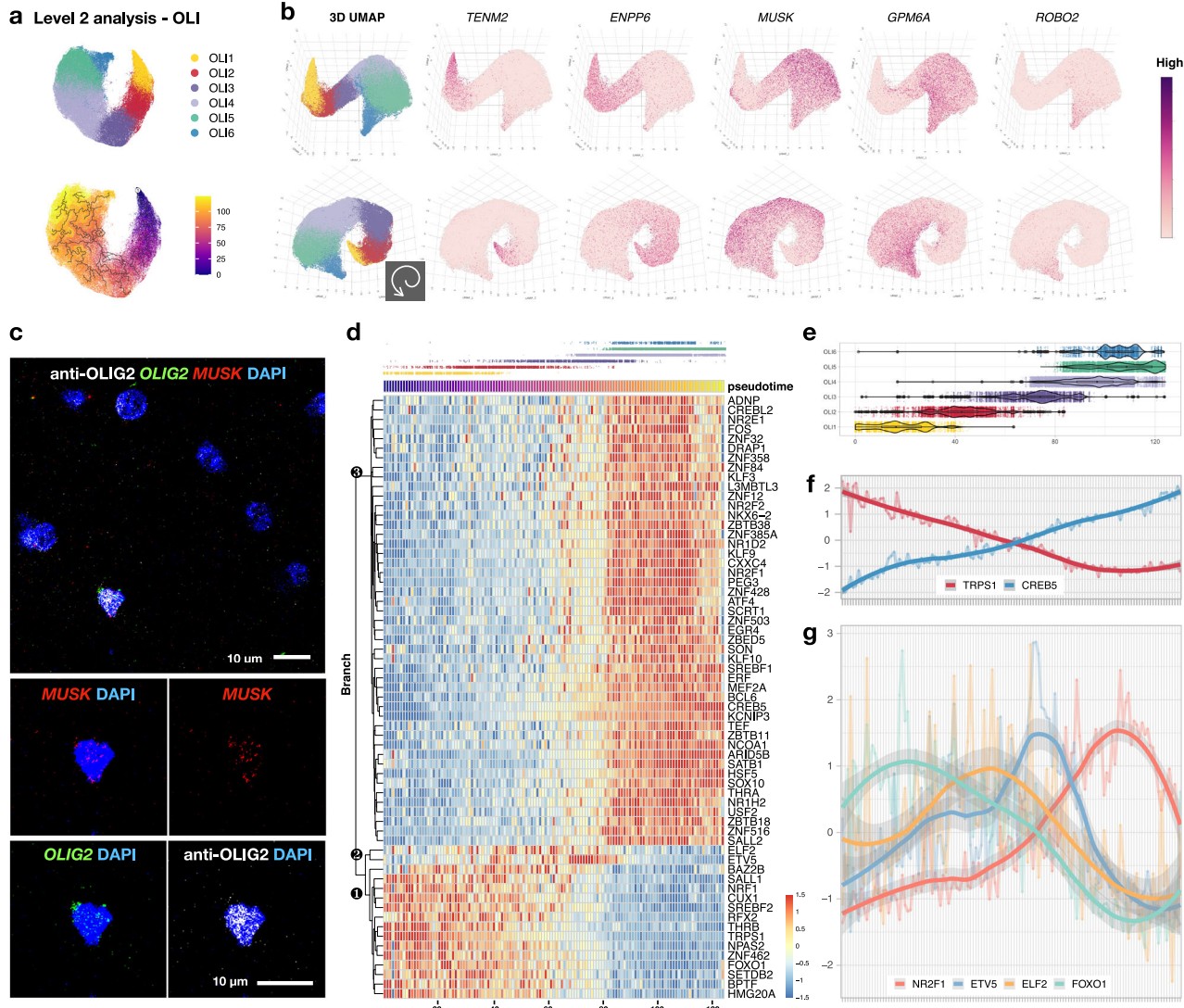

**Fig. 5 | The transcriptome of oligodendrocytes lies on a spiral trajectory flanked by *ENPP6*/*MUSK* gradient and marked by the succession of transcription factors. a** 2D UMAP plot of oligodendrocytes (OLI) colored by Level 2 subclustering (top) and transcriptomic distance in pseudotime (bottom, starting at OLI1, denoted by ①). **b** Two viewing angles of a 3D UMAP plot of OLI subclusters and corresponding expression of selected genes. **c** The expression of *MUSK* is detected in OLIG2⁺ cells in adult marmoset brain by combined immunofluorescence staining and fluorescent in situ hybridization (Hybridization chain reaction v3.0). **d** Heatmap showing the expression of transcription factors (TF) across pseudotime. Nuclei were grouped into 125 bins (columns, steps 1–125). The jitter plot above the heatmap is colored by OLI subcluster for visual reference. Three major branches of genes were annotated (❶, ❷, and ❸). The expression of 63 TF could be grouped into

three sets (Set 1: steps 0–60, present in OLI1–2, with Branch 1ʰⁱᵍʰ and Branch 3ˡᵒʷ; Set 2: steps 60–80, present in OLI3, with Branch 2ʰⁱᵍʰ; Set 3: steps 80–125, present in OLI4–6, with Branch 1ˡᵒʷ and Branch 3ʰⁱᵍʰ). **e** Box plots showing the distribution of OLI subclusters across pseudotime with median annotated; $n = 11{,}977$ (OLI1), 15,451 (OLI2), 18,623 (OLI3), 45,475 (OLI4), 30,691 (OLI5), 6,493 (OLI6) nuclei analyzed. The lower and upper hinges of the box plot correspond to the 25ᵗʰ and 75ᵗʰ percentiles, whiskers extend from the hinges to maxima or minima at most 1.5 times inter-quartile range. **f** The expression of TF with linearly decreasing (*TRPS1*) and increasing (*CREBS*) expression. **g** The expression of TF with levels that peak at various pseudotime points. The center of the error bands was defined by a locally estimated scatter plot smoothing (LOESS) curve fit for each expression pattern; the flanking gray bands indicate 95% confidence intervals.

compared to ~10% relative abundance in "WM" and "other" (Supplementary Fig. 25c).

Based on this expression pattern, one might surmise that OLI1 are the youngest and OLI6 the oldest oligodendrocytes; however, we found them to be close in the space of the 2D UMAP projection, which could indicate transcriptomic similarity; alternatively, a 2D projection may not be sufficient to capture important aspects of the data. Therefore, we pursued a 3D UMAP analysis of oligodendrocytes, finding that the two ends are separated along a spiral pattern that was also observed upon reanalysis of previously reported human[46,47] (Supplementary Figs. 28, 29) and mouse[40] (Supplementary Fig. 30) oligodendrocyte transcriptomes. This spiral pattern was also captured at the level of differentially expressed genes across oligodendrocyte

subclusters, most of which (*XYLT1*, *TNS1*, *TNS3*, *MAN1C1*, *BTBD16*, *CCP110*, *CSF1*, *DOCK5*, *PAM*, *MUSK*, *GPM6A*, and *DPP10*) were aligned across species to label the overall developmental trajectory (Fig. 5b and Supplementary Figs. 28–31).

Despite some discrepancy in the assignment of subcluster identity among datasets, five major stages of oligodendrocyte-lineage cells are widely accepted in the field to annotate the trajectory: OPC, differentiation-committed oligodendrocyte precursors (COP), newly formed oligodendrocytes (NFOL), myelin-forming oligodendrocytes (MFOL), and mature oligodendrocytes (MOL). Interestingly, however, *ENPP6*ʰⁱᵍʰ oligodendrocytes are located at both ends of the trajectory in mouse datasets (Supplementary Figs. 30, 31), but only at one end of the trajectory in marmoset and human. This observation raises the

question of whether the second *ENPP6*high population was selectively lost in primates, or, alternatively, whether labeling the marmoset *ENPP6*high population as "youngest" is not valid.

To address this question, we compared the transcriptomes from a bulk-RNA sequencing dataset from brain cells immunopanned with surface markers[36] and two single-cell sequencing datasets in mouse[38,40]. We found that the transcriptome of NFOL, as defined by their cell surface marker (GalC⁺), was most closely correlated with that of mature oligodendrocytes, as defined by single-cell analysis, and that the pattern of correlation did not track with the ordering of those oligodendrocyte subclusters in UMAP space (Supplementary Fig. 31). These observations raise the possibility that the single-path model of the differentiation trajectory of oligodendrocytes requires modification.

Whereas *ENPP6*, a choline-supplying enzyme[49], is enriched in OLI1–3, OLI4–6 were selectively labeled by *MUSK* (muscle-associated receptor tyrosine kinase) enrichment (Fig. 5b and Supplementary Fig. 25e). The oligodendroglial expression of *MUSK* has not been described; however, its functions in cholinergic signaling at the neuromuscular junction are known. With mechanism unknown, its expression in the brain is thought to mediate cholinergic responses, synaptic plasticity, and memory formation[50], suggesting that OLI4–6 might be a neuron activity-dependent population, consistent with findings from studies of adaptive myelination[51]. Moreover, it seems likely that there is species disparity in the expression of *MUSK* in oligodendrocytes, for it was detected in both marmoset and human but not in any mouse datasets. Although protein-level validation of MUSK expression in tissue was unsuccessful, we found that *MUSK* was indeed expressed by oligodendrocytes by fluorescent in situ hybridization: *MUSK*⁺/*OLIG2*⁺ double-labeled cells were found in both GM and WM of marmoset brain, and there was no noticeable difference in *MUSK* level per individual *OLIG2*⁺ in GM compared to WM (Fig. 5c and Supplementary Fig. 27d, e). Whether *MUSK* expression is unique to primates or animals in specific phylogenetic branches, and the extent to which it is developmentally regulated, require further investigation.

Next, we asked whether graded transcriptomic changes along the spiral oligodendrocyte trajectory can be modeled by waves of influence from within and/or directional stimuli from the environment. To address this, we performed a pseudotime analysis of marmoset oligodendrocytes and mapped the expression of transcription factors along pseudotime trajectories. We set *ENPP6*high oligodendrocytes as the starting point for this analysis and visualized gene expression dynamics along the pseudotime axis (Fig. 5b and Supplementary Fig. 25d). Although the pattern of expression dynamics agrees with the visual impression of gene expression along the spiral 3D UMAP path, we found that molecular distances from OLI4 to OLI5 and from OLI4 to OLI6 were similar, indicating that OLI5 and OLI6 might develop in parallel rather than dependently (Fig. 5d, e). We next mapped the expression pattern of transcription factors along the trajectory and found that different sets of regulator profiles were used by different subsets of oligodendrocytes (Fig. 5d–g). Of all transcription factors examined, only *ELF2* and *ETV5* peaked in the middle stages of oligodendrocytes (OLI2 and OLI3, respectively), whereas the other transcription factors were clustered either at the "early" (OLI1) or "late" (OLI4–6) stages. A positive correlation between *ELF2* and myelin was supported in a human snRNA-seq study, in which *ELF2* was high in control WM, normal-appearing WM, and remyelinated multiple sclerosis lesions but lower in WM lesions (active, chronic active, and chronic inactive)[46]. On the other hand, *Etv5* can act as a suppressor of oligodendrocyte differentiation, such that enforced expression of *Etv5* in rat OPC decreased the production of MBP⁺ oligodendrocytes[52]. That *ETV5* expression peaks in OLI3 (Step 60–80, Fig. 5d) suggests that OLI3 might be a population that is poised to further specialization upon appropriate signaling.

## Cell types at the barriers of the CNS

In Level 1 analysis, we observed an intermingled distribution of nuclei with shared transcriptomic features from the astrocyte (AST) and vascular (VAS) classes. Therefore, we pooled these two classes for the second round of quality control, which facilitated artifact imputation before further cell class splitting. A total of 74,204 nuclei comprising astrocytes and primary cell types (endothelial cells, meningeal cells, and ependymal cells) present at the CNS barriers (blood–brain, blood–CSF, and brain–CSF) remained after quality control (Supplementary Fig. 32). As the neurovascular unit is mostly established prenatally[53], we referred to a currently available ISH atlas of P0 marmoset[21,22] to confirm the localization of markers expressed by these cell types. We matched the gross histological morphology of the P0 brain to the adult marmoset MRI atlas (Supplementary Fig. 33)[54,55].

A total of 13,057 nuclei associated with CNS barriers comprised 11 VAS subclusters (Supplementary Figs. 34–36). Pericytes (Pericyte1–2), vascular endothelial cells (VE1–3), and vascular smooth muscle cells (VSMC) agreed with a human vascular atlas[56] and were broadly consistent across brain regions (Supplementary Figs. 34b, d). A relatively higher percentage of ependymal cells, which form a permissive interface between CSF and brain along the ventricular lining, was identified, as expected, in tissue samples that line ventricles (tWM, pCC, Cd, and cSC). The distribution of vascular and leptomeningeal cells (VLMC1–4, brain fibroblast-like cells) was variable and most highly detected in the hindbrain (pons and cerebellum; Supplementary Fig. 34b).

## The landscape of astrocytes can be mapped by GM-WM disparity and by compartments of the neural tube

For astrocytes, a total of 61,147 nuclei were partitioned into 8 subclusters (AST1–8; Fig. 6 and Supplementary Fig. 37). Similar to what was identified for microglia and OPC classes, AST1 was found most abundant in "GM," and AST3 was enriched in "WM" (Fig. 6a and Supplementary Fig. 37c). We noted that *ALDH1L1* and *GLI3* most effectively label the whole lineage of astrocytes across regions, including Bergmann glia (AST8) in the cerebellum (Supplementary Fig. 38). By contrast, GM-astrocytes (AST1) were enriched with *SLC1A2*, and WM-astrocytes (AST3) were enriched with *GFAP* and *AQP4*[57] (Supplementary Fig. 38). As expected, gene module and GO analysis showed that astrocytes are generally involved in sterol biosynthesis (Supplementary Fig. 39a and Supplementary Data 2), as they are the major cholesterol producers in the brain. For GM-astrocytes (AST1), terms related to neurotransmitter secretion and nervous system development were enriched. WM-astrocytes (AST3) were enriched with terms related to cell migration, intracellular signaling transduction, and morphogenesis (Supplementary Fig. 39b, c).

In "WM" samples, different profiles of astrocyte subtypes were observed; for example, AST4 and AST5 were enriched in the pCC and OpT but not in other WM areas, similar to what was found in Thal, LGN, MB, Pons, and cSC (Fig. 6a). Moreover, GFAP⁺ astrocytes greatly varied in density, size, and shape across the brain (Fig. 6b–d). This agrees with what has been described in the human brain[58], specifically that protoplasmic astrocytes are primarily found in the cortex, whereas WM-astrocytes are fibrous in morphology (Fig. 6b). The number and dimension of GFAP⁺ cells are diverse across cortical layers, tissue type, and even WM areas. These results lead to a prediction that the brain's astroglial response to stimuli may be heterogeneous even across WM areas (Fig. 6b–d).

Similar to what has been described in a mouse brain cell atlas[40], we found that grouping tissue by developmental category together with WM-GM disparity most effectively describes astrocyte segregation (Fig. 6e and Supplementary Fig. 37a). This observation led us to investigate further the effect of local neural tube patterning signals in defining astrocyte subclusters and whether these signals also affect other cell types in the same region. Therefore, we examined the expression of patterning genes along the anterior-posterior axis across

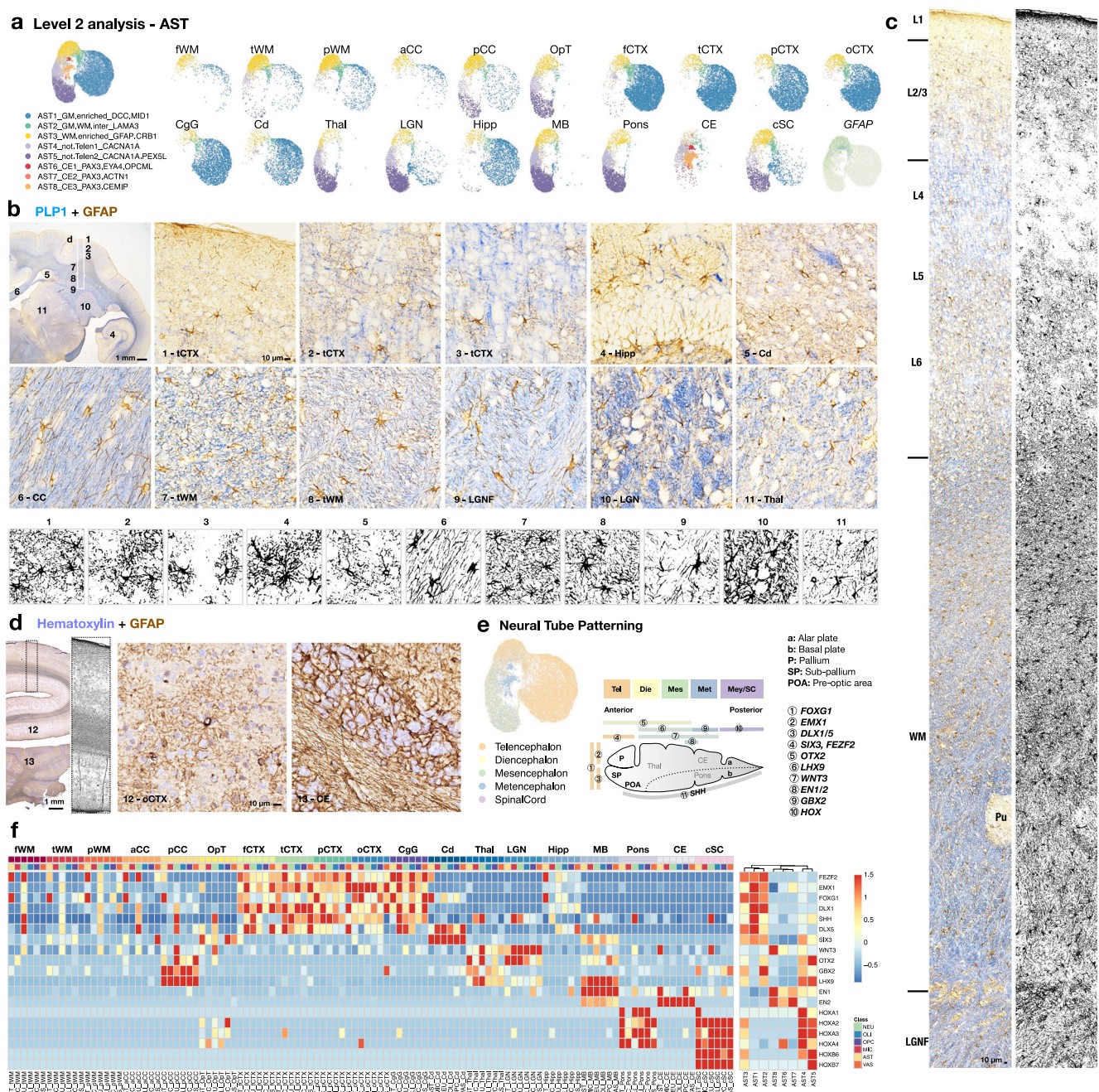

**Fig. 6 | Neural tube boundary-restricted astrocytes persist into adulthood.**
**a** UMAP plot visualization of astrocytes (AST) colored by Level 2 subclustering and split by tissue type. **b** The expression of GFAP in a mid-coronal section of the marmoset brain. Intense PLP1 labeling demarcates the white matter. Enlarged areas are boxed and numbered on the 1x image (top left panel). The morphology of GFAP⁺ cells across tissue types was extracted by image processing (Method). The experiment was repeated independently three times with similar results. **c** The distribution and morphology of GFAP⁺ cells along layers of cortex and adjacent white matter. The experiment was repeated independently three times with similar results. **d** The expression of GFAP in the occipital lobe and cerebellum; nuclei are stained with hematoxylin. Enlarged areas are boxed and numbered on the 1x image.

The experiment was repeated independently three times with similar results.
**e** UMAP plot visualization of astrocytes colored by tissue type and developmental category (left). Schematic illustration of the expression of patterning genes along the anterior-posterior axis of the neural tube during development (right).
**f** Heatmap showing the expression of selected patterning genes in Level 1 classes, split by sampling site. The sampling sites are ordered approximately along the anterior-posterior axis of the neural tube during development, from left to right; the genes enriched along the same axis are ordered from top to bottom. AST subclusters are grouped based on the expression similarity of these patterning genes, corresponding to the developmental origins of the sampling sites.

cell classes and tissue types (Fig. 6e, f). We found that the expression of patterning genes across brain regions was grossly preserved across cell types, with some interesting exceptions. In the telencephalon, all cell classes in cortical GM expressed high levels of patterning genes that were most prominently detected in the forebrain (*FEZF2*, *EMX1*, *FOXG1*, *DLX1*, *SHH*, and *DLX5*). Caudate (enriched with *SIX3*) and hippocampus,

though belonging to telencephalon gray (Supplementary Fig. 37a), have mostly lost the expression of forebrain patterning genes (Fig. 6f). Similarly, most WM cells appeared to have lost this specification, except for some *FEZF2* and *FOXG1* expression in astrocytes and neurons. Cells in the posterior corpus callosum had patterning gene expression similar to that observed in the thalamus, LGN, and midbrain

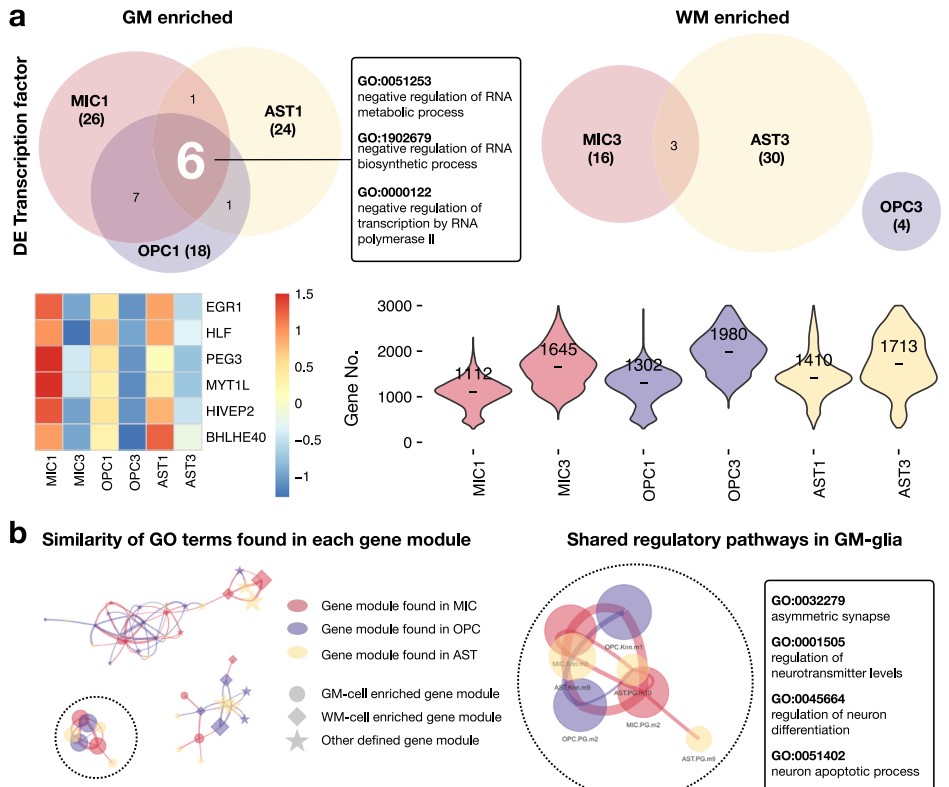

**Fig. 7 | GM-glia share regulatory pathways and express fewer genes than WM-glia. a** Venn diagram showing shared, differentially expressed transcription factors across GM-glia (left) and WM-glia (right). GO processes of the six shared transcription factors in GM-glia. Heatmap showing expression of the six shared transcription factors. Violin plot showing the number of differentially expressed genes detected in each cluster, with median annotated. DE differentially expressed. **b** Network plot showing the similarity of GO terms identified in each gene module. Node size represents the number of significant GO terms found in each gene module. Thicker edges reflect the higher similarity between two lists of GO processes. Edges were filtered (Jaccard index >0.25), resulting in three separate networks. The listed top GO terms are shared among GM-glia.

(*WNT3, OTX2, GBX2, LHX9,* and *EN1*). *EN2* was enriched in the cerebellum, but hindbrain patterning genes (*HOX* genes) were high in pons, cervical spinal cord, and sporadically in the optic tract.

In conclusion, we observed that the transcriptomes of WM cells seem to deviate from the profile of forebrain patterning. This forebrain, midbrain, and hindbrain specification was preserved prominently in astrocytes and indeed determined their identity as distinct AST subclusters (Fig. 6f). This suggests that heterogeneity in developmental origin might play a role in subtype specialization in addition to GM-WM disparity in diversifying astrocytes.

### GM-glia share regulatory pathways, and WM-glia interact with other resident cells more than GM-glia

The presence of gray-white matter segregation within some glial cell types, together with the observation of transcriptional similarity across glia within tissue types, led us to hypothesize that there might be regulatory programs that are shared by resident cells within the same microenvironment to execute intercellular functions properly. We reasoned that the similarity of enriched gene modules among resident glia might be due to the activation of common transcription factors. To explore this possibility, we extracted and compared differentially expressed transcription factors between matching GM-/WM-glia pairs (MIC1/MIC3, OPC1/OPC3, and AST1/AST3). We found greater overlap in differentially enriched transcription factors in GM-glia (15 overlapping transcription factors) than in WM-glia (three overlapping transcription factors). Interestingly, six transcription factors were shared across all GM-glia, whereas no transcription factors were shared across all WM-glia. GM-glia transcription factors (*EGR1, HLF, PEG3, MYT1L, HIVEP2,* and

*BHLHE40*) are known to restrict RNA biosynthesis, potentially explaining the observation that GM-glia are low in RNA features compared to their WM counterparts (Fig. 7a and Supplementary Fig. 40a).

Our observation of top-ranked GO terms that were similar among GM-glia but not among WM-glia led us to seek a better method to quantify this pattern systematically. We compared the similarity of terms by calculating the Jaccard index between module pairs across cell classes and visualizing their similarity as networks. Conformingly, GO terms were more similar among gene modules enriched in GM-glia than other gene modules, and regulatory programs in GM-microglia showed the highest similarity with those in GM-OPC (Fig. 7b and Supplementary Fig. 40b).

We reasoned that this cross-cell-type enrichment of similar regulatory programs might be achieved by close communication between neighboring cells; therefore, we modeled ligand-receptor interactions to test this hypothesis. To achieve this, we employed NicheNet analysis[59], which curates known ligand-receptor and receptor-target relationships and ranks them based on the level of support in published literature. We performed this analysis taking the subtypes of microglia, OPC, and astrocytes that were primarily enriched in "GM" (MIC1, OPC1, and AST1) and "WM" (MIC3, OPC3, and AST3) as "receivers" and other cells in the same tissue type as "senders" (Fig. 8a). We consistently found more, and more unique, ligand-target pairs in "WM" than in "GM," and these were generated by a wider variety of sender types (Fig. 8b, c, Supplementary Fig. 41, and Supplementary Data 3). In "WM," endothelial cells and astrocytes were the most frequently observed additional sender types (Supplementary Fig. 41f, i, l). Astrocytes (4830 pairs) formed more ligand-target pairs than microglia (2828 pairs) or OPC (1813 pairs). Among ligand-target pairs found

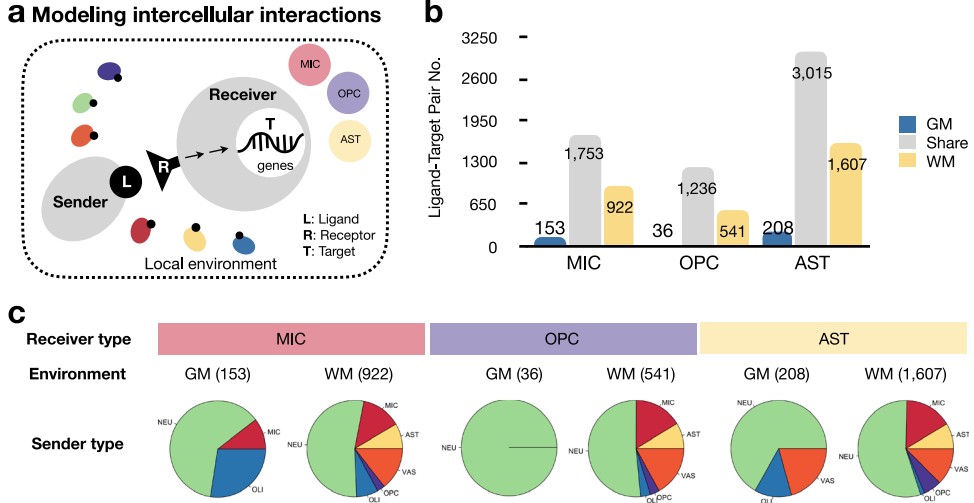

**Fig. 8 | WM and GM resident cells diverge in intercellular activity. a** Intercellular communication in WM and GM microenvironments were modeled with NicheNet. **b** Bar plot showing the number of established ligand-target paints that were unique to or shared across GM and WM. **c** Pie charts showing the sender cell types in Level 1 classes for ligand-target pairs that are unique to each environment.

shared in "GM" and "WM" (1753 for microglia, 1236 for OPC, 3015 for astrocytes), certain senders were represented disproportionately to their relative abundance (Figs. 1f, 3c, 4a and Supplementary Fig. 37c), with microglia and OPC constituting the most common top-ranked senders (Supplementary Fig. 42). This is consistent with prior reports that microglia and OPC can actively survey their microenvironments in both physiological and pathological conditions[60,61].

### GM- and WM-glia differentially contribute to neurological disorders

Finally, we explored the possible functional significance of our homeostatically defined subclusters as they might relate to pathological conditions. We reasoned that by examining the expression of known disease-associated genes in our healthy marmoset transcriptomic atlas, we might identify previously overlooked cellular contributors to human neurological disease. Based on manually curated information in the Ingenuity Pathway Analysis (IPA) database (Supplementary Data 4), we sorted genes into lists, ordered them based on the phenotypic similarity between disorders, and displayed the number of candidate genes in each list that were unique or shared across disorders; lists with <10 candidate genes were dropped for simplicity (Fig. 9a). We examined the cellular enrichment of genes associated with a spectrum of disorders using expression-weighted cell-type enrichment (EWCE) analysis[62]. We calculated fold-change, enrichment probability, and significance by comparison to gene expression in 100,000 randomly selected lists of genes with matching lengths from the background (Method).

Genes associated with an autism spectrum disorder or intellectual disability were enriched in both excitatory and inhibitory neurons[63,64], and there was a remarkably similar profile for seizures and schizophrenia (Supplementary Fig. 43). Genes related to migraine were also overrepresented in neuronal subclusters and notably absent from vascular subclusters. Astrocyte contribution was highlighted, in addition to the involvement of pyramidal neurons, in Huntington's disease, independently supporting reports of glial involvement in its pathogenesis[65]. In agreement with the view that neurovascular coupling plays a role in neurological disorders, a gene list that is uniquely shared by organic mental disorder, CNS tumor, and Huntington's disease (List.40) highlights the contribution of Pericyte2 (Supplementary Fig. 43). Although it affects the peripheral nervous system rather than the CNS, Charcot-Marie-Tooth disease mapped to oligodendrocytes, possibly due to shared gene expression between central

and peripheral myelinating cells. Interestingly, we observed the potential contribution of a subset of oligodendrocytes (OLI4–6) to parkinsonism, consistent with recent reports from postmortem brain transcriptomic data[66].

Consistent with the microenvironment specialization of glia reported here, we found examples in which genes associated with an organic mental disorder were differentially expressed in GM-microglia (MIC1) and GM-astrocytes (AST1) but not in their WM counterparts. Genes associated with CNS tumor were enriched in WM-microglia (MIC3) and WM-OPC (OPC3), but surprisingly not in astrocytes. By contrast, all microglia subtypes (MIC1–MIC3), but not other cell types, appear to contribute to multiple sclerosis pathogenesis (Fig. 9b), consistent with results from genome-wide association studies[67]. Interestingly, genes unique to CNS tumor and encephalitis (List.21) are differentially enriched in MIC2, a less dominant GM-microglia that is present in various proportions in microglia sampled from "WM" (Fig. 3c and Supplementary Fig. 14a). Together, these results support our contention that there is transcriptome diversity among GM- and WM-glia, and that these variations are significant enough for specific subtypes to be predicted to contribute differentially to various neurological disorders.

## Discussion

We have provided a resource and initial analysis for each major cell class across 19 CNS tissue types. We observed the greatest GM-WM spatial segregation in subclusters of microglia, OPC, and astrocytes. GM-glia are generally naïve, protoplasmic, and enriched in GO terms related to neuronal functioning, whereas WM-glia are more active, fibrous, and enriched in GO terms related to morphogenesis and signaling dynamics. We accumulated some evidence that WM-glia have accrued additional features, are further advanced in the program of specialization, and are more interactive than their GM counterparts. This atlas, therefore, serves as a bridge between rodent and human data that may prove useful for the understanding of the cellular and molecular basis of human neurological disorders.

Although our study was carefully designed and executed, and rigorous quality control steps were implemented at every stage of the experimental and analysis pipeline, technical variation and artifacts remain intermingled with biological effects. For example, spinal cord samples were outside the region covered by the MRI atlas, and results were derived from two libraries prepared with 10x v2 and 1 library with 10x v3 chemistry (fewer genes were recovered using v2

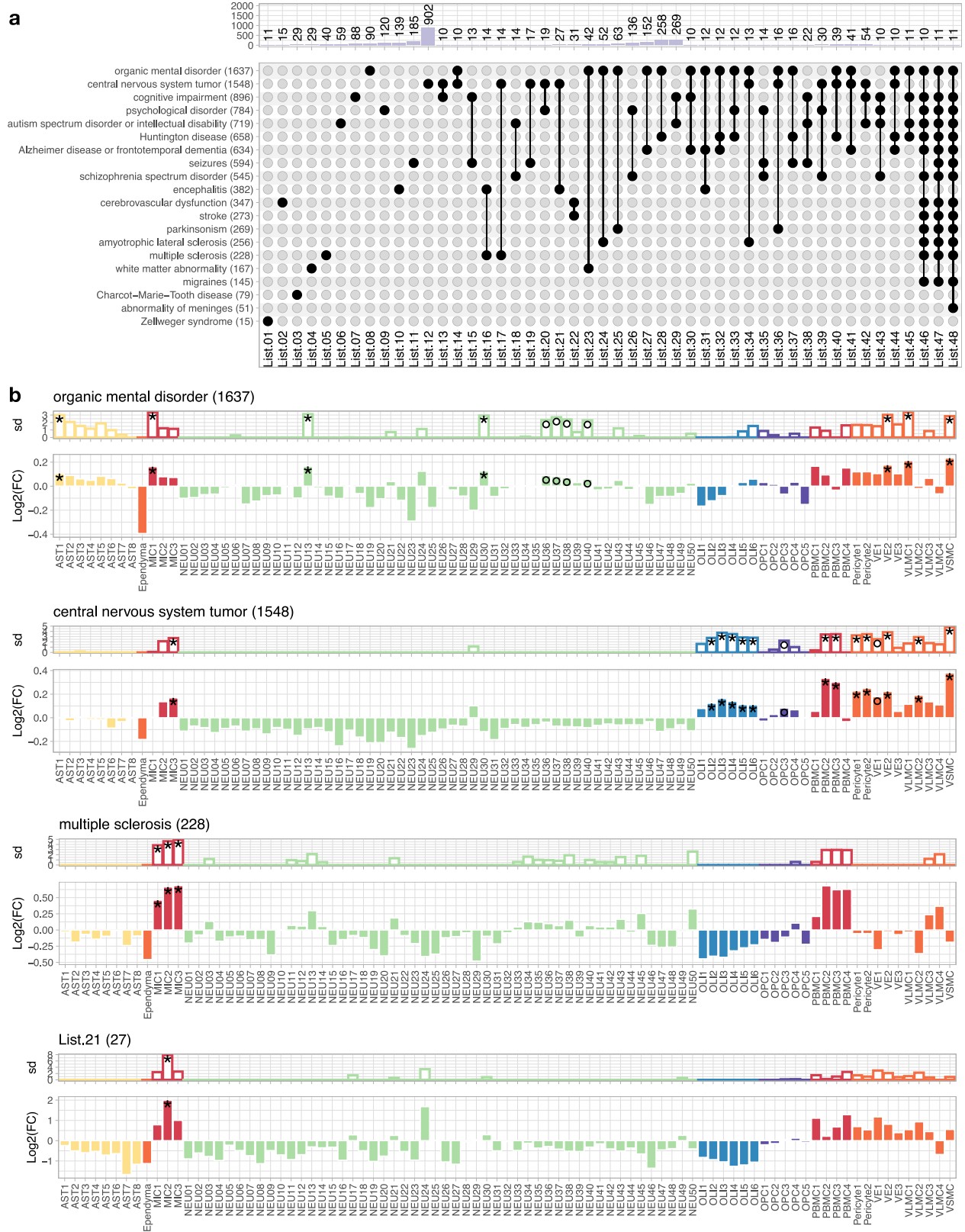

than from v3 on average). Additionally, while snRNA-seq has several advantages, including higher tolerance for tissue processing, it captures only nascent RNAs and cannot interrogate locally enriched species along neural cell processes. Another limitation is that annotations for genes and transcripts, and associated functional terms, are less complete for marmoset than for mouse or human, and some

annotations are inferred and might change in the future. We were therefore relatively conservative in clustering and defining cell types, and it is likely that further subclustering would have yielded more distinct cell types.

We defined cell types and linked their molecular properties to functions with more than one method, including pathway analysis,

**Fig. 9 | GM- and WM-glia diverge in disease contribution. a** Bar and UpSet plots showing the overlap of genes associated with various neurological disorders as defined in the Ingenuity Pathway Analysis (IPA) database. The number of genes is listed next to the name of the disorder (bottom panel, y axis label). The number of intersecting genes between indicated disorders (solid black dot and line), but not shared by any other disorders (empty gray dot), is labeled and shown on the bar graph (top panel). **b** Bar graph showing standard deviation (top) and fold-change (bottom) of the enrichment probability for neurological disorder-associated genes calculated by expression-weighted cell-type enrichment (EWCE) analysis, after bootstrapping. Bar color represents the Level 1 cell class of origin. Genes annotated in the IPA database associated with an organic mental disorder, CNS tumor, multiple sclerosis, and List.21 (27 genes, shared by encephalitis and CNS tumor but not by other neurological disorders) were analyzed and plotted. The significance of cell-type enrichment is denoted after Benjamini-Hochberg correction, *$q < 0.05$ and °$q < 0.1$.

disease association mapping, and morphological analysis; however, electrophysiological features remain unlinked. With respect to sampling, although we profiled as many nuclei as possible in each round, often <1% nuclei were studied from each region (Supplementary Fig. 1e), meaning that rare subclusters were probably missed. Additionally, given finite resources, we elected to sample the brain richly rather than to include samples from additional marmosets, limiting, for example, our ability to answer age-, sex-, or left-right asymmetry-related questions. Instead, we performed associational analysis and first highlighted shared features across coarse tissue types. Further analysis in combination with direct experimental tissue-level validation is necessary to assess region-specific phenotypes in each fine structure.

These limitations aside, the protocol described here is easily adapted to other settings and allows nuclei to be mapped onto relatively small regions to increase reproducibility and aid future validation studies. Our analysis, therefore, provides a framework for understanding the diversity of cell types in the marmoset brain, allowing us to form actionable hypotheses and laying the groundwork for future studies.

# Methods
## Animals
Our CNS marmoset cell atlas was generated from two healthy, 5.5-year-old common marmosets (*Callithrix jacchus*), one female (CJH01) and one male (CJR02). Staining was done using 4 healthy 4–6-year-old marmosets, two females (CJaT01 and CJaV02) and two males (CJaB03 and CJaD04). All marmosets were housed and handled with the approval of the NINDS/NIDCD/NCCIH Animal Care and Use Committee (ACUC). On the day of imaging, marmosets were anesthetized by intramuscular injection of 10 mg/kg ketamine, intubated, and ventilated with a mixture of isoflurane and oxygen during in vivo MRI scans. MRI was performed on a 7 T Bruker scanner to generate a series of proton density-weighted images with a resolution of $0.15 \times 0.15 \times 1 \, mm^3$ per voxel and a matrix of $213 \times 160 \times 36$ per session[68]. The images were used for volume reconstruction and anatomy identification. The volume was then used to create a custom-made brain holder by 3D printing (Ultimaker 2⁺) for each marmoset to guide ex vivo tissue sampling[69]. After each in vivo scan section, marmosets were weaned from 2% isoflurane, recovered with a lactated ringer injection subcutaneously, and returned to their original housing.

## Single-nucleus RNA sequencing (snRNA-seq)
**Tissue dissection for nuclei isolation.** On the day of tissue harvest, marmosets were deeply anesthetized with 5% isoflurane until any visible signs of breathing were no longer detected. Animals were transcardially perfused with ice-cold artificial cerebrospinal fluid (aCSF) for 5 min with a pump. Brains were removed from the skull and submerged into ice-cold aCSF, and after the removal of meninges within the aCSF solution, were positioned in a custom-designed brain holder within 10 min post-perfusion. The brain was sectioned at 3 mm into 12–13 slabs in one step with a homemade blade-separator set in the aCSF solution. Each brain slab was transferred into a 6-well plate, submerged into RNAlater (RNAlater™ Stabilization Solution, AM7021, Invitrogen) with a homemade brain trap, and stored at 4 °C overnight. The following day, brain slabs were positioned in 25 × 20 × 5 mm molds

(Tissue-Tek® Cryomold®, 4557, Sakura Finetek) on ice to facilitate target sampling. Slabs were matched to MRI for each animal, and tissue annotation for gray[54] and white matter[55] was informed by marmoset 3D MRI atlases V1 and V2. A cylinder of tissue 2 mm in diameter and 3 mm in height for each region (Fig. 1b and Supplementary Fig. 1) was collected with a tissue punch (EMS-core sampling tool, 69039-20, EMS). There were five white matter samples from temporal and parietal lobes that did not exactly match the two animals, however, they were paired in lobes of the brain and showed no significant differences in subsequent analysis (Fig. 1b, SS05, SS06, and SS08). The cylinders were ejected into PCR tubes filled with 100 μL of RNAlater and stored at −80 °C. The quality of RNAlater-preserved tissue was assessed by measuring RNA Integrity Number (RIN) on the Agilent 2100 Bioanalyzer (G2939BA, Agilent). Bulk RNA was isolated with TRIzol™ Reagent (15596026, Invitrogen) and measured with Agilent RNA 6000 Pico Kit (5067-1513, Agilent); samples with RIN >8.5 were used in the study.

**Single-nucleus dissociation.** Nuclei preparation was carried out as described[70], with minor modifications. Briefly, on the day of dissociation, tissue samples were thawed on ice, removed from the solution, dabbed with Kimwipes to remove residual RNAlater, and placed in a 1 mL douncer tube (Dounce Tissue Grinder, 357538, Wheaton). Each tissue was homogenized in 500 μL of lysis buffer containing 400 units of RNase inhibitor (RNaseOUT Recombinant Ribonuclease Inhibitor, 10777-019, Invitrogen) and 0.1% Triton-X100 in low sucrose buffer (0.32 M sucrose, 10 mM HEPES, 5 mM $CaCl_2$, 3 mM MgAc, 0.1 mM EDTA, and 1 mM DTT in $ddH_2O$, pH8) with loose pestle 25 times and tight pestle ten times. The homogenate was filtered through a 40-μm mesh (Falcon® 40 μm Cell Strainer, 352340, Corning) to a 50-mL Falcon tube on ice. An additional 5 mL of low sucrose buffer was used to rinse the douncer tube and cell strainer. The filtered homogenate was further mixed with a handheld homogenizer (VWR® 200 Homogenizer) at a speed of ~1000 rpm to brake nuclei clumps for 5 s. After homogenization, a serological pipet filled with 12 mL of high sucrose buffer (1 M sucrose, 10 mM HEPES, 3 mM MgAc, and 1 mM DTT in $ddH_2O$, pH8) was placed underneath the lysate and disconnected from the pipettor, and the buffer was released from the serological pipette by gravity and set on ice. When most of the high sucrose buffer was released to form a density layer underneath the homogenate, the serological pipet was retrieved along the wall of the Falcon tube gently, without disturbing the low-high sucrose interface. The Falcon tube was capped and placed in a swing bucket to be centrifuged at $3200 \times g$ for 30 min at 4 °C. At the end of a spin, the supernatant was decanted quickly without tabbing, and 1 mL of resuspension buffer (0.02% BSA in 1X PBS, pH7.4) containing 200 units of RNase inhibitor was added to the Falcon tube to rinse the nuclei. Slow pipetting was employed to resuspend nuclei along the Falcon tube wall below the 5-mL mark to preserve nuclei integrity. Specifically, nuclei were rinsed off the wall in courses of 2 s per trituration for 20 times total per tube. The Falcon tube was then capped and spun at $3200 \times g$ for 10 min at 4 °C. At the end of spin, the supernatant was removed by gently tabbing the tube until no visible liquid drop was left behind, and 200 μL of resuspension buffer was added to each sample to collect the nuclei. The nuclei suspension was filtered through a 35-μm mesh (Cell Strainer Snap Cap, 352235, Corning) twice and counted on a hemocytometer by trypan blue staining. During counting, the size and quantity of myelin and

other debris were visually inspected under the scope, and the suspension was filtered 1–3 more times through the 35-µm mesh if necessary. Only round and dark-blue stained nuclei were considered of good quality and included in the final count.

**cDNA library and sequencing.** Most single-nucleus libraries (38) were prepared using 10x Genomics Chromium Single Cell 3′ Library & Gel Bead Kit v3, though four libraries were done in v2 chemistry following the manufacturer's protocol. Briefly, nuclei suspensions were prepared as described above and diluted with resuspension buffer to achieve the desired concentration, and then loaded into Chromium Controller to generate Gel-beads in Emulsion (GEM). For both cDNA amplification and library sample index PCR, 12 cycles were used. Most libraries were sequenced on Illumina Novaseq S2, but some used Illumina Miseq, Hiseq 2500, or Hiseq 4000, according to the manufacturer's protocol; see Supplementary Fig. 2 and Supplementary Data 1 for details.

**Alignment.** The raw sequencing reads were aligned to a marmoset genome assembly, ASM275486v1 (GCA_002754865.1). To build a reference package suitable for analyzing both unspliced pre-mRNA and mature mRNA in the nuclei, as well as to include sequences of mitochondrial genome, marmoset DNA sequence (FASTA) and annotation (GTF) files were acquired from the Ensembl release-95 and modified as follows. The complete mitochondrial sequence (NC_025586.1, GenBank) and its annotation[71] were manually added to the FASTA and GTF files. Next, a pre-mRNA GTF was made by replacing "transcript" with "exon" as the feature-type entry in the original GTF before making a reference package with CellRanger software (v3.0.2, 10x Genomics). This custom-built reference package was then used in CellRanger (version 3.1.0, 10x Genomics) to align sequencing reads for all samples. The option to estimate cell number automatically was used for most of the samples, unless otherwise specified (see Supplementary Data 1 for details). A filtered cell barcode-to-gene feature matrix was generated from the software and used for downstream analysis (Supplementary Fig. 2a).

**Preprocessing and quality control.** The matrix was loaded to create an object in Seurat v3[72]. Cells with <200 genes, >5000 genes, or >5% of counts mapped to the mitochondrial genome were excluded. Genes observed in <5 cells were excluded. The filtered raw count matrix was then log normalized (ln(counts × 100,000 + 1)) within each cell and scaled to account for differences in sequencing depth with Seurat. Next, DoubletFinder[14] was used to estimate and remove putative doublets to mitigate technical confounding artifacts in droplet-based sequencing data analysis. The top 3000 variable genes calculated by Seurat were used in linear dimension reduction (principal components analysis, PCA), and the top 30 principal components (PC) were used for clustering at low resolution (parameter = 0.4) to define coarse cell types. These unsupervised clusters were used to provide a quick cluster annotation for homotypic doublet probability modeling in DoubletFinder. The doublet rate was estimated by fitting a linear equation over a multiplet rate table provided by 10x Genomics. The rate = (0.0008 × cell.number + 0.0527)/100 was used to calculate a Poisson distribution with and without homotypic doublet proportion to generate low confidence (DF.found.1) and high confidence (DF.found.2) doublet annotation. Unless otherwise specified, pN = 0.25, pK = 0.005, and automatic doublet removal based on DF.found.2 annotation were used, as the first line of screening.

In parallel, SoupX[13] was used for ambient RNA background correction. Taken from the output of the 10x Genomics pipeline (raw_feature_bc_matrix), the ambient RNA from empty droplets that contained <10 unique molecular identifiers (UMI) were profiled, and the "soup" contamination fraction was calculated for each cluster. Given that the nuclear transcriptome was profiled, genes that mapped to the mitochondrial genome could be considered as a marker of ambient input. Specifically, the top mitochondrial genes (species with >1000 accumulated counts across profiled empty droplets) were used to estimate the global contamination fraction and adjust the raw count matrix. Next, the cell barcode that passed the DoubletFinder was used as an index to subset the SoupX-corrected matrix to generate a new matrix as our downstream input (Supplementary Fig. 2a). For individual samples, a Seurat object was created, and the index labels (IL01_uniqueID, IL02_species, IL03_source, IL04_sex, IL05_ageDays, IL06_tissue.1 (coarse category), IL06_tissue.2 (developmental category), IL06_tissue.3 (fine category), IL07_location, IL08_condition, IL09_illumina, IL10_chemistry, IL11_batch, IL12_LMinDays, IL13_LMax-Days, IL14_dataset, IL15_annotation) were added to the metadata as cell attributes. For each sample, 90% of nuclei were randomly selected, then all 42 samples were merged for downstream comparison. The remaining 10% of nuclei were set aside for classification assessment and validation (Supplementary Fig. 8).

**Clustering and visualization.** Clustering was performed using Seurat, iteratively with different parameter sets, to understand the data structure. A merged Seurat object was created from the 42 samples, and the aggregated raw count matrix was log-normalized and scaled again as stated above.

**Preliminary exploratory data analysis.** The top 3000 variable genes calculated by Seurat were used in PCA. In the first round of clustering and visualization, 100 PC were computed and used for Harmony (v1)[15] to integrate different samples, specifically variability over the IL01_uniqueID attribute, with default setting (theta = 2, lambda = 1, sigma = 0.1). The UMAP space and nearest-neighbor analyses were calculated on the top 50 Harmony embeddings with resolutions from 0.4–1.2. Cell barcodes of a cluster of nuclei annotated as "low quality," which resided at the center of the 2D UMAP (H50), were recorded; these nuclei had a high percentage of reads mapped to the mitochondrial genome, low RNA counts and features, and/or expressed genes that mapped to multiple canonical markers of different cell types. No single set of parameters can adequately separate ~500 K nuclei to identify subclusters from all major cell types simultaneously, as either oversplitting for low complexity cells (e.g., glia) or under-splitting for high complexity cells (e.g., neurons) would result. Therefore, a stepwise clustering approach was used, whereby major cell classes (neurons and oligodendrocytes, etc.) were first identified and then divided into subclusters for each class (Fig. 2).

**Level 1 quality control and analysis.** To divide nuclei into classes and facilitate artifact identification, nuclei were first classified using a set of parameters that do not highlight granular detail. In this round of clustering, only 50 PC for Harmony were computed to perform linear correction over IL01_uniqueID, as the elbow plot from the preliminary analysis showing the standard deviation stopped visually decreasing after the top 50 PC. The top five Harmony-corrected embeddings (H5) were used for Seurat to learn the UMAP and find cell classes at a low resolution (0.2). Canonical cell-type markers (*PTPRC* for immune cells, *PDGFRA* for OPC, *MAG* for oligodendrocytes, *GFAP* and *SLC1A2* for astrocytes, *LEPR* and *CEMIP* for vasculature and meningeal cells, and *CNTN5* and *NRG1* for neurons) annotated 6 of the classes unambiguously. One cluster in the middle of the H5 UMAP had mixed expression of canonical markers, which suggested an artifact. The "low quality" cell barcodes that were found from the H50 condition (defined above) were overlaid on the H5 UMAP, which exclusively highlighted the putative artifact cluster. These nuclei were removed from further analysis, although the original UMAP embeddings were maintained for plotting purposes (Fig. 1e and Supplementary Fig. 4).

**Level 2 quality control and analysis.** Nuclei that passed Level 1 QC were divided into five classes (MIC, OPC, OLI, VAS/AST, NEU) based on

the H5 UMAP result. Astrocytes and vasculature/meningeal cells were pooled into a single class prior to subclustering to facilitate artifact identification (Fig. 2b). Shared features in this class of cells are potentially explainable by their close association at CNS barriers (blood–CSF, blood–brain, and CSF–brain interfaces). For each class, log-normalization and scaling were repeated from the divided raw count matrix, and the top 3000 variable genes were used for 50 PC computation, Harmony correction over IL01_uniqueID, UMAP learning, and clustering, as described above. The clustering resolution was iteratively increased from low to high (0–1.2), and clustering stability was tracked with clustree (v0.4.3). NEU Level 2 clustering stability was also tracked by calculating the Jaccard index at resolution 2 (res.2, Supplementary Fig. 13a, b)[42]. Aided by the branch visualization provided by clustree, a tentative resolution that was relatively stable was selected, then differentially expressed gene (DEG) analysis on the clusters found with this parameter set was performed. The expression patterns of the top-expressed genes for each cluster within and across classes were checked, and artifact clusters were manually imputed. Doublets tended to form small distinct clusters in the UMAP plots that branched early in the clustering tree analysis with a low splitting resolution, had mixed canonical marker-gene expression, and had similar expression patterns to cells in other partitions; thus, these doublets could be easily spotted and removed. For putative doublets within each class, additional rounds of DEG analysis were performed as necessary. Each time nuclei were removed, basic normalization, scaling, Harmony, and UMAP learning were repeated. To control for over-splitting, for clusters that appeared to be a single pile in the 2D UMAP space but were annotated into >1 cluster, additional rounds of DEG analysis were performed to see if binary labeling markers could be found. In addition, clustering was projected onto a 3D UMAP space to ensure effects were not masked due to overcrowding in 2D. This strategy helped to further elucidate cluster associations, aid decision-making with respect to groups of clusters that should be tested further, and spot potential gradient changes among clusters. If unique and/or binary patterns could not be found in the current splitting resolution after these steps were performed, a step lower in resolution on the clustering tree was examined, and the analysis was repeated. The following compound naming convention to label the 87 sub-clusters was used: general category in numeric order, major tissue or location contributor for each cell type, and binary marker combination where applicable.

**Preparation of objects for cross-cluster analysis.** Once the sub-clustering and UMAP embedding were finalized for each cell class, several annotated objects were created to facilitate downstream analysis and comparison. To enable cluster overview, compare global and local gene expression, and classify the 10% set-aside data, an object containing all 87 subclusters and 50 nuclei per cluster was prepared by random sampling (C50 object). For white and gray matter comparison, 4000 nuclei were randomly sampled from each tissue type and pooled into two objects (Fig. 2e), WM (containing 24,000 nuclei, including fWM, tWM, pWM, aCC, pCC, and OpT) and GM (containing 20,000 nuclei, including fCTX, tCTX, pCTX, oCTX, and CgG).

**Data visualization.** Unless otherwise specified, gene expression values in the dot plots and heatmaps were averaged, mean-centered, and z-score-scaled (from −1.5 to +1.5, to which values below or above these levels were assigned). Dot size indicates the percentage of nuclei in the subcluster in which the gene was detected. Among the nuclei in which a given gene was detected, the expression level was mean-centered and scaled. For aggregated gene lists or gene module expression, a relative color scheme was used to indicate the level of expression, from low to high. For dendrogram creation, the top 50 enriched genes calculated in Level 1 analysis were used to calculate Euclidean distances, using "hclust(dist())" functions in R. To aid cluster tracking, branches

of the dendrogram were reordered and colored to show the origin of cell classes while retaining the tree structure.

## Pseudotime analysis
Monocle3 (v0.2.0) was used to construct nuclei trajectories based on transcriptomic distance[23]. The OLI Seurat object with finalized UMAP from Level 2 analysis was converted to a Monocle object. All index labels and cell attributes, cluster assignment, and UMAP embeddings were transferred. A partition was then assigned for each nucleus by the cluster_cells() function, and a principal graph was fit within each partition by the learn_graph() function. From the principal graph, Monocle3 defined a unitless transcriptome progression along the learned trajectory as "pseudotime." The distance between two given points along the trajectory path indicates the amount of expression change required to connect the ends. The starting point of pseudotime is self-defined by the order_cells() function. Based on prior knowledge[48], the node at the side of the $ENPP6^{high}$ oligodendrocyte cluster was selected as the starting point. To visualize gene expression dynamics along pseudotime, the plot_genes_in_pseudotime() function was used to fit a spline using the following trend formula: "~sm.ns(Pseudotime, df=3)". The calculated pseudotime value was extracted for further analysis as indicated in the figure legend (cds@principal_graph_aux@listData[["UMAP"]][["pseudotime"]]).

## Gene module analysis
Monocle3 was used to find and group genes by similarity along the learned principal graph[23]. Genes that passed Moran's *I* statistic spatial test (<5% FDR) over the *k*-nearest-neighbor graph (Knn, k = 25), or trajectory learned principal graph (PG) by Monocle3 graph_test() function, were used for module assignment. Genes were grouped into modules identified in each type of graph test by the find_gene_modules() function with a resolution of $10^{-3}$. The list of genes of each module was then aggregated and added back to the Seurat object through the AddModuleScore() function and visualized in Seurat v3. Genes that mapped to the mitochondrial genome were dropped before performing gene ontology and pathway analysis. See Supplementary Data 2 for the full list.

## Gene ontology (GO) and pathway analysis
The list of genes from the selected modules and/or DEG, discovered as stated above, were used for various pathway analysis. The GO analysis for marmoset was performed by gprofiler2 (v0.1.9)[73] with the gost() function. The database for "cjacchus" was used, electronic GO annotations (IEA) were included, and g:SCS threshold was used for multiple testing correction as suggested by gprofiler2. Three major sub-ontologies—Molecular Function (MF), Biological Process (BP), and Cellular Component (CC)—were included in the analysis. Additional annotations from the KEGG and HP databases were included when available. Terms that passed a significance cutoff of $p = 0.05$ after correction were filtered at the following criteria in case of over-crowding. The parent terms were removed if child terms from the same branch were present in the same list, and if the term had at least one parent term in the database prioritized, as terms lower in each branch are usually more specific and informative. For terms that passed filtering, the corrected *p* value and fold enrichment were plotted. The fold enrichment was calculated as follows: (intersection_size/query_size)/(term_size/effective_domain_size).

## NicheNet ligand-receptor-target analysis
Potential intercellular communication in WM and GM was modeled using nichenetr (v0.1.0)[59] The cross-partition objects for WM and GM generated as described above were used for this analysis (Fig. 2e). Bioinformatic resources and protocols were modified from https://github.com/saeyslab/nichenetr. Briefly, NicheNet studies intercellular communication computationally by leveraging known ligand-to-

receptor and receptor-to-target relationships in its database. It allows the prediction of interactions between ligands expressed by "sender" cells and receptors expressed by "receiver" cells, and models how these interactions might drive gene expression changes in cells of interests (target DEG in the receivers).

We hypothesized that differences between WM and GM might partially explain tissue-type-specific subpopulations of microglia, OPC, and astrocytes. Therefore, NicheNet was used to test if transcriptome changes between subclusters could be explained by environmental signals from nearby cells. WM and GM differentially enriched microglia, OPC, and astrocyte subpopulations were defined as receivers in each test, and DEG between MIC1 and MIC3, OPC1 and OPC3, and AST1 and AST3 were derived. DEG were filtered at adjusted $p$ value (<0.05) and absolute log (ln) fold-change (>0.25). Potential senders were defined from clusters with >50 nuclei in the same tissue type as each receiver, and genes detected in >10% of the nuclei in a cluster were kept for further analysis. Preconstructed databases were downloaded for ligand_target_martix [https://zenodo.org/record/3260758/files/ligand_target_matrix.rds], ligand_receptor_database [https://zenodo.org/record/3260758/files/lr_network.rds], and weighted networks [https://zenodo.org/record/3260758/files/weighted_networks.rds]. Gene names for these databases were built with human data, therefore human genes with one-to-one orthologs were translated to marmoset gene names with BioMart. The weighted ligand-receptor-target (LRT) matrix was thereby constructed, with weighting factors implemented so that informative data sources maximized prediction accuracy in the final model. The list of sources used to build this database and the method to calculate the weighted scores have been specified[59]. After "expressed ligands" were defined for senders and "expressed receptors" for receivers (>10% detection rate), the existence of ligand-target pairs was established. The ligand-target pairs were ranked based on the presence of the target genes (defined by the receptor-target database) in the calculated DEG using the predict_ligand_activities() function. The filtered DEG present in the top 200 predicted target genes per ligand were kept for further ligand-target analysis.

For visualization of this complicated intercellular interaction, Circos plots were generated. First, the lists of ligand-target pairs in WM and GM were compared and divided into three categories (GM, WM, and shared), and a Venn diagram was generated for each type of receiver (microglia, OPC, and astrocytes; Fig. 8c). The unique ligand-target pairs for each environment were plotted in enlarged Circos plots, and the shared ligand-target pairs in smaller Circos plots, for categorical visual reference (Supplementary Fig. 41e, h, k, bottom panels). The shared Circos plots were also enlarged to aid the visibility of individual genes (Supplementary Fig. 42). Since any given ligand might be expressed by >1 cell type, each ligand was assigned to the cell cluster that ranks highest in the product of detection rate (%) and expression level ($z$-score scaled). Since the probability is low for any ligand to be assigned to a particular cell type with this strategy, it is sufficient to map intercellular interaction qualitatively and categorically (see Supplementary Data 4 for full report). The senders by Level 1 classes (MIC, OPC, OLI, AST, VAS, and NEU) were colored, and pie charts tabulating the proportion of unique (Supplementary Fig. 41) and shared (Supplementary Fig. 42) ligand-target pairs were generated. For each Circos plot, up to 100 weighted ligand-target interactions were presented to limit overcrowding. For shared ligand-target pairs, an inter-categorical agreement was calculated and presented in Sankey diagrams using the networkD3 (v0.4) package.

## Gene set enrichment analysis

Themes were collected from the following sources, after which the aggregated score was calculated by Seurat v3 AddModuleScore() function. Gene groups included ion channels, scavenger receptors (SCAR), and histocompatibility complex (HLA) from the HUGO Gene Nomenclature Committee (HGNC). Cell-cycle genes were pulled from

the built-in gene list in Seurat (cc.genes.updated.2019) for S and G2M phases. Genes enriched in the G0G1 phase[44] and the list of human transcription factors[74] were informed by the literature. Neurological disorder-associated genes were acquired from the database curated in the Ingenuity Pathway Analysis (IPA) software. See Supplementary Data 4 for the full gene lists.

## Expression-weighted cell-type enrichment (EWCE) analysis

Cellular phenotypes of neurological disorders were calculated by EWCE[62]. Briefly, the expression of a list of $n$ genes associated with a disease or disease category was compared with those in 100,000 randomly selected lists of $n$ genes from the background. The proportional expression of genes associated with each cell type was calculated to compute the probability of enrichment. Tested disease/disease categories were: organic mental disorder, CNS tumor, cognitive impairment, psychological disorder, autism spectrum disorder or intellectual disability, Huntington's disease, Alzheimer's disease or frontotemporal dementia, seizures, schizophrenia spectrum disorder, encephalitis, cerebrovascular dysfunction, stroke, parkinsonism, amyotrophic lateral sclerosis, multiple sclerosis, white matter abnormality, migraine, Charcot-Marie-Tooth disease, abnormality of meninges, and Zellweger syndrome. After Benjamini-Hochberg correction, the significance of cell-type enrichment was denoted with * for $q < 0.05$, and ° for $q < 0.1$

## Cross-cluster comparison and validation

**Comparison between marmoset subclusters.** The C50 object was used to assess transcriptomic similarity across all 87 subcluster pairs. The expression levels of all genes within each subcluster were normalized and averaged before calculating the linear correlation. The lm() function was used in R, and the adjusted $r^2$ values were extracted for heatmap plotting. Similarly, the transcriptomic distances between all subcluster pairs were assessed by counting the number of DEG, both increased and decreased, between them. DEG were filtered by their log (ln) fold-change (>0.25) and detection frequency (detected in ≥10% of nuclei).

**Comparison between clusters from different species.** Deposited data from zebrafish, mouse, and humans (Table 1) were reanalyzed. The top 3000 variable genes were used to calculate 50 PC and harmonized over sample ID if available in the deposited data. Gene names for each species were translated to human gene names using the one-to-one orthologs index with BioMart. The expression levels of all humanized genes within each compared cluster were normalized and averaged before calculating the Pearson's correlation coefficients, which were used for heatmap plotting.

**Comparison between cleaned classifiers and semi-cleaned 10% set-aside data.** To assess the reproducibility of our derived subclusters, the C50 object was used as an unbiased classifier to annotate the 10% of nuclei that had been set aside a priori, as described above. A total of 61,852 nuclei were compared. The nuclei were intentionally over-split using the top 5000 variable genes (maximum gene number detected per nucleus) to calculate 100 PC, harmonized over IL01_uniqueID labels. All 100 Harmony embeddings were used to compute UMAP and nearest-neighbor distances with extremely high resolution (12; normal suggested resolution range is 0.4–1.2). A total of 140 clusters were found, and the expression levels of genes within each cluster were normalized and averaged before calculating the Pearson's correlation coefficients across each pair of subclusters in the two datasets, which were used for heatmap plotting.

## Histology

**H&E staining.** Sections used for histology were archival formalin-fixed, paraffin-embedded (FFPE) contained in an in-house marmoset tissue

library. Serial sections were cut at 5 μm from the brain and spinal cord tissue blocks from each animal using a Leica RM2235 Manual Rotary Microtome. Sections were mounted onto Superfrost⁺/Colorfrost⁺ microslides (Daigger, 75 mm × 25 mm, #EF15978Z) and stored at room temperature. Before staining, sectioned slides were deparaffinized with xylene three times for 5 min each, rehydrated with EtOH (100, 70, 50% for 5 min each), and rinsed in DI H$_2$O for 5 min at RT. Hematoxylin & eosin staining was subsequently performed. Hematoxylin (basophilic) stains nucleic acids and nuclei purple, whereas eosin (acidophilic) stains cytoplasmic components of the cell pink. For hematoxylin staining: slides were dipped one-by-one in hematoxylin (Leica, 100% Surgipath SelecTech Hematoxylin 560MX, 3801575) for 1 min and immediately placed in running tap water to stop the reaction and rinse off excess stain. Slides were then dipped one-by-one for 30 s in the Define solution (Leica, Surgipath SelecTech Define MX-aq, 3803598) to reduce the intensity of hematoxylin and immediately placed in running tap water to stop the reaction. Sections were dipped one by one in Blue Buffer solution (Leica, Surgipath SelecTech Blue Buffer 8, 3802918) for 1 min to change tissue color to blue. This reaction was stopped by placing slides in 80% ethanol for 1 min. For eosin staining: slides were dipped together for 30 s in eosin (Leica, Surgipath SelecTech Alcoholic Eosin Y 515, 3801615) and put in 100% ethanol for 1 min, three times. Coverslips were mounted on slides right away using VectaMount Permanent Mounting Medium (Vector Laboratories, #H-000-60).

**Immunostaining.** For immunostaining, deparaffinized and rehydrated slides were submerged in 1X antigen retrieval solution (AG unmasking solution, H-3000, Vector) and placed in a tissue streamer for 2 h to perform heat-induced epitope retrieval (HIER). At the end of HIER, sections were left cooled for 10 min inside the steamer. The sections were transferred to 1X TBS (pre-cooled at 4 °C) for 5 min, then blocked for endogenous peroxidase by submersion in 3% H$_2$O$_2$ for 10 min, and then rinsed in 1X TBST (0.05% tween 20 in 1X TBS) for 1 min at room temperature (RT). A parafilm pan was used to demarcate the surrounding of each section after removing excess liquid with Kimwipes, and 200 μL of blocking solution (Protein block, serum-free, X090930-2, Dako) was applied per section for 30 min at RT. Primary antibodies were diluted in antibody diluent (S080983-2, Dako) and applied to sections overnight at 4 °C. Sections were rinsed in 1X TBST once for 1 min, then twice for 5 min, and appropriate secondary antibodies were applied for 30 min at RT. Sections were rinsed in 1X TBST once for 1 min, then twice for 5 min, and 200 μL of immunoperoxidase development solution (DAB Substrate Kit, ab64238, Abcam) was applied per section for 45 s at RT. Chromogenic reactions were stopped by switching to DI water, and sections were rinsed with tap water for 5 min at RT. For double staining, 200 μL of alkaline phosphatase substrate solution (Vector® Blue Substrate Kit, SK-5300, Vector) was applied to each section for 5 min at RT. Chromogenic reactions were stopped by switching to DI water, and sections were rinsed with tap water for 5 min at RT. The following antibodies were used: mouse anti-PLP (Bio-Rad, MCA839G, 1:200), rabbit anti-IBA1 (Wako, 019-19741, 1:200), mouse anti-IBA1 (Sigma, SAB2702364, 1:100), rabbit anti-SLC15A1 (Sigma, HPA002827, 1:100), rabbit anti-OLIG2 (Chemicon®, AB9610, 1:200), rabbit anti-GFAP (Dako, Z033429-2, 1:200), PV Poly-HRP Anti-Rabbit IgG (Leica, PV6119, 1:1), PV Poly-HRP Anti-Mouse IgG (Leica, PV6114, 1:1), ImmPRESS®-AP Horse Anti-Rabbit IgG Polymer (Vector, MP-5401-50, 1:1), ImmPRESS®-AP Horse Anti-Mouse IgG Polymer (Vector, MP-5402-50, 1:1), Goat anti-Rabbit IgG (H + L) Cross-Adsorbed Secondary Antibody, Alexa Fluor 594 (Invitrogen, A-11012, 1:400).

**Fluorescence in situ hybridization (FISH).** For FISH, HIER-treated slides as described above were submerged in 1X PBS for 5 min, then treated with 10 μg/mL proteinase K (Proteinase K, recombinant, PCR Grade, Roche, 03115879001) for 10 min at 37 °C. At the end of

incubation, slides were rinsed in 1X PBS for 1 min, submerged in fresh 1X PBS for 5 min, and rinsed in 1X TBST for 1 min at RT. A parafilm pan was used to demarcate the surrounding of each section after removing excessive liquid with Kimwipes. In the case of combining immunofluorescence staining and FISH, 200 μL of blocking solution was applied per section for 30 min at RT, and primary antibodies were diluted in antibody diluent and applied on sections overnight at 4 °C. Sections were rinsed in 1X TBST once for 1 min, then twice for 5 min, then post-fixed with 4% PFA (made from 32% paraformaldehyde aqueous solution, 15714-S, Electron Microscopy Sciences, in 1X PBS) for 10 min at RT. Slides were rinsed in 1X PBS for 1 min, then submerged in fresh 1X PBS twice for 5 min, prior to FISH. Slides were incubated at 37 °C for 10 min with a 30% probe hybridization buffer constituted of 30% formamide (F9037, Sigma-Aldrich), 5X SSC (46-020-CM, Corning), 9 mM citric acid (C0706, Sigma-Aldrich), 0.1% Tween 20 (1610781, Bio-Rad), 50 μg/mL heparin (H3393, Sigma-Aldrich), 1X Denhardt's solution (D2532, Sigma-Aldrich), and 10% dextran sulfate (D8906, Sigma-Aldrich) in ddH$_2$O. At the end of pre-hybridization, excess hybridization buffer was removed by blotting the edges on Kimwipes.

HCR probe set (Hybridization chain reaction v3.0)[75] (targeting marmoset *OLIG2* (PRL850, Molecular Instruments) and *MUSK* (PRI863, Molecular Instruments) were prepared in a 30% probe hybridization buffer. In a leveled and humidified chamber, 1.2 pmol probe solution (250 μL per brain section) was applied onto a slide, covered with a parafilm, and then incubated overnight at 37 °C. At the end of incubation, the parafilm was floated off by submerging the slide in 30% probe wash buffer (30% formamide, 5X SSC, 9 mM citric acid, 0.1% Tween 20, 50 μg/mL heparin in ddH$_2$O) at 37 °C. After parafilm removal, slides were incubated with 75%, 50%, and 25% serial diluted 30% probe wash buffer in 5X SSC-Tw containing 0.1% Tween 20 for 15 min each at 37 °C. Slides were then brought to RT and submerged with 100% 5X SSC-Tw for 5 min and dried by blotting the edges with Kimwipes. In a humidified chamber, 200 μL of amplification buffer (5X SSC, 0.1% Tween 20, and 10% dextran sulfate in ddH$_2$O) was applied to the slide and incubated for 30 min at RT. Snap-cooled (heat to 95 °C for 90 s and cool to RT for 30 min) hairpin H1 and H2 were kept in the dark chamber and reconstituted in 150 μL of amplification buffer. Excessive amplification buffer was removed from the slide by blotting the edges with Kimwipes, and 150 μL hairpin solution was applied onto each section and covered with parafilm overnight at RT in a humidified dark chamber. At the end of incubation, the parafilm was floated off by submerging the slide in 5X SSC-Tw at RT, and the excessive hairpin solution was removed by incubating the slide in 5 C SSC-Tw 3 times for 15 min each at RT. In the case of combining immunofluorescence and FISH, slides were incubated with matching Alexa-conjugated secondary antibody were prepared in antibody diluent for 1.5 h at RT. At the end of incubation, sections were rinsed in 1X TBST once for 1 min, then twice for 5 min. Sections were then incubated with 1X PBS for 5 min before applying TrueBlack Lipofuscin Autofluorescence Quencher (23007, Biotium) for 5 min at RT to reduce background. Sections were rinsed in 1X PBS twice for 1 min, then once for 5 min. The excessive liquid was removed by suction, 50 μL of mounting solution with nuclei stain (DAPI Fluoromount-G®, 0100-20, SouthernBiotech) was applied, and the slide was covered with glass (Premium Cover Glasses, EF15972L, Daigger Scientific).

## Microscopy and cell quantification

On hematoxylin & eosin-stained slides from each animal, boxes were drawn around each 2-mm area of interest in the brain and spinal cord. Each region was imaged at 10X magnification with a Nikon Eclipse Ci microscope. The number of cells in each area of interest was counted using Fiji ImageJ. A color image threshold of 0–165 was chosen to highlight an optimal number of cells and limit the number of falsely identified cells. Using the "Analyze Particles" function, the number of cells at the chosen threshold was counted automatically, with the pixel

size range set to 20–200, and the circularity range to 0.25–1.00, to exclude as much background noise and as many linear particles as possible. After the automatic counting, falsely identified cells were manually deleted, and a new count was saved. Additional cells not detected by the automatic counter were manually added using the CellCounter plugin. A final image with automatic and manual cell count markers was saved, and the total number of cells (including manual deletions and additions to the automatic count) was recorded. Cell counts were normalized to the imaged area to get the density of nuclei per tissue type in each animal. The averaged nuclei density per mm² in each tissue type was then quantified. To estimate the initial number of nuclei for single-nucleus sequencing per cylinder of 2 mm diameter and 3 mm height (V = 3π μL), the averaged 2D density measured from hematoxylin & eosin-stained sections was used after multiplication by section thickness (5 μm). The percentage of nuclei recovered after Level 1 quality control for each sample was then plotted, each circle representing the percentage of one sample (Supplementary Fig. 1e).

For particle and morphology analysis, slides with PLP1/IBA1 double staining were imaged at 20X with Nikon Eclipse Ci microscope and analyzed with Fiji ImageJ. The color image was split into RGB channels, thresholding was done on the blue channel to highlight the IBA1⁺ area, and the "Fill Holes" function, located under the Process-Binary tab, was applied to the image. Areas with artifacts, such as tissue folding, were manually corrected on the binary image. The "Find Connected Regions" function, located under the Plugins-Process tab, was then applied with the following parameters: allow a diagonal connection, display one image for all regions, display results table, and minimum number of points in a region >450. In parallel, the "Analyze Particles" function, under Analyze tab, was applied to the artifact-corrected binary image, through which count, area, perimeter, and circularity were recorded to quantify the morphology of IBA1⁺ cells (Fig. 3f–h).

For fluorescent imaging, slides stained with anti-OLIG2 (Alexa-594 nm), HCR-*OLIG2* (Alexa-488 nm), HCR-*MUSK* (Alexa-647 nm), and DAPI were imaged at 40X (EC Plain-Neofluar 40x/1.30 Oil DIC objective) with LSM 880 (AxioObserver, Zeiss) laser scanning confocal microscope equipped with 405 nm diode, 488 nm argon, 594 nm HeNe, and 633 nm HeNe lasers. A single image was taken with 0.21-μm pixel size, 2 averaging, and 0.85 airy unit. Pseudocolors were assigned to each channel with detection wavelength 415–467 nm in blue, 490–553 nm in green, 597–642 nm in white, and 642–695 nm in red.

**Reporting summary**

Further information on research design is available in the Nature Research Reporting Summary linked to this article.

## Data availability

Raw and processed datasets are submitted to Gene Expression Omnibus (GEO) under session GSE165578. Data can also be visualized at https://cjpca.ninds.nih.gov. Source data are provided with this paper. Databases and datasets used in the study are listed in "Table 1" also with the following accession codes and links: GSE121654, GSE132166, GSE75330, SRP135960, GSE52564, phs001836, GSE97930, GSE104525, GSE73721, GSE118257, GSE180759, Marmoset Gene Atlas [https://gene-atlas.brainminds.riken.jp/], and Marmoset Brain Mapping [https://marmosetbrainmapping.org/].

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

## Acknowledgements

We thank Dr. Pascal Sati, Dr. Maxime Donadieu, and Mr. Roger Depaz (NINDS Translational Neuroradiology Section) for their help in collecting in vivo MRI data and monitoring animals. We thank Dr. Ariel Levine (NINDS), Ms. Kaya J.E. Matson (NINDS), Dr. Yuesheng Li (NHLBI), Dr. Poching Liu (NHLBI), Dr. Yan Luo (NHLBI), Dr. Qing Wang (UCLA), and the Adelson Medical Research Foundation (AMRF) Functional Genomics Resource (UCLA) for expertise, advice, and assistance with sequencing. We thank Dr. Chang-Ting Lin for his insights into quantitative data analysis. We thank the AMRF's Program in Neurodegenerative Diseases–Multiple Sclerosis (APND-MD, DSR) and the Intramural Research Program of NINDS (ZIA NS 003119-08, DSR) for funding. This work utilized the computational resources of the NIH HPC Biowulf cluster.

## Author contributions

J.-P.L. and D.S.R. designed the study and interpret the results. J.-P.L., D.S.R., and S.J. developed protocols. J.-P.L and D.S.R. wrote and prepared manuscripts. J.-P.L. acquired, processed, and analyzed snRNA-seq data. J.-.P.L. and H.M.K. acquired, processed, and analyzed histology data. J.-P.L. and Y.S. cleaned and processed published datasets. D.H.G. and R.K. analyzed the data. D.S.R. supervised the study.

## Funding

## Competing interests

The authors declare no competing interests.
