## [Peer Review File · Nature Communications]

Transcriptomic architecture of nuclei in the marmoset CNSREVIEWER COMMENTS

Reviewer #1 (Remarks to the Author):

This manuscript from Reich lab offers an extensive inspection of cellular composition and location-specific specialization of white matter compared with gray matter profiling many nuclei in total from 19 tissue types from marmoset central nervous system. The authors were able to successfully map the effect of gray vs. white matter on primary glial cell types and functional heterogeneity. It will be interesting to study the role of this heterogeneity in neurological disorders and this work marks great information for that. The effort from the authors to develop an atlas CjPCA seems to be valuable as a resource to the field in general. This work is an important step forward to understanding the cellular composition and heterogeneity in white and gray matter and will be a rich resource for others to build on.

The details and statistics of the samples and sequencing data are well described in the main text as well as supplementary data. The overall presentation is satisfactory, and the methodology is elaborated in detail which is appreciated. This is a well written article - the design, methodology and conclusion seem to be reasonable, reliable, and valid, though with a few minor concerns that are mentioned below.

1. Although the abstract is well written, given how extensively the authors have investigated the data, a schematic summarizing the analysis workflow in the main body of the manuscript will be helpful for readers to keep the track of the multiple analyses performed. Although the list of resources provided in the manuscript is useful, I will suggest providing one main diagrammatic workflow and another just for the data analyses part mentioning the steps and key resources.
2. How do authors address the effect of technical variation, that is sort of inevitable for such an extensive study, on biological interpretations especially when marmoset has less information available compared to mouse and humans?
3. The Harmony integration is well implemented and elaborated extensively but a UMAP image split by batches (without batch correction) should be provided in the manuscript. This will be helpful for readers to notice the occurring batch effect(s) if any and compare it with harmony correction.
4. Since integration is the one of the most crucial steps to interpret the data but it comes with no universal set of parameters to identify subtypes ideally due to over or under splitting of specific cells, it is advised that authors should incorporate another integration method like RPCA and compare it with Harmony specifically at the subcluster level.
5. The authors did not mention the nuclear RNA integrity check. It would be worth mentioning the quality measures for the RNA samples.
6. I will highly recommend providing a table containing cluster assignment for each nucleus. This will be helpful to compare this dataset with other studies.
7. There are some typos in the manuscript. For example, Figure S5 is mislabeled as S6 in the supplementary figure. In line #200, it should be Fig S17B.

Reviewer #2 (Remarks to the Author):

The authors are to be commended in their approach and thoroughness in the computational methodologies used to analyze and validate their data. However, the depth to which these methods were applied did not correlate to an increase in the scientific/biological significance of the finding. This study can be groundbreaking in understanding regional heterogeneity of neuronal and glial cells in multiple regions of the primate brain but falls short in delivering the results to a sufficient depth, complexity or significance.

This paper presents an interesting paradox whereby a comprehensive methodology in the computational analysis of data was applied, yet relatively superficial insights into the relationship between cell-type heterogeneity and function were described. The characterization of each glial cell type provides a fascinating GM/ WM diversity in their gene expression and potential function but stops short of addressing the impact or significance of the finding. Instead, this paper reads more as a comprehensive pipeline for the computational/ bioinformatic validation of cellular heterogeneity rather than a hypothesis-driven exploration of the relationship between heterogeneity and function. Every result section appears to be stand-alone with very little connecting them except for the pervasive GM vs. WM theme, which is a particular concern.

In its current form, this manuscript might be better presented as a resource for marmoset data or transcriptomic analysis pipeline. Otherwise, in light of the sheer amount of data presented, contrasting with the superficial analysis of the data (not in methodology but content and significance), the authors should seriously consider splitting this manuscript up into separate, more digestible components with more in-depth analysis of each component.

Additionally, the authors have performed an impressive task of sequencing cells from many marmoset brain areas, including cortical, subcortical, midbrain and brain stem regions. However, despite this wealth of information at their disposal, relatively little effort has been placed in analyzing the cellular heterogeneity outside of relatively superficial GM/WM differences. Why are these data (subcortical/thalamic/brainstem, etc.) included in the manuscript if no analysis or results is included? The authors appear to devote more effort in showcasing computational methodology and inter-species comparisons rather than answering any specific research question in any significant depth by fully exploring the heterogeneity of their existing dataset.

Major comments

Fig. 1/ Methods: The sampling of the tissue for nuclei extraction is of concern. The 2mm diameter biopsy of cortical tissue in particular. Considering the vast heterogeneity of neurons and glia with various laminar specific niches in the neocortex (e.g. Allen human database isolates layers), how do the authors ensure that the sampled tissue represents the area? E.g. neurons and glia have been shown to have distinct laminar gene expression identity, and different subpopulations of these cells occupy separate neocortical niches.

2 x 5.5 yo marmosets were used in the current study, which is considered middle-aged and may likely have some aging pathology (Tardif et al., 2013). Why were not young adults (2-3 yo) chosen to reduce the risk of including inflammatory artefacts, mainly when cells associated with such processes (e.g. microglia) constitute a significant feature of the study?

How do the authors control for their sampling of cortical tissue that comprises particular cortical layers? Was this consistent in both animals and throughout all neocortex sampled? How does this accurately represent the laminar heterogeneity of brain cells if the entire laminar structure is not sampled?

Fig. S10. ADAMTS17 overview panel appears to be corrupted. Please justify the use of the P0 ISH data to qualify results obtained from 5.5YO adults? Do the authors have evidence of sustained expression of these marker genes throughout life without changes in lamina expression?

Are the differences in microglia and OPC proportions between GM and WM correlated to in vivo cellular representation or a function of the tissue-dependent nuclei purification/capture procedures?

Fig.S13. It appears that a large proportion of microglia from the midbrain and Pons cluster highly with MIC3 microglia that are white matter enriched. However, SC appears more homogenous, despite being a midbrain area. Additionally, diencephalon structures appear to be more homogeneously clustered. The authors provided some insight into the cortical differences between WM/GM microglia but provided no further analysis of subcortical areas. Why are these data presented without providing any insight into their comparative heterogeneity?

In (B), did the authors perform correlation using generalized/total microglia, oligo, neurons or was this only performed using cortical microglia? This section reads as if all nuclei within the clustered were used without separating neocortical fraction. If so, this is not scientifically acceptable in the comparisons being made unless the author can provide adequate justifications.

Fig.2,S13. Can the authors explain the scientific rationale for correlating GM/WM microglial fractions with mouse maturation modules? What is the biological relevance for doing so? Additionally, what are the threshold and criteria used to define the expression of genes identified in mouse microglia to the marmoset? There has been previous evidence that microglia possess area-specific gene expression identity. Considering the wealth of information the authors have at their disposal, why has no effort been put into identifying area-specific heterogeneity within the cortex compared to subcortical regions. This appears to be a massive missed opportunity.

Fig.2/S13. The identification of FLT1 as a microglial identification marker is interesting. Why have the authors not taken the next step to confirm this specificity in the tissue?

Line 191-194: "These findings suggest that the transcriptomic profile of WM microglia is more mature than that of GM microglia. In homeostasis, GM microglia are predicted to be more involved in modulating neuronal synaptic activity, whereas WM microglia are primed to a more active, migratory state."

This result could have significant implications to our understanding of the microglial function that should have been followed up with additional validation. Additionally, the significance of this difference in GM/WM expression of maturation modules (rodent) and its correlation to the actual level of maturation in marmosets and function should be explored and discussed in further detail. Do the expression of markers associated with immature microglia mean the marmoset WM microglia are, in fact, more immature? How does a population such as this maintain a state of sustained immaturity? How and why is the 'level of maturation' influencing microglial function in a region-specific manner?

Fig.3/S17: Same issues as Fig.S13 comment above.

Line 203-205: We, therefore, hypothesize that the divergent environment influenced the molecular profile of its resident cells; WM-OPC acquired additional features specific to their microenvironment. This hypothesis is incongruent with the conclusion the authors drew:

Line 211-212 "Together, these observations support our hypothesis that adult WM-OPC, in homeostasis, is a population tuned to a more active, migratory state than their GM counterparts." The hypothesis laid out by the authors indicates that extrinsic/ microenvironmental states influence the molecular profile of its resident cells. However, the analysis performed to test this hypothesis did not account for any of the actual environmental factors that may be present outside of the fact that the OPCs were dissected from GM or WM. Instead, the conclusion drawn was purely based on GO terms without further validation of the genes associated, regulatory pathways, the actual microenvironmental constituents nor the influence of other resident cells that contribute to it. In short, there is insufficient evidence that the conclusion supports the authors' hypothesis. Please expand or clarify.

Fig.S23. Can the authors comment on the inconsistent Pearson's correlation between marmoset nuclei to mouse and human datasets? It appears that there is little consistency between the 3 mice and 4 human datasets in correlation with marmoset. Why have the authors not taken the opportunity

to perform the local and global interspecies comparison using the data curated from other species?

Line 221-225 “Compared to oligodendrocytes, we found a less cross-species and cross-dataset agreement for OPC and also observed consistently larger differences between marmoset WM OPC and OPC from all other species analyzed. We quantified this observation by comparing the fold-change of similarity between OPC subclusters, measured as the ratio of r^2 values across clusters (Fig. 3C).”

Please cite evidence for the cross-species comparison of oligodendrocytes?

Line 265-272 “We, therefore, pursued a 3D UMAP analysis of oligodendrocytes, finding a spiral pattern that was also observed upon reanalysis of previously reported human (Jäkel et al., 2019) (Fig. S27) and mouse (Zeisel et al., 2018) (Fig. S28) oligodendrocyte transcriptomes. This spiral pattern was also captured at the level of differentially expressed genes (DEG) across oligodendrocyte subclusters. Most oligodendrocyte DEG (XYLT1, TNS1, TNS3, MAN1C1, BTBD16, CCP110, CSF1, DOCK5, PAM, MUSK, GPM6A, DPP10) were aligned across species and were therefore used to label the gross developmental trajectory of oligodendrocytes (Fig. 4B, Fig. S27–29).”

Can the author provide evidence that the appearance and shape of a given cluster generated through any of the dimension reduction methodologies, UMAP in particular here, can bear any biological significance?

Can the authors provide some perspective into why 5.5 yo (ostensibly middle-aged) monkeys would have a substantial mixed population of oligodendrocytes at varying stages of maturation resident in their CNS? Please clarify if this state extends to all oligos or is restricted to a particular region (e.g. neocortex/thalamus etc.).

Line 276-277. “MUSK expression in oligodendrocytes may be unique to primate, as it can also be detected in human oligodendrocytes (Fig. S27) but not mouse.”

This assertion is tenuous at best. The expression is barely visible in the human feature plot (Fig.S27C) and has not been quantitatively assessed, and the expression threshold to indicate expression is unclear.

Line 317-319 “we found that molecular distances from OLI4 to OLI5 and from OLI4 to OLI6 were similar, indicating that OLI5 and OLI6 might develop in parallel rather than dependently (Fig. 4C–D, top).”

This result is fascinating. Do the authors have evidence of a pseudotime trajectory analysis that could indicate a bifurcation that might provide additional validation of this?

Line 328-330 “That ETV5 expression peaks in OLI3 (Step 60–80) suggests that OLI3 might be a population that is poised to further differentiate upon appropriate signaling (Fig. 4D).”

Is there prior literary evidence that can support this assertion as it appears pretty speculative?

Assuming the pseudotime conclusion is correct and that the OLI subclusters are organized along with maturation states, could ETV5 in the middle of the continuum merely indicate suppression of differentiation from immature states towards functional maturation and functional specialization?

Line 332-335 “We observed shared transcriptomic features and intermingled distribution of nuclei from astrocyte (AST) and vascular (VAS) classes in Level 1 analysis, so we pooled these two classes for the second round of quality control to facilitate artifact imputation.”

The rationale for the pooling of vastly different cell types for the analysis was not clear to me. Can the Authors explain the scientific rationale for this? For this analysis, it does not seem appropriate to pool astrocytes, one of the fundamental neural cell types, with non-neural cell types. Further, classifying astrocytes as ‘cells at the barriers of the CNS’ (Fig. S30) is highly reductive and not representative of the various (and more significant) homeostatic functions performed by astrocytes in the parenchyma. I acknowledge that astrocytes interact with, as well as regulate and maintain, the neurovasculature and the components of the BBB. However, interaction with the BBB and being a constituent of the BBB is a significant distinction. There is evidence that the BBB is formed before astrocytic colonization of the brain (DOI: 10.1038/nature09513), indicating the astrocytes are participants in its regulation but not a

biophysical constituent of the BBB. Please clarify/justify and provide evidence that astrocytes can be classified in this way.

Fig. S31. Please provide evidence and citations of the various categories/ levels of permeability of the CNS. More importantly, the authors lumped astrocytes in as 'cells at the barriers of the CNS' (currently without sufficient evidence and rationale); why then are astrocytes not included in this classification. By omission, this indicates that astrocytes are not, in fact, typically considered 'cells at the barriers of the CNS' and is, therefore, a significant flaw in the study design. Please clarify/justify and provide evidence that astrocytes can be classified in this way.

Fig. S31. ALDH1L1, SLC1A2, AQP4, GFAP are ASTROCYTE specific/rich markers. This figure is inappropriate and is highly misleading as these markers are not typically considered "genes enriched in endothelia, meningeal and ependymal cells". Additionally, it is inappropriate and misleading to compare ISH data at P0 to the adult snRNAseq dataset as it does not consider age-dependent changes in gene expression and is not directly representative of the dataset.

Fig. S30-36. What is the point of grouping astrocytes with 'cells at the barriers of the CNS' (Fig. S30) when there was no subsequent analysis to analyze or dissect their relationship with each other? The analysis split them up and analyzed them individually in Fig. S32-34 (Astrocyte) and 35-36 (VAS).

Fig. S32. Again, with all this information available regarding cortical and subcortical regions, why was these data not explored in more depth to look at the regional diversity in astrocyte identity and function beyond markers? As I understand it, The human astrocyte comprises multiple morphologies that may be identified based on gene expression identity. Why was no effort placed in aligning these marmoset data to that of humans, similar to that performed in (<https://doi.org/10.1101/2020.03.31.016972>)?

Line 354-355 "This result leads to the prediction that the brain's response to injury may not be uniform across WM areas."

This seems highly speculative and is not based on sufficient empirical or literary evidence. How did the authors come to this conclusion? Is it reasonable to suggest this merely based on differences in the expression of GFAP/AQP4? Can the authors provide evidence to support this?

Fig. S32F. Please clarify why the rationale behind grouping astrocyte clusters using the telencephalon/non-telencephalon designations. I am not convinced that neocortical entorhinal and thalamic structures should be grouped considering the clustering differences demonstrated in Fig. S32A. Additionally, rodent studies have ascribed vast region (<https://doi.org/10.1038/s41467-019-14198-8>, <https://doi.org/10.3389/fnins.2020.00061>) and laminar (<https://doi.org/10.1038/s41593-020-0602-1>) specific gene expression identity to astrocytes. In this context, what was the rationale for grouping astrocytes in this manner?

Line 385-400. Can the authors clarify the rationale behind the use of this subset of patterning genes in Fig. 5C?

Line 391-393. Is there evidence of these patterning genes in the caudate and HC that could support the authors' statement that they have "...mostly lost the expression of forebrain patterning genes."?

Line 398-399. "forebrain, midbrain, and hindbrain specification was preserved more prominently in AST than any other cell classes."

I would argue that the neuronal subclass has relative patterning gene expression as astrocytes. Can the authors provide clarification or a more explicit graphical representation of the data that supports this statement?

Fig. S34 - S36 is not cited in the main text. What is the relevance of these data?

Line 404-406 "To explore this possibility, we extracted and compared differentially expressed TF

between analogous GM- and WM-specific glia pairs (MIC1/MIC3, OPC1/OPC3, AST1/AST3).” Can the authors clarify if this comparison was performed using only neocortical glia within the listed clusters or total glia within the clusters, comprising subcortical, entorhinal and brainstem glia? Can the authors clarify the scientific rationale for focusing on the shared TF between 3 different glial types with different origins, functions, and identities? Additionally, in this context, why were Oligodendrocytes separate from this comparison? The authors need to acknowledge that Glia is not a homogenous population that can/should necessarily be subjected to comprehensive wholesale analysis. As demonstrated in the authors’ dataset. Glia is diverse and heterogeneous, and reducing all of them to this level, considering their vastly different gene expression profile, niche, origin, and function, would require much more scientific justification and validation than just because they are ‘glia’.

Line 410-411 “consistent with the observation that GM glia are low in RNA complexity compared to their WM counterparts (Fig. 6A, Fig. S37A).”

What is the authors’ definition of “RNA complexity”. I ask because the only quantitative measure of this presented in the manuscript is the number of genes expressed. Is this a sufficient measure of “RNA complexity”, or is it simply a matter of fewer genes expressed. How does this correlate to the ‘similar functional modules’ found in white matter MIC and OPC/OLIGO regarding a more ‘active’ state primed for migration?

Fig. 6B I am struggling to find the relevance in GO/ gene module analysis between GM glia. In particular, what is the relevance and significance of making this comparison in a mixed population of glia, why is this a scientifically justifiable method of analysis, and the significance of this finding? E.g. Why do the “regulatory programs in GM microglia showed the highest similarity with those in GM-OPC”? What are the significance and the justification of the mixed-cell type comparison, etc.? This section started without adequate scientific justification and ended abruptly with no conclusion or statement regarding the significance of the findings. Perhaps the authors could expand this section to clarify the analysis and the findings to make it more easily understandable. Please address.

Line 416-442 This is an exciting section of results, and the authors have done an excellent job in categorizing and illustrating the potential contribution of cell types to various neurological disorders. This data section stood out as an exciting multicellular interrogation of probabilistic contribution to neurological disorders and neurodegenerative diseases. It would be exciting to see this section expanded with more in-depth analysis and in vivo validation of these findings.

Does the number of genes expressed in a particular cell type cluster influence/affect the EWCE analysis? E.g. in Fig. 6, the authors showed that MIC3 exhibited a higher number of genes detected than MIC1. Can this increase in genes expressed be correlated to increased contribution by this cluster to neurological disorders?

This result section is titled “GM and WM glia differentially contribute to neurological disorders”. However, only 1 example of GM vs. WM differences were reported (microglia). How can the authors justify using such a broad term in this context when the example provided was only limited to microglia? The authors have so much information at their disposal yet was not prepared to adequately synthesize these data into a more coherent and conclusive demonstration of the point they were trying to make. Please expand.

Grouping the charts in Fig.6C and Fig.S38 along GM/ WM lines would significantly assist with data interpretation for the reader. At this stage, a reader has to expend a significant amount of effort to identify the origin of any particular cell cluster.

Can the authors clarify if the cell clusters contained neocortex only GM and WM in results or the legend, or is a mixed population of neocortical and subcortical cells grouped as GM and WM? This is significant because the classifications of neurological disorders range from pathologies that are generally localized to the neocortex/ thalamus/ brainstem etc. Please clarify and justify if a mixed population is used.

Fig. 6C. Can the authors comment on the heavy association of microglia but not oligodendrocytes to MS? Was this expected based on previous literature? What is the functional/pathological significance

of this, and can the authors cross-reference this finding to human MS datasets?

Line 422 (Fig. S37B) cited should be (Fig. S37C)

Line 443 – again, why were OLIGOS not added to this analysis? Please clarify the origin of ‘cerebral white matter. Cortical only or includes subcortical white matter? If it includes subcortical white matter, what is the scientific justification? Do the authors have evidence that cortical and subcortical white matter cells are homogenous and can be grouped in this manner?

Fig.6D, Fig. S39. Can the authors explain why no OPC-OPC or AST-AST crosstalk was reported in the GM?

Fig. S39, Fig. S40. I do not understand the distinction between these 2 LT pairs analyses. Can the authors expand and elaborate to clarify?

The result section ends abruptly with no conclusion or perspective about the authors’ dataset.

Line 550-551 “In summary, although environmental cues contribute to diversifying astrocytes, the heterogeneity in their developmental origin plays a larger role in subtype specialization.”

I am not convinced that the authors have performed sufficient or accurate AST cluster heterogeneity to make this statement. Please justify.

Line 557-558 “We examined the OPC expression of ion channel genes as a surrogate of electrophysiological function and examined the tissue origin of differentiating OPC (OPC5).”

How can the authors justifiably consider the expression of these ion channels a “surrogate of electrophysiological function”? The authors only observe and report the transcription of these ion channels without any form of evidence translation or in vivo/ ex vivo histochemical proof of the channels’ subcellular localization.

Line 563-570 Interestingly, the final discussion section suggests that this study aimed to develop and validate computational tools or develop a protocol pipeline to analyze heterogeneity. If this is the case, then it really should be made much clearer at the start of the main text because this final paragraph does confuse the context of this study. If this is true, why was there so little in the analysis pipeline included in the discussion? The probabilistic multicellular neurological disorders association data, which is a significant highlight of this paper for me, was only relegated to one line in the discussion. The discussion ultimately falls flat without highlighting the enormous potential of the work to drive additional avenues of research forward and its implications for interspecific comparison of CNS cellular heterogeneity.

Reviewer #3 (Remarks to the Author):

This interesting study by Lin et al. focuses on understanding how microenvironmental influences in different regions of the CNS impact developmental and functional heterogeneity of glia. The authors utilize the marmoset to investigate this question in white matter and gray matter regions of the marmoset brain. The marmoset CNS displays unique properties when compared to mouse, in particular the marmoset white matter (WM) is closer in size to higher primates and humans. Furthermore, the marmoset is an animal model that bridges mouse and higher primates not only genetically, but also immunologically and behaviorally. Therefore, results obtained in the marmoset will more closely resemble cellular and molecular heterogeneity found in human brain.

The authors used snRNAseq profiling to define cells (total of 500,000 cells analyzed) from 19 tissue types from the normal marmoset CNS, and spatially mapped 87 distinct subclusters onto a 3D MRI atlas. They also performed a broad range of molecular analyses - including cross-species comparisons, developmental and regulatory pathways, screening of cellular determinants of neurological disorders, and developed models of intercellular regional communication. The authors conclude that the most significant finding resulting from this analysis is the marked effect of gray

matter (GM) vs. WM on a broad range of glial cell types, in particular microglia, astrocytes and OPCs – indicating not only persistent developmental influences, but also highlighting functional heterogeneity in these cell types. Molecular complexity of glia is higher in WM than in GM, where bioinformatic analysis predicts more communication among adjacent cells.

Overall, this is a well performed study, which provides a wealth of data on glial cells of the WM – a brain region that has been significantly understudied in different animal models. Furthermore, this is a significant effort in defining important regional differences in these glial cell populations in WM vs. GM. The outcomes of this study will also provide a solid platform for analysis in animal models of injury and disease. However, the paper is really a molecular atlas that could be useful to investigators and many of the results are significantly overinterpreted, or explained without considering potential alternative interpretations.

Remarks to the authors:

- The paper is a great resource, and the experiments and analysis were well performed but conclusions based on molecular data are not validated by cell specific protein expression patterns that would directly support the conclusions.
- The paper highlights the interaction between neurons and glia. It would be interesting to examine the opposite communication as well. The authors assume that the OPCs are influenced differently in WM vs GM microenvironment, but they do not provide strong experimental evidence in favor of this conclusion.
- The authors show that GM microglia are predicted to be more involved in modulating neuronal synaptic activity and the WM more primed to active migratory state. It would be interesting to see if this is true in specific regions (for example for in WM corpus calosum vs. internal capsule vs. subcortical WM; and for GM cortex vs. basal ganglia vs cerebellum).
- The authors show that MUSK is uniquely expressed in primates, but not in mouse brain oligodendrocytes. This is a very interesting finding particularly in homeostatic conditions of the mouse basal forebrain. Can this be validated at the protein level? Is MUSK differentially expressed in WM vs. GM, and is it developmentally regulated?
- The authors show that the gene transcription in OL lies on a spiral trajectory, and modeled whether this can be affected by the environment. In the pseudotime analysis of the marmoset, why did the authors set the starting point for the analysis as the ENPP6^{high} ?
- The authors report that GM and WM glia differentially contribute to neurological disorders and tumors. However, the authors used IPA, which often overrepresents genes associated with solid tumors.
- The analysis of autism-related genes is superficial and biased. The authors are basing their analysis of genes associated with autism on a study by Polioudakis et al., 2019, in which human fetal brains were analyzed (midgestation) and not adult brains. Importantly, Polioudakis et al., referenced previous studies for gene markers of autistic disorders and schizophrenia (adolescent and adult brains) to create an atlas in fetuses. This is an important point that the authors should take into consideration when they analyze their data in adult marmoset brain. This may be resolved either by selecting a different (adult) database, or by applying this dataset to the developing marmoset brain.
- The authors show that WM glia interact with other resident cells more than GM glia. This is an important finding by which GM glia is characterized as naïve protoplasmic, and not very active compare to the WM glia. It would be very useful if the authors could provide some morphological and immunohistochemical assessment of the WM vs. GM.

REVIEWER COMMENTS

Reviewer #1 (Remarks to the Author):

This manuscript from Reich lab offers an extensive inspection of cellular composition and location-specific specialization of white matter compared with gray matter profiling many nuclei in total from 19 tissue types from marmoset ~~central~~ nervous system. The authors were able to successfully map the effect of gray vs. white matter on primary glial cell types and functional heterogeneity. It will be interesting to study the role of this heterogeneity in neurological disorders and this work marks great information for that. The effort from the authors to develop an atlas CjPCA seems to be valuable as a resource to the field in general. This work is an important step forward to understanding the cellular composition and heterogeneity in white and gray matter and will be a rich resource for others to build on.

The details and statistics of the samples and sequencing data are well described in the main text as well as supplementary data. The overall presentation is satisfactory, and the methodology is elaborated in detail which is appreciated. This is a well written article - the design, methodology and conclusion seem to be reasonable, reliable, and valid, though with a few minor concerns that are mentioned below.

We thank reviewer #1 for recognizing the value and all our efforts in making this resource transparent, reproducible, and robust. Further, we address the concerns as below in blue text:

R1-1. Although the abstract is well written, given how extensively the authors have investigated the data, a schematic summarizing the analysis workflow in the main body of the manuscript will be helpful for readers to keep the track of the multiple analyses performed. Although the list of resources provided in the manuscript is useful, I will suggest providing one main diagrammatic workflow and another just for the data analyses part mentioning the steps and key resources.

We thank the reviewer for the great suggestion; we now added Fig.2, containing a schematic summary of analysis workflow with figure index to help readers keep track of which analysis was done on which cell type. Another workflow diagram on the data analysis part can be found in the supplementary Fig. 2.

R1-2. How do authors address the effect of technical variation, that is sort of inevitable for such an extensive study, on biological interpretations especially when marmoset has less information available compared to mouse and humans?

We thank the reviewer for bringing up this point. We have taken pains to control for technical variation and its impact on biological interpretation and summarize the following aspects:

Sample preparation –

We choose snRNA-seq over scRNA-seq because: (a) it has been widely applied in human tissue, which improves the compatibility of our dataset with clinical samples; (b) it is the only proven method to analyze tissue that cannot be readily dissociated into single-cell suspensions without introducing additional artifacts; (c) it can be applied to RNAlater-protected tissue, allowing accurate region sampling with the guidance of MRI.

Analysis approach –

To control over-splitting in defining any given subtype, we looked for a variety of pieces of evidence to support claims that it has a different biological meaning from other closely associated subtypes, and as a result we have been relatively conservative in subclustering. As gene expression often falls along a spectrum, we considered the transcriptomic landscape in its entirety instead of using one or a few genes to define each subpopulation. To take advantage of prior knowledge in gene annotation, we invested heavily in cross-species comparisons. We reasoned that genes with one-to-one orthologs preserve the most essential function, which will be more robust because they consider the entire shared transcriptomic profile rather than a few marker genes.

Data interpretation –

Strictly speaking, there is always a gap in concluding cross-species and cross-dataset comparisons, and the interpretation is only as good as the sample annotation. Our protocol allows the transcriptomes of nuclei to be mapped onto a small and confined region with detailed 3D MRI annotation, sufficient to gain biological insights within the current dataset.

R1-3. The Harmony integration is well implemented and elaborated extensively but a UMAP image split by batches (without batch correction) should be provided in the manuscript. This will be helpful for readers to notice the occurring batch effect(s) if any and compare it with harmony correction.

We thank reviewer for the suggestion; we added UMAP image colored by animal and tissue type before and after Harmony integration in Supplementary Fig. 3a.

R1-4. Since integration is the one of the most crucial steps to interpret the data but it comes with no universal set of parameters to identify subtypes ideally due to over or under splitting of specific cells, it is advised that authors should incorporate another integration method like RPCA and compare it with Harmony specifically at the subcluster level.

We agree with reviewer that data integration is very critical. We tried to apply RPCA integration to our dataset; however, there is an ongoing issue of this application to integrate samples over a certain number, in our case 42. It is a bug that the developer does not understand and is currently investigating; please see more relevant discussion over the issue report: <https://github.com/satijalab/seurat/issues/2902>. However, as shown in Supplementary Fig. 3a, PCA plot without data integration, it seems there is only a mild batch effect presented in our dataset across animals and tissue types. The annotation for level 1 analysis is robust, which is the foundation of the cell class partition at the subcluster level.

R1-5. The authors did not mention the nuclear RNA integrity check. It would be worth mentioning the quality measures for the RNA samples.

We thank the reviewer for the reminder; we now added that in the Method section: Lines 556–559

“The quality of RNAlater-preserved tissue was assessed by measuring RNA Integrity Number (RIN) on the Agilent 2100 Bioanalyzer (G2939BA, Agilent). Bulk RNA was isolated with TRIzol™ Reagent (15596026, Invitrogen) and measured with Agilent RNA 6000 Pico Kit (5067-1513, Agilent); samples with RIN > 8.5 were used in the study.”

R1-6. I will highly recommend providing a table containing cluster assignment for each nucleus. This will be helpful to compare this dataset with other studies.

We thank the reviewer for emphasizing data transparency. The processed data with nucleus annotation are submitted to Gene Expression Omnibus (GEO) under session GSE165578. Data can also be visualized at <https://cjpca.ninds.nih.gov>.

R1-7. There are some typos in the manuscript. For example, Figure S5 is mislabeled as S6 in the supplementary figure. In line #200, it should be Fig S17B.

We thank reviewer for pointing this out, and we have updated and double-checked the manuscript accordingly.

Reviewer #2 (Remarks to the Author):

The authors are to be commended in their approach and thoroughness in the computational methodologies used to analyze and validate their data. However, the depth to which these methods were applied did not correlate to an increase in the scientific/biological significance of the finding. This study can be groundbreaking in understanding regional heterogeneity of neuronal and glial cells in multiple regions of the primate brain but falls short in delivering the results to a sufficient depth, complexity or significance.

This paper presents an interesting paradox whereby a comprehensive

methodology in the computational analysis of data was applied, yet relatively superficial insights into the relationship between cell-type heterogeneity and function were described. The characterization of each glial cell type provides a fascinating GM/ WM diversity in their gene expression and potential function but stops short of addressing the impact or significance of the finding. Instead, this paper reads more as a comprehensive pipeline for the computational/ bioinformatic validation of cellular heterogeneity rather than a hypothesis-driven exploration of the relationship between heterogeneity and function. Every result section appears to be stand-alone with very little connecting them except for the pervasive GM vs. WM theme, which is a particular concern.

In its current form, this manuscript might be better presented as a resource for marmoset data or transcriptomic analysis pipeline. Otherwise, in light of the sheer amount of data presented, contrasting with the superficial analysis of the data (not in methodology but content and significance), the authors should seriously consider splitting this manuscript up into separate, more digestible components with more in-depth analysis of each component.

Additionally, the authors have performed an impressive task of sequencing cells from many marmoset brain areas, including cortical, subcortical, midbrain and brain stem regions. However, despite this wealth of information at their disposal, relatively little effort has been placed in analyzing the cellular heterogeneity outside of relatively superficial GM/WM differences. Why are these data (subcortical/thalamic/brainstem, etc.) included in the manuscript if no analysis or results is included? The authors appear to devote more effort in showcasing computational methodology and inter-species comparisons rather than answering any specific research question in any significant depth by fully exploring the heterogeneity of their existing dataset.

We thank the reviewer for recognizing the potential value of our study as a resource for the field. Indeed, as pointed out by the reviewer, our major goal was in fact to understand glial diversity of the WM, which is a significantly understudied brain region that is highly relevant for many diseases, including diseases for which marmosets serve as important experimental models (e.g., multiple sclerosis). However, to put the findings in WM in a relevant context, we also included many cortical and other brain regions of adult marmoset, as such a dataset and information did not exist when we started the project, and the findings remain novel today. We hope that the revision is more clearly aligned with our overarching goals, and that the resource we have provided will prove useful for many studies going forward.

We have now significantly revised our manuscript to contextualize our work better along the following lines:

1. Nuclei from all 19 tissue types (including subcortical gray matter, thalamus, and brainstem) are included throughout the paper except Fig. 8 and Supplementary Fig. 41–42, in which nuclei from caudate, thalamus, LGN, hippocampus, midbrain, pons, cerebellum, and spinal cord are **not** included.
2. Lines 70–73: “As indexed in Fig. 1d, we used “**WM**,” “**GM**,” and “**other**” (in quote marks) to indicate sampling sites as specifically defined in our paper, whereas WM/GM (without quote marks) is used for general descriptive purposes, including when mentioning published works.”
3. Lines 138–141: “We further used GM-glia and WM-glia (i.e., WM-microglia, WM-OPC, WM-astrocytes, written here without quote marks) to indicate regionally enriched glial subtypes, as opposed to glia sampled

from "GM" or "WM," which include all glia collected from the indicated area regardless of subtype."

4. We added Fig. 2, which contains a schematic summary of the analysis workflow with figure index to help readers keep track of which analysis was done on which cell type and by what method.

Major comments

R2-1. Fig. 1/ Methods: The sampling of the tissue for nuclei extraction is of concern. The 2mm diameter biopsy of cortical tissue in particular. Considering the vast heterogeneity of neurons and glia with various laminar specific niches in the neocortex (e.g. Allen human database isolates layers), how do the authors ensure that the sampled tissue represents the area? E.g. neurons and glia have been shown to have distinct laminar gene expression identity, and different subpopulations of these cells occupy separate neocortical niches.

We have responded to this concern as follows, on lines 109–116:

"With respect to neurons, it was not our primary focus to define new subtypes or quantify region and layer specificity, but we performed some basic analyses to anchor the resolution of our atlas with published datasets collected primarily from cortical regions¹⁷. In the current atlas, we profiled 5 different cortical areas and employed MRI-guided tissue collection to ensure consistency across animals. We note that a 2 mm-diameter tissue punch is sufficient to cover nearly the full thickness of marmoset cortex. Furthermore, the purity of cortical sampling can be estimated by the number of oligodendrocytes presented in "GM" (~8.2% median abundance; Fig. 1f)."

R2-2. 2 x 5.5 yo marmosets were used in the current study, which is considered middle-aged and may likely have some aging pathology (Tardif et al., 2013). Why were not young adults (2-3 yo) chosen to reduce the risk of including inflammatory artefacts, mainly when cells associated with such processes (e.g. microglia) constitute a significant feature of the study?

4–6-year-old marmosets are in a typical range for use in disease models; as such, the results reported here are relevant for the interpretation of data obtained from such models. Furthermore, we performed in vivo MRI on the two animals included in the current study, and no signs of brain atrophy or white matter signal abnormality were noted. Histological studies on prior archived brain sections from healthy animals of a similar age show few to no signs of reactive microglia.

R2-3. How do the authors control for their sampling of cortical tissue that comprises particular cortical layers? Was this consistent in both animals and throughout all neocortex sampled? How does this accurately represent the laminar heterogeneity of brain cells if the entire laminar structure is not sampled?

Please see the response to R2-1. Though we appreciate its importance, laminar resolution at the single nucleus transcriptomic level was not our aim in this study

R2-4. Fig. S10. ADAMTS17 overview panel appears to be corrupted. Please justify the use of the P0 ISH data to qualify results obtained from 5.5YO adults? Do the authors have evidence of sustained expression of these marker genes throughout life without changes in lamina expression?

We thank the reviewer for spotting the corrupted panel, and we have updated the image. We agree with the reviewer that it is important to consider

developmental stages to reach more relevant conclusions. We removed the following sentence from the original submission: "In addition, *ADAMTS17* was exclusively detected in NEU37 and uniquely labeled L4 of primary visual cortex (Fig. S10B)." We have further revised the manuscript as follows:

Lines 131–134:

"Given that the establishment of lamination is completed prenatally¹⁹, we cross-referenced our findings in the adult with an available in situ hybridization (ISH) database (Marmoset Gene Atlas) from P0 marmoset²⁰. We found that the expression of lamina-enriched genes agreed with what has been examined spatially in the database (Supplementary Fig. 11)"

R2-5. Are the differences in microglia and OPC proportions between GM and WM correlated to in vivo cellular representation or a function of the tissue-dependent nuclei purification/capture procedures?

We thank the reviewer for bringing up this concern, in response to which we quantified and compared the density of IBA1⁺ cells in GM and WM.

Lines 166–170:

"Next, we performed particle and morphological analysis on IBA1 labeling to compare the density and the shape of microglia in GM and adjacent WM. We found 2–3 times more IBA⁺ cells present in WM compared to GM, which agrees with the relative abundance of microglia profiled from "GM" and "WM" with snRNA-seq (Fig. 3c)."

R2-6. Fig.S13. It appears that a large proportion of microglia from the midbrain and Pons cluster highly with MIC3 microglia that are white matter enriched. However, SC appears more homogenous, despite being a midbrain area. Additionally, diencephalon structures appear to be more homogeneously clustered. The authors provided some insight into the cortical differences between WM/GM microglia but provided no further analysis of subcortical areas. Why are these data presented without providing any insight into their comparative heterogeneity?

In (B), did the authors perform correlation using generalized/total microglia, oligo, neurons or was this only performed using cortical microglia? This section reads as if all nuclei within the clustered were used without separating neocortical fraction. If so, this is not scientifically acceptable in the comparisons being made unless the author can provide adequate justifications.

We apologize for the confusion and have now clarified our annotations. In particular, we use WM-microglia and MIC3, and GM-microglia and MIC1, interchangeably. Nuclei from all 19 tissue types are included in this part of the analysis.

Lines 156–164:

"We identified regionally enriched subtypes across 19 tissue types. We denoted 2 subtypes (MIC1 and MIC2) as GM-microglia, for they were found to be most abundant in "GM." We then named the other major cluster (MIC3) WM-microglia for its absence in "GM" and enrichment in "WM." All 3 subtypes of microglia present with various proportions in "other," which had cellular composition intermediate between relative pure WM and GM (Fig. 3c, Supplementary Fig. 14a–b). This GM-WM segregation of microglia was so strong that the abundance of WM-microglia (MIC3) was positively and negatively correlated with the number of oligodendrocytes and neurons, respectively. In contrast, GM-microglia (MIC1 and MIC2) had similar densities across brain regions (Supplementary Fig. 14c)."

R2-7. Fig.2,S13. Can the authors explain the scientific rationale for correlating GM/WM microglial fractions with mouse maturation modules? What is the biological relevance for doing so? Additionally, what are the threshold and criteria used to define the expression of genes identified in mouse microglia to the marmoset? There has been previous evidence that microglia possess area-specific gene expression identity. Considering the wealth of information the authors have at their disposal, why has no effort been put into identifying area-specific heterogeneity within the cortex compared to subcortical regions. This appears to be a massive missed opportunity.

Our rationale for making this comparison is now explained on lines 175–183: “It has been shown that normal aging impacts GM and WM asynchronously²²⁻²⁴. We therefore sought to compare these regionally enriched modules in microglia against a dataset with temporal resolution. We linked marmoset gene names to their mouse orthologs, then cross-referenced the expression pattern of the defined modules in microglia extracted from whole mouse brain (ages E14.5 to P540)²⁵. After splitting mouse microglia into 3 age groups (embryo, neonate, adult; Fig. 3i and Supplementary Fig. 16), we found that gene modules enriched in marmoset GM-microglia were highly expressed in microglia of young mice, whereas gene modules enriched in marmoset WM-microglia were also highly expressed in microglia of adult mice (Fig. 3j).”

Please note, as discussed above, that the comparison we are making is with microglia subtypes that are enriched in either GM or WM, not the microglial fractions in these tissue types.

With respect to within-cortex heterogeneity, we agree with the reviewer that this is an important area for further study, and results could easily be integrated into our atlas; however, it was not the aim of our current work.

R2-8. Fig.2/S13. The identification of FLT1 as a microglial identification marker is interesting. Why have the authors not taken the next step to confirm this specificity in the tissue?

We thank reviewer for finding this discovery interesting. We now cross-reference the enrichment of FLT1 expression in microglia relative to peripheral immune cells in a human transcriptome dataset (Absinta et al., 2021). In that dataset, *PTPRC* is a general immune cell marker; *TOX*, a T cell marker; *EBF1* and *POU2AF1*, B cell markers; *P2RY12* and *TREM2*, along with *FLT1*, microglia markers. However, we agree with the reviewer that further validation at the protein level — particularly the absence of Flt1 protein in peripheral cells — would be required to fully validate this conclusion. This work is currently in process.

R2-9. Line 191-194: "These findings suggest that the transcriptomic profile of WM microglia is more mature than that of GM microglia. In homeostasis, GM microglia are predicted to be more involved in modulating neuronal synaptic activity, whereas WM microglia are primed to a more active, migratory state." This result could have significant implications to our understanding of the microglial function that should have been followed up with additional validation. Additionally, the significance of this difference in GM/WM expression of maturation modules (rodent) and its correlation to the actual level of maturation in marmosets and function should be explored and discussed in further detail. Do the expression of markers associated with immature microglia mean the marmoset WM microglia are, in fact, more immature? How does a population such as this maintain a state of sustained immaturity? How and why is the 'level of maturation' influencing microglial function in a region-specific manner?

We now elaborate on these results in lines 181–188:

"...we found that gene modules enriched in marmoset GM-microglia were highly expressed in microglia of young mice, whereas gene modules enriched in marmoset WM-microglia were also highly expressed in microglia of adult mice (Fig. 3j). These findings suggest that the transcriptomic profile of WM-microglia appears further aged than that of GM-microglia. GM-WM segregation of the microglial transcriptome is observed as early as P7 (during myelinogenesis) in mouse⁷ and persists with normal aging in both human and mouse^{6,26,27}. Understanding whether environmental cues in myelin-rich regions drive microglial specialization requires further study.

R2-10. Fig.3/S17: Same issues as Fig.S13 comment above.

As for microglia, we have now clarified our nomenclature and use WM-OPC and OPC3, and GM-OPC and OPC1, interchangeably. This does not imply that white matter contains exclusively WM-OPC and gray matter exclusively GM-OPC. In this figure, we include nuclei from all 19 tissue types and note that all 5 subtypes of OPC (OPC1–5) present in various proportions in the "other" tissue type, which includes caudate, thalamus, LGN, hippocampus, midbrain, pons, cerebellum, and spinal cord.

R2-11. Line 203-205: We, therefore, hypothesize that the divergent environment influenced the molecular profile of its resident cells; WM-OPC acquired additional features specific to their microenvironment.

This hypothesis is incongruent with the conclusion the authors drew:

Line 211-212 "Together, these observations support our hypothesis that adult WM-OPC, in homeostasis, is a population tuned to a more active, migratory state than their GM counterparts."

The hypothesis laid out by the authors indicates that extrinsic/microenvironmental states influence the molecular profile of its resident cells. However, the analysis performed to test this hypothesis did not account for any of the actual environmental factors that may be present outside of the fact that the OPCs were dissected from GM or WM.

Instead, the conclusion drawn was purely based on GO terms without further validation of the genes associated, regulatory pathways, the actual microenvironmental constituents nor the influence of other resident cells that contribute to it. In short, there is insufficient evidence that the conclusion supports the authors' hypothesis. Please expand or clarify.

As requested, we now expand on and clarify our thinking in lines 205–234:

"As with microglia, GM-WM segregation is prominent in OPC, which we grouped into 5 subclusters (OPC1–5) from a total of 20,306 nuclei (Fig. 4a,

Supplementary Fig. 18–20). The number of WM-OPC (OPC3) was positively correlated with the abundance of oligodendrocytes and negatively with the abundance of neurons, whereas GM-OPC (OPC1) were similar in density regardless of sampling site (Supplementary Fig. 18c). Interestingly, several top-enriched genes related to general nervous system functioning were shared between GM-OPC (OPC1) and GM-microglia (MIC1), and both populations had fewer detected genes compared to their WM counterparts (Supplementary Fig. 14b, 18b). We therefore hypothesized that the divergent environments might influence the molecular profile of their resident cells, and that WM-OPC acquired additional features in response to their intercellular microenvironment.

To explore this hypothesis, we performed gene module analysis (Supplementary Fig. 18e–19, Supplementary Table 2) and found that WM-OPC were enriched with GO processes related to component organization, molecule modification, and stress granules (Knn.m6; Supplementary Fig. 18d), whereas GM-OPC enriched pathways are involved in neuronal support (PG.m2; Supplementary Fig. 18c) similar to those enriched in GM-microglia. Markers enriched in WM-OPC are known for regulating OPC dispersal (*SLIT2*)²⁸ and inhibiting CNS angiogenesis (*SEMA3E*)²⁹ (Supplementary Fig. 18d, 20). Together, these observations suggest that WM-OPC, in homeostasis, are a population tuned to a more reactive state, whereas GM-OPC are more involved in supporting neuronal functions.

In line with our finding that marmoset WM-microglia appear transcriptionally more advanced in normal aging than their GM counterparts (Fig. 3), it has been reported that rat OPC in WM are more mature than those in GM³⁰, and that they differentiate into mature oligodendrocytes more efficiently than OPC in GM³¹. Electrophysiological properties of OPC vary between WM and GM and with age, and they correlate with differentiation potentiality^{32,33}. We examined the OPC expression of ion channel genes as a surrogate of electrophysiological function and examined the tissue origin of differentiating OPC (OPC5). We found different profiles of ion channels in GM-OPC and WM-OPC (Supplementary Fig. 21) but a similar abundance (<1.5%) in OPC5 across brain regions (Fig. 4a). How these observations translate to actual differences in stimulus responses in health and disease requires further study.”

R2-12. Fig.S23. Can the authors comment on the inconsistent Pearson’s correlation between marmoset nuclei to mouse and human datasets? It appears that there is little consistency between the 3 mice and 4 human datasets in correlation with marmoset. Why have the authors not taken the opportunity to perform the local and global interspecies comparison using the data curated from other species?

OPC are well known to be a versatile population, and it is challenging to find OPC-specific markers. Hence, we clarify our thinking on this issue in lines 235–257:

“To understand how marmoset OPC subclusters compare to those from other species, we re-analyzed data from prior studies^{34–41} and performed a Pearson’s correlation analysis (Fig. 2, Supplementary Fig. 22–24). Consistent with what has been reported for OPC derived from adult human brain, we did not observe a separate cycling OPC cluster (*TOP2A*⁺) in adult marmoset brain, as has been reported for OPC derived from zebrafish, adult mouse, and developing human cortex (Supplementary Fig. 22–23). Instead, cells expressing S and G2/M phase genes were dispersed across the OPC1–3 subclusters. OPC4, however, was enriched with G0G1 genes (Supplementary Fig. 18d), i.e., they are quiescent cells⁴².

Prior to this comparison, we humanized gene names of each species with one-to-one orthologs and only included genes that are detected in all datasets, which

might limit the depth of the comparison. However, we found agreement in oligodendrocyte lineage differentiation features across species (Fig. 4b): marmoset differentiating OPC (OPC5) and oligodendrocytes (OLI) correlate with *ENPP6*⁺/*MAG*⁺ oligodendrocyte lineage cells in mouse (mm1_2, mm2_1, mm3_NFO, mm3_OLI), and human (hs1_4, hs2_2, hs3_3, hs4_OLI), but less so in zebrafish (dr_3). Also, we observed consistently larger differences between marmoset WM-OPC (OPC3) and OPC from all other species analyzed. We quantified this observation by comparing the fold-change of similarity between OPC subclusters, measured as the ratio of r^2 values across clusters (Fig. 4c). Although the underrepresentation of a marmoset WM-OPC-like population in other datasets may partially be due to technical differences, such as sampling site, it is also likely that OPC were broadly undersampled in other datasets. As clustering resolution is sensitive to cell counts in single-cell studies, low recovery number of OPC (particular those derived from humans) and/or lack of inclusion of enough equivalent WM regions (especially in mice, where there is little WM) might contribute to this observation. "

R2-13. Line 221-225 "Compared to oligodendrocytes, we found a less cross-species and cross-dataset agreement for OPC and also observed consistently larger differences between marmoset WM OPC and OPC from all other species analyzed. We quantified this observation by comparing the fold-change of similarity between OPC subclusters, measured as the ratio of r^2 values across clusters (Fig. 3C)." Please cite evidence for the cross-species comparison of oligodendrocytes?

Please see R2-14 for further details about cross-species comparison of oligodendrocytes. We agree that the claim is oversimplified and have revised it to (lines 249–262):

"Also, we observed consistently larger differences between marmoset WM-OPC (OPC3) and OPC from all other species analyzed. We quantified this observation by comparing the fold-change of similarity between OPC subclusters, measured as the ratio of r^2 values across clusters (Fig. 4c)."

R2-14. Line 265-272 "We, therefore, pursued a 3D UMAP analysis of oligodendrocytes, finding a spiral pattern that was also observed upon reanalysis of previously reported human (Jäkel et al., 2019) (Fig. S27) and mouse (Zeisel et al., 2018) (Fig. S28) oligodendrocyte transcriptomes. This spiral pattern was also captured at the level of differentially expressed genes (DEG) across oligodendrocyte subclusters. Most oligodendrocyte DEG (*XYLT1*, *TNS1*, *TNS3*, *MAN1C1*, *BTBD16*, *CCP110*, *CSF1*, *DOCK5*, *PAM*, *MUSK*, *GPM6A*, *DPP10*) were aligned across species and were therefore used to label the gross developmental trajectory of oligodendrocytes (Fig. 4B, Fig. S27–29)."

Can the author provide evidence that the appearance and shape of a given cluster generated through any of the dimension reduction methodologies, UMAP in particular here, can bear any biological significance?

We have added a citation (McInnes et al., 2020) in the revised manuscript, entitled, "UMAP: Uniform Manifold Approximation and Projection for Dimension Reduction." We did not mean to assign any special significance to a spiral shape, but rather to discuss the implications of proximity of cells within a UMAP scatter plot and how in some instances adding a 3rd dimension can inform that discussion.

Along these lines, please see lines 261–266 in the revision:

"Different from other cell classes, marmoset oligodendrocytes were arranged into a continuous path in 2D dimension-reduced space (Fig. 5a), in which nuclei with similar transcriptomes are arranged as neighbors⁴³. We found this to be similar to the patterns in human^{44,45} and mouse^{36,38}. In the following sections, we

describe how this trajectory cannot be parsimoniously explained by a unidirectional path in oligodendrocyte lineage development...”

R2-15. Can the authors provide some perspective into why 5.5 yo (ostensibly middle-aged) monkeys would have a substantial mixed population of oligodendrocytes at varying stages of maturation resident in their CNS? Please clarify if this state extends to all oligos or is restricted to a particular region (e.g. neocortex/thalamus etc.).

Nuclei from all 19 tissue types were included in this part of the analysis, which includes caudate, thalamus, LGN, hippocampus, midbrain, pons, cerebellum, and spinal cord. We found relatively few regional differences in oligodendrocytes compared to other glial cell types, and, indeed, all oligodendrocyte subclusters were present across these tissue types. We revised our interpretation regarding the observation as follows (Lines 264–301):

“In the following sections, we describe how this trajectory cannot be parsimoniously explained by a unidirectional path in oligodendrocyte lineage development.

Based on mouse studies, differentiation-committed oligodendrocyte precursors are *Pdgfra*⁻/*Tns3*⁺³⁶, and the expression of *Enpp6* is a marker of newly forming oligodendrocytes^{34,46}. Therefore, we denoted as OLI1 the subcluster that is *PDGFRA*⁻/*TNS3*⁺/*ENPP6*^{high} and named the other OLI clusters (OLI2–6) consecutively (Supplementary Fig. 25e). Instead of clear GM-WM segregation, we found proportional differences along the intermingled OLI subtypes across brain regions. OLI1 was lowest in “GM” (median abundance ~0.5%), compared to ~10% relative abundance in “WM” and “other” (Supplementary Fig. 25c).

Based on this expression pattern, one might surmise that OLI1 are the youngest, and OLI6 the oldest, oligodendrocytes; however, we found them to be close in the space of the 2D UMAP projection, which could indicate transcriptomic similarity; alternatively, a 2D projection may not be sufficient to capture important aspects of the data. Therefore, we pursued a 3D UMAP analysis of oligodendrocytes, finding that the two ends are separated along a spiral pattern that was also observed upon reanalysis of previously reported human^{44,45} (Supplementary Fig. 28–29) and mouse³⁸ (Supplementary Fig. 30) oligodendrocyte transcriptomes. This spiral pattern was also captured at the level of differentially expressed genes across oligodendrocyte subclusters, most of which (*XYLT1*, *TNS1*, *TNS3*, *MAN1C1*, *BTBD16*, *CCP110*, *CSF1*, *DOCK5*, *PAM*, *MUSK*, *GPM6A*, *DPP10*) were aligned across species to label the overall developmental trajectory (Fig. 5b, Supplementary Fig. 28–31).

Despite some discrepancy in the assignment of subcluster identity among datasets, 5 major stages of oligodendrocyte lineage cells are widely accepted in the field to annotate the trajectory: OPC, differentiation-committed oligodendrocyte precursors (COP), newly formed oligodendrocytes (NFOL), myelin-forming oligodendrocytes (MFOL), and mature oligodendrocytes (MOL). Interestingly, however, *ENPP6*^{high} oligodendrocytes are located at both ends of the trajectory in mouse datasets (Supplementary Fig. 3031), but only at one end of the trajectory in marmoset and human. This observation raises the question of whether the second *ENPP6*^{high} population annotated is selectively lost in primates, or, alternatively, labeling the marmoset *ENPP6*^{high} population as “youngest” is not valid.”

In short, we clarified that the use of *ENPP6*^{high} labeling to inform the maturation stages of oligodendrocytes in marmoset based on mouse studies might not be accurate. Instead, we think the graded transcriptome pattern might be related to dynamic myelin modification throughout the lifespan.

R2-16. Line 276-277. "MUSK expression in oligodendrocytes may be unique to primate, as it can also be detected in human oligodendrocytes (Fig. S27) but not mouse."

This assertion is tenuous at best. The expression is barely visible in the human feature plot (Fig.S27C) and has not been quantitatively assessed, and the expression threshold to indicate expression is unclear.

We agree that the evidence for this claim was tenuous in the original submission. We have now added another human dataset (Absinta et al., 2021) to show MUSK expression in human oligodendrocytes (Supplementary Fig. 29). We have further performed in situ hybridization for tissue-level validation. We have revised the text as follows (lines 309–314):

"Moreover, it seems likely that there is species disparity in the expression of MUSK in oligodendrocytes, for it was detected in both marmoset and human but not in any mouse datasets. We validated that *MUSK* was indeed expressed by oligodendrocytes by fluorescent in situ hybridization: *MUSK*⁺/*OLIG2*⁺ double-labeled cells were found in both GM and WM of marmoset brain (Fig. 5c). Whether MUSK expression is unique to primates or animals in specific phylogenetic branches requires further investigation. "

R2-17. Line 317-319 "we found that molecular distances from OLI4 to OLI5 and from OLI4 to OLI6 were similar, indicating that OLI5 and OLI6 might develop in parallel rather than dependently (Fig. 4C–D, top)."

This result is fascinating. Do the authors have evidence of a pseudotime trajectory analysis that could indicate a bifurcation that might provide additional validation of this?

Monocle3 (v0.2.0) was used to construct nuclei trajectories based on transcriptomic distance (Cao et al., 2019). This analysis calculates the transcriptomic distance between nuclei and finds the shortest path from the first nucleus to the end nucleus. If two possible directions have a similar score, a bifurcation will be placed on the connected spine. This is essentially a pseudotime analysis.

R2-18. Line 328-330 "That ETV5 expression peaks in OLI3 (Step 60–80) suggests that OLI3 might be a population that is poised to further differentiate upon appropriate signaling (Fig. 4D)."

Is there prior literary evidence that can support this assertion as it appears pretty speculative? Assuming the pseudotime conclusion is correct and that the OLI subclusters are organized along with maturation states, could ETV5 in the middle of the continuum merely indicate suppression of differentiation from immature states towards functional maturation and functional specialization?

Correct, that is exactly what we thought. We have clarified our thinking as follows (lines 326–334):

"Of all transcription factors examined, only *ELF2* and *ETV5* peaked in the middle stages of oligodendrocytes (OLI2 and OLI3, respectively), whereas the other transcription factors were clustered either at the "early" (OLI1) or "late" (OLI4–6) stages. A positive correlation between *ELF2* and myelin was supported in a human snRNA-seq study, in which *ELF2* was high in control WM, normal appearing WM, and remyelinated multiple sclerosis lesions but lower in WM lesions (active, chronic active, and chronic inactive)⁴⁴. On the other hand, *Etv5* can act as a suppressor of oligodendrocyte differentiation, such that enforced expression of *Etv5* in rat OPC decreased the production of MBP⁺ oligodendrocytes⁵⁰."

R2-19. Line 332-335 "We observed shared transcriptomic features and

intermingled distribution of nuclei from astrocyte (AST) and vascular (VAS) classes in Level 1 analysis, so we pooled these two classes for the second round of quality control to facilitate artifact imputation.”

The rationale for the pooling of vastly different cell types for the analysis was not clear to me. Can the Authors explain the scientific rationale for this? For this analysis, it does not seem appropriate to pool astrocytes, one of the fundamental neural cell types, with non-neural cell types. Further, classifying astrocytes as ‘cells at the barriers of the CNS’ (Fig. S30) is highly reductive and not representative of the various (and more significant) homeostatic functions performed by astrocytes in the parenchyma. I acknowledge that astrocytes interact with, as well as regulate and maintain, the neurovasculature and the components of the BBB. However, interaction with the BBB and being a constituent of the BBB is a significant distinction. There is evidence that the BBB is formed before astrocytic colonization of the brain (DOI: 10.1038/nature09513), indicating the astrocytes are participants in its regulation but not a biophysical constituent of the BBB. Please clarify/justify and provide evidence that astrocytes can be classified in this way.

Our decision to pool AST and VAS to perform Level 2 quality control prior to splitting was technical in nature, based on observation of the cluster features of our dataset and not based on biology. Specifically, we noticed that some AST or VAS nuclei were intermingled in the UMAP plots.

We revised the manuscript as follows (lines 336–355):

“Cell types at the barriers of the CNS

In Level 1 analysis, we observed an intermingled distribution of nuclei with shared transcriptomic features from the astrocyte (AST) and vascular (VAS) classes. Therefore, we pooled these two classes for the second round of quality control, which facilitated artifact imputation before further cell class splitting. A total of 74,204 nuclei comprising astrocytes and primary cell types (endothelial cells, meningeal cells, and ependymal cells) present at the CNS barriers (blood–brain, blood–CSF, and brain–CSF) remained after quality control (Supplementary Fig. 32). As the neurovascular unit is mostly established prenatally⁵¹, we referred to a currently available ISH atlas of P0 marmoset²⁰ to confirm the localization of markers expressed by these cell types. We matched the gross histological morphology of P0 brain to the adult marmoset MRI atlas^{52,53}.

A total of 13,057 nuclei associated with CNS barriers comprised 11 VAS subclusters (Supplementary Fig. 34–36). Pericytes (Pericyte1–2), vascular endothelial cells (VE1–3), and vascular smooth muscle cells (VSMC) agreed with a human vascular atlas⁵⁴ and were broadly consistent across brain regions (Supplementary Fig. 34e). A relatively higher percentage of ependymal cells, which form a permissive interface between CSF and brain along the ventricular lining, was identified, as expected, in tissue samples that line ventricles (tWM, pCC, Cd, and cSC). The distribution of vascular and leptomeningeal cells (VLMC1–4, brain fibroblast-like cells) was variable and most highly detected in the hindbrain (pons and cerebellum; Supplementary Fig. 34b).”

R2-20. Fig. S31. Please provide evidence and citations of the various categories/levels of permeability of the CNS. More importantly, the authors lumped astrocytes in as ‘cells at the barriers of the CNS’ (currently without sufficient evidence and rationale); why then are astrocytes not included in this classification. By omission, this indicates that astrocytes are not, in fact, typically considered ‘cells at the barriers of the CNS’ and is, therefore, a significant flaw in the study design. Please clarify/justify and provide evidence that astrocytes can be classified in this way.

We thank the reviewer for reminder, and we now added an appropriate citation

in the caption of Supplementary Fig. 33. Also, please see our detailed response to R2-19.

R2-21. Fig. S31. ALDH1L1, SLC1A2, AQP4, GFAP are ASTROCYTE specific/rich markers. This figure is inappropriate and is highly misleading as these markers are not typically considered "genes enriched in endothelia, meningeal and ependymal cells". Additionally, it is inappropriate and misleading to compare ISH data at P0 to the adult snRNAseq dataset as it does not consider age-dependent changes in gene expression and is not directly representative of the dataset.

We apologize for the confusion and have added additional annotation in the figure to make clear the difference.

With respect to our use of the P0 dataset, please see lines 343–346: "As the neurovascular unit is mostly established prenatally⁵¹, we referred to a currently available ISH atlas of P0 marmoset²⁰ to confirm the localization of markers expressed by these cell types. We matched the gross histological morphology of P0 brain to the adult marmoset MRI atlas^{52,53}"

R2-22. Fig. S30-36. What is the point of grouping astrocytes with 'cells at the barriers of the CNS' (Fig. S30) when there was no subsequent analysis to analyze or dissect their relationship with each other? The analysis split them up and analyzed them individually in Fig. S32-34 (Astrocyte) and 35-36 (VAS).

Please see our response to R2-19.

R2-23. Fig. S32. Again, with all this information available regarding cortical and subcortical regions, why was these data not explored in more depth to look at the regional diversity in astrocyte identity and function beyond markers? As I understand it, The human astrocyte comprises multiple morphologies that may be identified based on gene expression identity. Why was no effort placed in aligning these marmoset data to that of humans, similar to that performed in (<https://doi.org/10.1101/2020.03.31.016972>)?

Our major focus in this paper is to understand glial diversity in the WM, a significantly understudied brain region that is highly relevant to many diseases. To put our findings in WM into a relevant context, we also included many cortical and other brain regions of adult marmoset. We have now included morphology analysis in the revised manuscript (lines 371–379):

"Moreover, GFAP⁺ astrocytes greatly varied in density, size, and shape across the brain (Fig. 6b–d). This agrees with what has been described in the human brain⁵⁶, specifically that protoplasmic astrocytes are primarily found in cortex, whereas WM astrocytes are fibrous in morphology (Fig. 6b). The number and dimension of GFAP⁺ cells are diverse across cortical layers, tissue type, and even WM areas. These results lead to a prediction that the brain's astroglial response to stimuli may be heterogeneous even across WM areas (Fig. 6b–d)."

R2-24. Line 354-355 "This result leads to the prediction that the brain's response to injury may not be uniform across WM areas."

This seems highly speculative and is not based on sufficient empirical or literary evidence. How did the authors come to this conclusion? Is it reasonable to suggest this merely based on differences in the expression of GFAP/AQP4? Can the authors provide evidence to support this?

We did not mean to suggest this idea as a conclusion, rather a hypothesis that arises out of the analysis of our resource dataset and remains to be tested. We have revised the manuscript as follows:

Lines 40–44:

“The effect of the WM environment on priming glia to be more advanced in their response to pathological challenges has been observed for astrocytes^{2,3}. For example, compared to astrocytes in GM, astrocytes in WM have a higher capacity for glutamate clearance to handle excitotoxic insults and disproportionately higher senescence-induced expression of GFAP (a reactive gliosis indicator)⁴.”

Lines 371–379:

“In “WM” samples, different profiles of astrocyte subtypes were observed; for example, AST4 and AST5 were enriched in the pCC and OpT but not other WM areas, similar to what was found in Thal, LGN, MB, Pons, and cSC (Fig. 6a). Moreover, GFAP⁺ astrocytes greatly varied in density, size, and shape across the brain (Fig. 6b–d). This agrees with what has been described in the human brain⁵⁶, specifically that protoplasmic astrocytes are primarily found in cortex, whereas WM astrocytes are fibrous in morphology (Fig. 6b). The number and dimension of GFAP⁺ cells are diverse across cortical layers, tissue type, and even WM areas. These results lead to a prediction that the brain’s astroglial response to stimuli may be heterogeneous even across WM areas (Fig. 6b–d).”

R2-25. Fig. S32F. Please clarify why the rationale behind grouping astrocyte clusters using the telencephalon/non-telencephalon designations. I am not convinced that neocortical entorhinal and thalamic structures should be grouped considering the clustering differences demonstrated in Fig. S32A. Additionally, rodent studies have ascribed vast region (<https://doi.org/10.1038/s41467-019-14198-8>, <https://doi.org/10.3389/fnins.2020.00061>) and laminar (<https://doi.org/10.1038/s41593-020-0602-1>) specific gene expression identity to astrocytes. In this context, what was the rationale for grouping astrocytes in this manner?

When we annotated the nuclei based on the tissue type, we observed a pattern that can be explained by this classification, similar to what was described in a mouse brain atlas. We should clarify that the bioinformatic parameters used to define subcluster identities by their transcriptomes are orthogonal to other classifications (e.g., detailed location within the brain) in the search for biological association. This is similar to the way in which we denoted MIC1 as GM-microglia because it is a subtype enriched in cortical regions. Here again, our annotation and grouping are meant to contextualize the results and facilitate comparison.

We clarify our thinking as follows (lines 380–387):

“Similar to what has been described in a mouse brain cell atlas³⁸, we found that grouping tissue by developmental category together with WM-GM disparity most effectively describes astrocyte segregation (Fig 6e, Supplementary Fig. 37a). This observation led us to investigate further the effect of local neural tube patterning signals in defining astrocyte subclusters and whether these signals also affect other cell types in the same region. Therefore, we examined the expression of patterning genes along the anterior-posterior axis across cell classes and tissue types (Fig. 6e). We found that the expression of patterning genes across brain regions was grossly preserved across cell types, with some interesting exceptions.”

R2-26. Line 385-400. Can the authors clarify the rationale behind the use of this subset of patterning genes in Fig. 5C?

These are the patterning genes that we detected in the marmoset dataset. Additional hindbrain patterning genes were available (e.g., HOX genes), but they were consistent with the ones we report in their expression pattern, so we only included a few for simplicity.

R2-27. Line 391-393. Is there evidence of these patterning genes in the caudate and HC that could support the authors' statement that they have "...mostly lost the expression of forebrain patterning genes."?

This is a description of our dataset, and we have added the figure citation (Fig. 6e) in the revised manuscript.

Lines 387-391:

"In telencephalon, all cell classes in cortical GM expressed high levels of patterning genes that were most prominently detected in the forebrain (*FEZF2*, *EMX1*, *FOXP1*, *DLX1*, *SHH*, *DLX5*). Caudate (enriched with *SIX3*) and hippocampus, though belonging to telencephalon gray (Supplementary Fig. 37a), have mostly lost the expression of forebrain patterning genes (Fig. 6e)."

R2-28. Line 398-399. "forebrain, midbrain, and hindbrain specification was preserved more prominently in AST than any other cell classes."

I would argue that the neuronal subclass has relative patterning gene expression as astrocytes. Can the authors provide clarification or a more explicit graphical representation of the data that supports this statement?

We note the overstatement and have revised the text as follows (lines 398-401):

"This forebrain, midbrain, and hindbrain specification was preserved prominently in astrocytes, and indeed determined their identity as distinct AST subclusters (Fig. 6e). This suggests that heterogeneity in developmental origin might play a role in subtype specialization in addition to GM-WM disparity in diversifying astrocytes."

R2-29. Fig. S34 - S36 is not cited in the main text. What is the relevance of these data?

We thank reviewer for pointing out and have updated the citation in the revised manuscript.

R2-30. Line 404-406 "To explore this possibility, we extracted and compared differentially expressed TF between analogous GM- and WM-specific glia pairs (*MIC1/MIC3*, *OPC1/OPC3*, *AST1/AST3*)."

Can the authors clarify if this comparison was performed using only neocortical glia within the listed clusters or total glia within the clusters, comprising subcortical, entorhinal and brainstem glia?

Can the authors clarify the scientific rationale for focusing on the shared TF between 3 different glial types with different origins, functions, and identities? Additionally, in this context, why were Oligodendrocytes separate from this comparison? The authors need to acknowledge that Glia is not a homogenous population that can/should necessarily be subjected to comprehensive wholesale analysis. As demonstrated in the authors' dataset. Glia is diverse and heterogeneous, and reducing all of them to this level, considering their vastly different gene expression profile, niche, origin, and function, would require much more scientific justification and validation than just because they are 'glia'.

Nuclei from all 19 tissue types were included in this part of the analysis, which includes caudate, thalamus, LGN, hippocampus, midbrain, pons, cerebellum, and spinal cord. See also our response to R2-38, about why oligodendrocytes are not included in this analysis. We acknowledge the glia are far from a homogeneous group of cells. We decided to perform a cross-glia analysis not because these cell types are glia per se, but rather based on observation of our dataset (enrichment of specific clusters in "WM" vs "GM" samples). Indeed, the exclusion of oligodendrocytes from this specific analysis is consistent with this rationale, as

no oligodendrocyte subclusters can be reliably assigned as “WM-oligodendrocytes” vs “GM-oligodendrocytes”.

Lines 38–50:

“Differences between gray and white matter start with the obvious differences in density of neurons and oligodendrocytes, but the extent of structural and functional heterogeneity of other resident cells remains unclear. The effect of the WM environment on priming glia to be more advanced in their response to pathological challenges has been observed for astrocytes^{2,3}. For example, compared to astrocytes in GM, astrocytes in WM have a higher capacity for glutamate clearance to handle excitotoxic insults and disproportionately higher senescence-induced expression of GFAP (a reactive gliosis indicator)⁴. Similarly, more microglia are found in WM than in GM of normal human brain⁵, and microglia in WM are primed to be more active and respond to injury faster than their GM counterparts^{3,6,7}. Moreover, it has been shown that the timing and efficiency of remyelination mediated by oligodendrocyte progenitor cell (OPC) differentiation varies significantly between GM and WM⁸. Thus, we wondered if location-specific regulatory programs influence resident cells broadly, and if these microenvironmental cues lead to transcriptomic segregation that further defines cell identities.”

Line 150–171 for GM-, WM-microglia.

Line 202–215 for GM-, WM-OPC.

Line 358–370 for GM-, WM-astrocytes.

Lines 404–417:

“The presence of gray-white matter segregation within some glial cell types, together with the observation of transcriptional similarity across glia within tissue types, led us to hypothesize that there might be regulatory programs that are shared by resident cells within the same microenvironment to execute intercellular functions properly. We reasoned that the similarity of enriched gene modules among resident glia might be due to the activation of common transcription factors. To explore this possibility, we extracted and compared differentially expressed transcription factors between matching GM-WM glia pairs (MIC1/MIC3, OPC1/OPC3, AST1/AST3). We found greater overlap in differentially enriched transcription factors in GM-glia (15 overlapping transcription factors) than WM-glia (3 overlapping transcription factors). Interestingly, 6 transcription factors were shared across GM-glia, whereas no transcription factors were shared across WM-glia. GM-glia transcription factors (*EGR1*, *HLF*, *PEG3*, *MYT1L*, *HIVEP2*, *BHLHE40*) are known to restrict RNA biosynthesis, consistent with the observation that GM-glia are low in RNA features compared to their WM counterparts (Fig. 7a, Supplementary Fig. 40a).”

R2-31. Line 410-411 “consistent with the observation that GM glia are low in RNA complexity compared to their WM counterparts (Fig. 6A, Fig. S37A).”
What is the authors’ definition of “RNA complexity”. I ask because the only quantitative measure of this presented in the manuscript is the number of genes expressed. Is this a sufficient measure of “RNA complexity”, or is it simply a matter of fewer genes expressed. How does this correlate to the ‘similar functional modules’ found in white matter MIC and OPC/OLIGO regarding a more ‘active’ state primed for migration?

We acknowledge the deficiency of our usage of the term and have revised the text as follows (lines 413–417):

“Interestingly, 6 transcription factors were shared across GM-glia, whereas no transcription factors were shared across WM-glia. GM-glia transcription factors (*EGR1*, *HLF*, *PEG3*, *MYT1L*, *HIVEP2*, *BHLHE40*) are known to restrict RNA biosynthesis, consistent with the observation that GM-glia are low in RNA

features compared to their WM counterparts (Fig. 7a, Supplementary Fig. 40a)."

R2-32. Fig. 6B I am struggling to find the relevance in GO/ gene module analysis between GM glia. In particular, what is the relevance and significance of making this comparison in a mixed population of glia, why is this a scientifically justifiable method of analysis, and the significance of this finding? E.g. Why do the "regulatory programs in GM microglia showed the highest similarity with those in GM-OPC"? What are the significance and the justification of the mixed-cell type comparison, etc.? This section started without adequate scientific justification and ended abruptly with no conclusion or statement regarding the significance of the findings. Perhaps the authors could expand this section to clarify the analysis and the findings to make it more easily understandable. Please address.

Please see our response to R2-30

R2-33. Line 416-442 This is an exciting section of results, and the authors have done an excellent job in categorizing and illustrating the potential contribution of cell types to various neurological disorders. This data section stood out as an exciting multicellular interrogation of probabilistic contribution to neurological disorders and neurodegenerative diseases. It would be exciting to see this section expanded with more in-depth analysis and in vivo validation of these findings. Does the number of genes expressed in a particular cell type cluster influence/affect the EWCE analysis? E.g. in Fig. 6, the authors showed that MIC3 exhibited a higher number of genes detected than MIC1. Can this increase in genes expressed be correlated to increased contribution by this cluster to neurological disorders?

We thank reviewer for their interest in this analysis and for raising the question. We certainly agree that in vivo validation of the predictions is important but believe it is beyond the scope of our paper, which is intended as a resource for further study. The EWCE method applied here compares the averaged expression of a disease-relevant gene list to expression profile of 100,000 random samplings from the background of the same cell. Therefore, our result should be robust to different gene numbers.

R2-34. This result section is titled "GM and WM glia differentially contribute to neurological disorders". However, only 1 example of GM vs. WM differences were reported (microglia). How can the authors justify using such a broad term in this context when the example provided was only limited to microglia? The authors have so much information at their disposal yet was not prepared to adequately synthesize these data into a more coherent and conclusive demonstration of the point they were trying to make. Please expand.

We appreciate the opportunity to expand on this in revised manuscript (lines 470-481):

"Consistent with the microenvironment specialization of glia reported here, we found examples in which genes associated with organic mental disorder were differentially expressed in GM-microglia (MIC1) and GM-astrocytes (AST1) but not in their WM counterparts. Genes associated with CNS tumor were enriched in WM-microglia (MIC3) and WM-OPC (OPC3), but surprisingly not in astrocytes. By contrast, all microglia subtypes (MIC1-MIC3), but not other cell types, appear to contribute to multiple sclerosis pathogenesis (Fig. 9b), consistent with results from genome-wide association studies⁶³. Interestingly, genes unique to CNS tumor and encephalitis (List.21) are differentially enriched MIC2, a less dominant GM-microglia that is present in various proportions in microglia sampled from "WM" (Fig. 3c, Supplementary Fig.14a). Together, these results support our

contention that there is transcriptome diversity among GM- and WM-glia, and that these variations are significant enough for specific subtypes to be predicted to contribute differentially to various neurological disorders.”

R2-35. Grouping the charts in Fig.6C and Fig.S38 along GM/ WM lines would significantly assist with data interpretation for the reader. At this stage, a reader has to expend a significant amount of effort to identify the origin of any particular cell cluster.

Can the authors clarify if the cell clusters contained neocortex only GM and WM in results or the legend, or is a mixed population of neocortical and subcortical cells grouped as GM and WM? This is significant because the classifications of neurological disorders range from pathologies that are generally localized to the neocortex/ thalamus/ brainstem etc. Please clarify and justify if a mixed population is used.

We thank the reviewer for the suggestion, and we now group the results on standard deviation and fold-change together into one panel for each disorder. We identified 87 subclusters across brain regions (all 19 tissue types are included), and each tissue has a different proportion of the 87 subclusters. For example, 7 of the 87 subclusters (NEU23–29) were detected only in “GM,” and 6 of the 87 subclusters (OLI1–6) are (not surprisingly) enriched in “WM.” All 87 subclusters are included in the analysis of Fig. 9 and Supplementary Fig. 43 (old Fig 6C and Fig. S38).

R2-36. Fig. 6C. Can the authors comment on the heavy association of microglia but not oligodendrocytes to MS? Was this expected based on previous literature? What is the functional/pathological significance of this, and can the authors cross-reference this finding to human MS datasets?

Yes, it was expected based on previous literature (genome-wide association studies) (International Multiple Sclerosis Genetic Consortium, 2019: “Multiple sclerosis genomic map implicates peripheral immune cells and microglia in susceptibility”).

R2-37. Line 422 (Fig. S37B) cited should be (Fig. S37C)

We have corrected this in the revised manuscript.

R2-38. Line 443 – again, why were OLIGOS not added to this analysis? Please clarify the origin of cerebral white matter. Cortical only or includes subcortical white matter? If it includes subcortical white matter, what is the scientific justification? Do the authors have evidence that cortical and subcortical white matter cells are homogenous and can be grouped in this manner?

We apologize for the confusion. Indeed, nuclei from all 19 tissue types are included in the OLI analysis. All OLI subclusters are detected in every tissue type, including caudate, thalamus, LGN, hippocampus, midbrain, pons, cerebellum, and spinal cord. As described more explicitly in the revised OLI analysis section (lines 261–273):

“Different from other cell classes, marmoset oligodendrocytes were arranged into a continuous path in 2D dimension-reduced space (Fig. 5a), in which nuclei with similar transcriptomes are arranged as neighbors⁴³. We found this to be similar to the patterns in human^{44,45} and mouse^{36,38}. In the following sections, we describe how this trajectory cannot be parsimoniously explained by a unidirectional path in oligodendrocyte lineage development.

Based on mouse studies, differentiation-committed oligodendrocyte precursors are *Pdgfra*⁺/*Tns3*⁺³⁶, and the expression of *Enpp6* is a marker of newly forming

oligodendrocytes^{34,46}. Therefore, we denoted as OLI1 the subcluster that is *PDGFRA*⁺/*TNS3*⁺/*ENPP6*^{high} and named the other OLI clusters (OLI2–6) consecutively (Supplementary Fig. 25e). Instead of clear GM-WM segregation, we found proportional differences along the intermingled OLI subtypes across brain regions. OLI1 was lowest in “GM” (median abundance ~0.5%), compared to ~10% relative abundance in “WM” and “other” (Supplementary Fig. 25c).”

Given the gradation in OLI transcriptomes, and the lack of clear GM-WM segregation across subtypes, we do not have enough support to assign specific subclusters as WM- or GM-oligodendrocytes. However, in the section on cellular contribution of neurological disorders, all OLI subtypes (indeed, all 87 subclusters) defined across all tissue types are included in the analysis.

R2-39. Fig.6D, Fig. S39. Can the authors explain why no OPC-OPC or AST-AST crosstalk was reported in the GM?

Fig. S39, Fig. S40. I do not understand the distinction between these 2 LT pairs analyses. Can the authors expand and elaborate to clarify?

In Fig. 8c (old Fig. 6D) and Supplementary Fig. 41 (old Fig. S39), the pie charts describe only the interactions that are unique to each environment. Shared intercellular interactions are summarized in Supplementary Fig. 41 (old Fig. S40), in which OPC-OPC and AST-AST interactions are indeed present. We apologize for any confusion.

R2-40. The result section ends abruptly with no conclusion or perspective about the authors’ dataset.

We revised the manuscript accordingly, adding a conclusion and perspective on the dataset in the new discussion (lines 483–491):

“We have provided a resource and initial analysis for each major cell class across 19 CNS tissue types. We observed the greatest GM-WM spatial segregation in subclusters of microglia, OPC, and astrocytes. GM-glia are generally naïve, protoplasmic, and enriched in GO terms related to neuronal functioning, whereas WM-glia are more active, fibrous, and enriched in GO terms related to morphogenesis and signaling dynamics. We accumulated some evidence that WM-glia have accrued additional features, are further advanced in the program of specialization, and are more interactive than their GM counterparts. This atlas therefore serves as a bridge between rodent and human data that may prove useful for the understanding of the cellular and molecular basis of human neurological disorders.”

R2-41. Line 550-551 “In summary, although environmental cues contribute to diversifying astrocytes, the heterogeneity in their developmental origin plays a larger role in subtype specialization.”

I am not convinced that the authors have performed sufficient or accurate AST cluster heterogeneity to make this statement. Please justify.

We acknowledge the statement was oversimplified and have now revised as follows (lines 398–401):

“This forebrain, midbrain, and hindbrain specification was preserved prominently in astrocytes, and indeed determined their identity as distinct AST subclusters (Fig. 6e). This suggests that heterogeneity in developmental origin might play a role in subtype specialization in addition to GM-WM disparity in diversifying astrocytes.”

R2-42. Line 557-558 “We examined the OPC expression of ion channel genes as a surrogate of electrophysiological function and examined the tissue origin of differentiating OPC (OPC5).”

How can the authors justifiably consider the expression of these ion channels a “surrogate of electrophysiological function”? The authors only observe and report the transcription of these ion channels without any form of evidence translation or in vivo/ ex vivo histochemical proof of the channels’ subcellular localization.

We acknowledge the limitation of this analysis; however, we believe it unlikely that OPC subclusters that possess different expression profiles of ion channels have the same electrophysiological properties. We have expanded on this idea in the following (lines 228–234):

“Electrophysiological properties of OPC vary between WM and GM and with age, and they correlate with differentiation potentiality^{32,33}. We examined the OPC expression of ion channel genes as a surrogate of electrophysiological function and examined the tissue origin of differentiating OPC (OPC5). We found different profiles of ion channels in GM-OPC and WM-OPC (Supplementary Fig. 21) but a similar abundance (<1.5%) in OPC5 across brain regions (Fig. 4a). How these observations translate to actual differences in stimulus responses in health and disease requires further study.”

R2-43. Line 563-570 Interestingly, the final discussion section suggests that this study aimed to develop and validate computational tools or develop a protocol pipeline to analyze heterogeneity. If this is the case, then it really should be made much clearer at the start of the main text because this final paragraph does confuse the context of this study. If this is true, why was there so little in the analysis pipeline included in the discussion? The probabilistic multicellular neurological disorders association data, which is a significant highlight of this paper for me, was only relegated to one line in the discussion. The discussion ultimately falls flat without highlighting the enormous potential of the work to drive additional avenues of research forward and its implications for interspecific comparison of CNS cellular heterogeneity.

We thank reviewer for the direction, and we have extended the section by including more discussion of the analysis pipeline and the predictions regarding neurological disorders.

Reviewer #3 (Remarks to the Author):

This interesting study by Lin et al. focuses on understanding how microenvironmental influences in different regions of the CNS impact developmental and functional heterogeneity of glia. The authors utilize the marmoset to investigate this question in white matter and gray matter regions of the marmoset brain. The marmoset CNS displays unique properties when compared to mouse, in particular the marmoset white matter (WM) is closer in size to higher primates and humans. Furthermore, the marmoset is an animal model that bridges mouse and higher primates not only genetically, but also immunologically and behaviorally. Therefore, results obtained in the marmoset will more closely resemble cellular and molecular heterogeneity found in human brain.

The authors used snRNAseq profiling to define cells (total of 500,000 cells analyzed) from 19 tissue types from the normal marmoset CNS, and spatially mapped 87 distinct subclusters onto a 3D MRI atlas. They also performed a broad range of molecular analyses - including cross-species comparisons, developmental and regulatory pathways, screening of cellular determinants of neurological disorders, and developed models of intercellular regional communication. The authors conclude that the most significant finding resulting from this analysis is the marked effect of gray matter (GM) vs. WM on a broad range of glial cell types, in particular microglia, astrocytes and OPCs - indicating not only persistent developmental influences, but also highlighting functional

heterogeneity in these cell types. Molecular complexity of glia is higher in WM than in GM, where bioinformatic analysis predicts more communication among adjacent cells.

Overall, this is a well performed study, which provides a wealth of data on glial cells of the WM – a brain region that has been significantly understudied in different animal models. Furthermore, this is a significant effort in defining important regional differences in these glial cell populations in WM vs. GM. The outcomes of this study will also provide a solid platform for analysis in animal models of injury and disease. However, the paper is really a molecular atlas that could be useful to investigators and many of the results are significantly overinterpreted, or explained without considering potential alternative interpretations.

We thank the reviewer for affirming the value of our work as a resource. Along these lines, and as suggested by the editor, we have revised our manuscript to emphasize this point, adding morphological validation of some of our findings.

Remarks to the authors:

R3-1. The paper is a great resource, and the experiments and analysis were well performed but conclusions based on molecular data are not validated by cell specific protein expression patterns that would directly support the conclusions.

We agree with the reviewer that tissue-level validation of the findings would strengthen our conclusions. As a first step in this direction, we have added particle/morphological analysis of microglia and compared their density and morphology in WM and GM (Fig. 3f–h). We have now specifically validated that anti-SLC15A1 selectively labels microglia in WM (Fig. 3d). We also confirmed the oligodendroglial expression of *MUSK* by FISH (Fig. 5c). In the revision, we add a survey of the morphology of astrocytes across cortical layers and multiple CNS regions (Fig. 6b–d).

R3-2. The paper highlights the interaction between neurons and glia. It would be interesting to examine the opposite communication as well. The authors assume that the OPCs are influenced differently in WM vs GM microenvironment, but they do not provide strong experimental evidence in favor of this conclusion.

We acknowledge the limitation of the analysis: the interpretation of intercellular interactions is only as good as the curated database on ligand-receptor-target relationships. Our atlas provides initial analysis and associational observations to help prioritize pathways of interests. Additional genetic tools are required to answer this question properly, which is well beyond the scope of the current work.

R3-3. The authors show that GM microglia are predicted to be more involved in modulating neuronal synaptic activity and the WM more primed to active migratory state. It would be interesting to see if this is true in specific regions (for example for in WM corpus calosum vs. internal capsule vs. subcortical WM; and for GM cortex vs. basal ganglia vs cerebellum).

We agree with the reviewer that this would be an interesting comparison; however, due to the limited space in the manuscript, we first highlighted shared features across coarse tissue types. Further analysis in combination with tissue-level validation will be necessary to assess region-specific phenotypes in each fine structure.

Lines 511–513:

"Finally, due to the limited space in the manuscript, we highlighted shared

features across coarse tissue types; further analysis in combination with tissue-level validation is necessary to assess region-specific phenotypes in each fine structure.”

R3-4. The authors show that MUSK is uniquely expressed in primates, but not in mouse brain oligodendrocytes. This is a very interesting finding particularly in homeostatic conditions of the mouse basal forebrain. Can this be validated at the protein level? Is MUSK differentially expressed in WM vs. GM, and is it developmentally regulated?

We agree with the reviewer that this is an interesting finding and would have liked to validate it in the protein level; however, due to the limited pool of antibodies that are available for marmoset, we turned to the FISH approach and confirmed a peri-nucleus labeling of *MUSK* in OLIG2⁺ cells. The MUSK⁺/OLIG2⁺ cells are detected in both GM and WM, which agrees with what we found in our snRNA-seq atlas.

R3-5. The authors show that the gene transcription in OL lies on a spiral trajectory, and modeled whether this can be affected by the environment. In the pseudotime analysis of the marmoset, why did the authors set the starting point for the analysis as the ENPP6^{high} ?

We thank reviewer for raising this question. In the revised manuscript, we have clarified our thought process on how we define subclusters based on ENPP6 expression and the limitation of this approach (lines 267–273). In short: “Based on mouse studies, differentiation-committed oligodendrocyte precursors are *Pdgfra*⁻/*Tns3*⁺³⁶, and the expression of *Enpp6* is a marker of newly forming oligodendrocytes^{34,46}. Therefore, we denoted as OLI1 the subcluster that is *PDGFRA*⁻/*TNS3*⁺/*ENPP6*^{high} and named the other OLI clusters (OLI2–6) consecutively (Supplementary Fig. 25e). Instead of clear GM-WM segregation, we found proportional differences along the intermingled OLI subtypes across brain regions. OLI1 was lowest in “GM” (median abundance ~0.5%), compared to ~10% relative abundance in “WM” and “other” (Supplementary Fig. 25c).”

For these reasons, we set OLI1 (ENPP6^{high}) as the starting point for the pseudotime analysis.

R3-6. The authors report that GM and WM glia differentially contribute to neurological disorders and tumors. However, the authors used IPA, which often overrepresents genes associated with solid tumors.

We acknowledge the limitation of the analysis; indeed, the interpretation is only as good as the annotation curated in the IPA database. We were unaware that annotation for solid tumors is overrepresented in that database. Therefore, we reached out to the QIAGEN Digital Insight Support to understand how the IPA database was built. We were led to this [article](https://qiagen.secure.force.com/KnowledgeBase/KnowledgeIPAPage?id=kA41i000000CjU2CAK)

(<https://qiagen.secure.force.com/KnowledgeBase/KnowledgeIPAPage?id=kA41i000000CjU2CAK>), explaining that genes related to many fundamental biological processes are often dysregulated in cancer, therefore cancerous terms appear more often than other diseases.

Our analysis aims to highlight that the variations of transcriptome diversity among GM- and WM-glia are significant enough for each subtype to be predicted to contribute differentially to various neurological disorders. The EWCE method applied here controlled for different numbers of genes in each list by sampling 100,000 times from a background of the same size. Therefore, different strengths of annotation among neurological disorders should not affect the

statistical power.

R3-7. The analysis of autism-related genes is superficial and biased. The authors are basing their analysis of genes associated with autism on a study by Polioudakis et al., 2019, in which human fetal brains were analyzed (midgestation) and not adult brains. Importantly, Polioudakis et al., referenced previous studies for gene markers of autistic disorders and schizophrenia (adolescent and adult brains) to create an atlas in fetuses. This is an important point that the authors should take into consideration when they analyze their data in adult marmoset brain. This may be resolved either by selecting a different (adult) database, or by applying this dataset to the developing marmoset brain.

We agree with the reviewer and acknowledge the limitation of the analysis. We compared our analysis to that of Polioudakis et al. 2019. That we reached similar conclusions demonstrates the agreement across different bioinformatic tools and serves as a positive control for our approach. However, it does not address the biases rooted in the particular curated list in interpreting disease susceptibility, as mentioned in R3-6. We agree with the reviewer that it is important to consider developmental stages to reach a clinically more relevant conclusion.

R3-8. The authors show that WM glia interact with other resident cells more than GM glia. This is an important finding by which GM glia is characterized as naive protoplasmic, and not very active compare to the WM glia. It would be very useful if the authors could provide some morphological and immunohistochemical assessment of the WM vs. GM.

We thank reviewer for the suggestion, and we have now explored the morphological variation of microglia (Fig. 3) and astrocytes (Fig. 4) in WM and GM through immunohistochemical assessment.

Lines 168–171:

“We found 2–3 times more IBA⁺ cells present in WM compared to GM, which agrees with the relative abundance of microglia profiled from “GM” and “WM” with snRNA-seq (Fig. 3c). Moreover, the shape of microglia in WM is more elongated, indicated by a larger value of reciprocal circularity, compared to GM (Fig. 3e–h).”

Lines 373–377:

“Moreover, GFAP⁺ astrocytes greatly varied in density, size, and shape across the brain (Fig. 6b–d). This agrees with what has been described in the human brain⁵⁶, specifically that protoplasmic astrocytes are primarily found in cortex, whereas WM astrocytes are fibrous in morphology (Fig. 6b). The number and dimension of GFAP⁺ cells are diverse across cortical layers, tissue type, and even WM areas.”

REVIEWER COMMENTS

Reviewer #1 (Remarks to the Author):

I acknowledge authors' attention to all the points raised after review. The authors satisfactorily answered all the comments. I believe the manuscript is better explained and with newly introduced plots/workflow and better elaborated pre-processing and other steps, it should be better received and understood by the readers now. In my opinion, this manuscript should be accepted for the publication.

Reviewer #2 (Remarks to the Author):

We have been through the revised version of the manuscript and are completely satisfied with the responses and the way they clarified data and recontextualised it, which was very much appreciated. The manuscript now reads better and will be clearer to a general neuroscience reader. There were a few minor figure citation issues that will obviously be picked up at the copy-editing stage

Reviewer #3 (Remarks to the Author):

The authors have extensively revised the manuscript. However, I believe that there are still significant issues in this study that have not been addressed. These were clearly outlined in my previous review and have been dismissed by the authors.

R3-2: This issue was not addressed, as it was considered "outside the scope"

I respectfully disagree with the authors. I previously asked whether the authors could provide some direct evidence that OPCs are influenced differently in WM vs GM microenvironment, but they still do not provide strong experimental evidence in favor of this conclusion.

R3-3: Issue not addressed because of limited space in manuscript

I believe this is not a good reason for not addressing this point and dismiss it, as a supplemental figure could be added. The authors show that GM microglia are predicted to be more involved in modulating neuronal synaptic activity and the WM more primed to active migratory state. It would be interesting to see if this is true in specific regions (for example for in WM corpus callosum vs. internal capsule vs. subcortical WM; and for GM cortex vs. basal ganglia vs cerebellum). Addressing this point is crucially important to support the paper's conclusions.

R3-4: Question of differential expression in GM vs. WM not addressed, nor is developmental regulation.

This is another basic and crucial issue about MUSK expression and function that was not addressed: The authors show that MUSK is uniquely expressed in primates, but not in mouse brain oligodendrocytes. This is a very interesting finding particularly in homeostatic conditions of the mouse basal forebrain. Can this be validated at the protein level? Is MUSK differentially expressed in WM vs. GM, and is it developmentally regulated?

R3-7: Issue acknowledged, but not really addressed

This is a very serious issue, as the analysis of autism-related genes provided in this paper is superficial and biased, and therefore incomplete and incorrect. I am repeating my points: The authors are basing their analysis of genes associated with autism on a study by Polioudakis et al., 2019, in which human fetal brains were analyzed (midgestation) and not adult brains. Importantly, Polioudakis et al., referenced previous studies for gene markers of autistic disorders and schizophrenia (adolescent and adult brains) to create an atlas in fetuses. This is an important point that the authors should take into consideration when they analyze their data in adult marmoset brain. This may be resolved either by selecting a different (adult) database, or by applying this dataset to the developing marmoset brain.

REVIEWER COMMENTS

Reviewer #1 (Remarks to the Author):

I acknowledge authors' attention to all the points raised after review. The authors satisfactorily answered all the comments. I believe the manuscript is better explained and with newly introduced plots/workflow and better elaborated pre-processing and other steps, it should be better received and understood by the readers now. In my opinion, this manuscript should be accepted for the publication.

We again thank the reviewer for the suggestions and agree that they helped us improve the paper.

Reviewer #2 (Remarks to the Author):

We have been through the revised version of the manuscript and are completely satisfied with the responses and the way they clarified data and recontextualized it, which was very much appreciated. The manuscript now reads better and will be clearer to a general neuroscience reader. There were a few minor figure citation issues that will obviously be picked up at the copy-editing stage

We thank the reviewer for the suggestions and for pointing out the citation issue; we have corrected the errors.

Reviewer #3 (Remarks to the Author):

The authors have extensively revised the manuscript. However, I believe that there are still significant issues in this study that have not been addressed. These were clearly outlined in my previous review and have been dismissed by the authors.

R3-2: This issue was not addressed, as it was considered "outside the scope"

I respectfully disagree with the authors. I previously asked whether the authors could provide some direct evidence that OPCs are influenced differently in WM vs GM microenvironment, but they still do not provide strong experimental evidence in favor of this conclusion.

We thank the reviewer for pointing out the ways in which our analysis is limited in this regard. Indeed, we did not prove that OPCs are influenced differentially in white and gray matter in marmoset, for example by transplanting the same clone of marmoset OPCs to WM or GM and testing if the transcriptomes become distinct over time. Such a study was in fact done in mice (Viganò et al. 2013), revealing regional differences in OPC maturation potential: *"Taken together, our results demonstrate that cells from the white matter differentiate efficiently into mature, myelinating oligodendrocytes in both more (white matter) and less (gray matter) supportive environments, whereas gray matter-derived cells do so less efficiently. Our data suggest that there are intrinsic differences between adult OPCs from the white and gray matter, which may be a result of the long residence of the cells in these environments. However, the limited differentiation of gray matter-derived cells could be overcome, to some extent, by transplantation into the supportive white matter environment."* We hope the reviewer and editor will agree that replicating this study in the nonhuman primate would have been a significant undertaking, likely meriting its own paper.

Our transcriptomic work does lead us to associational observations, which are very much in line with the views in the references we cited (Lentferink et al. 2018, Viganò et al. 2013). Specifically, WM-OPC appear transcriptionally more mature and form more ligand-receptor pairs with neighboring cells compared to GM-OPC. To acknowledge that we do not have direct experiment support, we have rearranged the section to report our findings from the pathway analysis and then presenting our hypothesis regarding environmental effects on OPC specialization:

Lines 230–234

"Taken together, these findings lead us to hypothesize that divergent CNS environments might influence the molecular profile of their resident cells in primates, and specifically that WM-OPC acquired additional features in response to their intercellular microenvironment. Testing this hypothesis and determining whether our observations translate to actual differences in stimulus responses in health and disease requires further experimental study."

We also added the following clarification to the Discussion:

Lines 515–518:

Instead, we performed associational analysis and first highlighted shared features across coarse tissue types. Further analysis in combination with direct experimental tissue-level validation is necessary to assess region-specific phenotypes in each fine structure.

R3-3: Issue not addressed because of limited space in manuscript

I believe this is not a good reason for not addressing this point and dismiss it, as a supplemental figure could be added. The authors show that GM microglia are predicted to be more involved in modulating neuronal synaptic activity and the WM more primed to active migratory state. It would be interesting to see if this is true in specific regions (for example for in WM corpus callosum vs. internal capsule vs. subcortical WM; and for GM cortex vs. basal ganglia vs cerebellum). Addressing this point is crucially important to support the paper's conclusions.

We apologize for not appreciating the importance of the reviewer's request in the previous revision. In the new version, we added a figure (SupFig14f; see also the dotplot below) plotting the expression level of GM-microglia (MIC1) enriched module (Knn.m3) and WM-microglia (MIC3) enriched module (PG.m4) across specific tissue areas within WM, GM, and other tissue types, as suggested. Within the WM tissue type, PG.m4 module is more enriched in MIC3 of fWM/tWM/pWM compared to corpus callosum (aCC/pCC) and subcortical WM (Opt). Across GM, basal ganglia, and cerebellum, Knn.m3 is more enriched in MIC1 of fCTX/tCTX/pCTX/oCTX/CgG/LGN compared to basal ganglia (Cd) and cerebellum (CE). These results further emphasize the presence of regional transcriptomic diversity.

SupFig14f caption:

f. Dot plot showing the averaged and scaled expression of PG.m4 (WM-microglia enriched) and Knn.m3 (GM-microglia enriched) modules across MIC subclusters in each tissue type. Compared to deep GM, the module involved in neuronal activity (Knn.m3) is highly enriched in MIC1 and MIC2 of cortical areas and LGN. In WM, the module involved in stimulus response (PG.m4) is mostly enriched in fWM, tWM, and pWM. Although there appears to be high expression of Knn.m3 in CgG for MIC3, this is due to low abundance of MIC3 (only 4 nuclei) in CgG. Taken together, these results further emphasize the presence of regional transcriptomic diversity.

We also adjusted the statement in the discussion to acknowledge the limitation of our study:

Lines 515–518:

Instead, we performed associational analysis and first highlighted shared features across coarse tissue types. Further analysis in combination with direct experimental tissue-level validation is necessary to assess region-specific phenotypes in each fine structure.

R3-4: Question of differential expression in GM vs. WM not addressed, nor is developmental regulation.

This is another basic and crucial issue about MUSK expression and function that was not addressed: The authors show that MUSK is uniquely expressed in primates, but not in mouse brain oligodendrocytes. This is a very interesting finding particularly in homeostatic conditions of the mouse basal forebrain. Can this be validated at the protein level? Is MUSK differentially expressed in WM vs. GM, and is it developmentally regulated?

We thank the reviewer for reiterating this point, and we agree with the reviewer that we would have liked to validate the finding at the protein level. Unfortunately, despite significant effort, we have not been able to detect MUSK protein in marmoset with IHC unambiguously; therefore, we turned to the FISH approach. Additionally, since we only have one adult time point in this study, we cannot make any definitive statements about whether MUSK is developmentally regulated. From the FISH results, we confirmed that MUSK⁺/OLIG2⁺ cells are detected in both GM and WM, which agrees with what we found in our snRNA-seq atlas (please see below microscope images, also new SupFig27d). By plotting the expression of MUSK genes across oligodendrocyte subclusters within each coarse tissue type (see below violin plot and SupFig27e), we found that the level of

MUSK is consistently enriched in OLI3–6 compared to OLI1–2 in both GM and WM. More MUSK+ nuclei were found in WM compared to GM in each subcluster; however, the overall expression level per MUSK+ nuclei in WM and GM was largely comparable (SupFig27e), and we indeed did not find a noticeable difference in MUSK level per individual OLIG2+ cell in WM compared to GM by FISH (SupFig27d). As further discussed in Lines 259–336, the expression of genes (including MUSK) in oligodendrocytes is often graded within a mixed population along the transcriptome trajectory, such that a clear-cut oligodendrocyte “subtype” (i.e., a distinct cluster) cannot be defined to perform the head-to-head comparison; therefore, we are limited and can only summarize the observations into trends in such dotplot. Hence, with the current experimental resolution, we cannot conclude quantitatively if a subpopulation of oligodendrocytes expresses MUSK differentially in WM versus GM.

SupFig27d–e caption:

d. The expression of *MUSK* is detected in OLIG2+ cells in the cortex and corpus callosum of adult marmoset brain by combined immunofluorescent staining and fluorescent in situ hybridization (Hybridization chain reaction v3.0).

e. Violin plot showing the expression of *MUSK* across OLI subclusters in GM and WM.

Lines 311–317:

“Although protein-level validation of *MUSK* expression in tissue was unsuccessful, we found that *MUSK* was indeed expressed by oligodendrocytes by fluorescent in situ hybridization: *MUSK*+/*OLIG2*+ double-labeled cells were found in both GM and WM of marmoset brain, and there was no noticeable difference in *MUSK* level per individual *OLIG2*+ in GM compared to WM (Fig. 5c, Supplementary Fig. 27d–e). Whether *MUSK* expression is unique to primates or animals in specific phylogenetic branches, and the extent to which it is developmentally regulated, require further investigation.”

R3-7: Issue acknowledged, but not really addressed

This is a very serious issue, as the analysis of autism-related genes provided in this paper is superficial and biased, and therefore incomplete and incorrect. I am repeating my points: The authors are basing their analysis

of genes associated with autism on a study by Polioudakis et al., 2019, in which human fetal brains were analyzed (midgestation) and not adult brains. Importantly, Polioudakis et al., referenced previous studies for gene markers of autistic disorders and schizophrenia (adolescent and adult brains) to create an atlas in fetuses. This is an important point that the authors should take into consideration when they analyze their data in adult marmoset brain. This may be resolved either by selecting a different (adult) database, or by applying this dataset to the developing marmoset brain.

We are sorry for any confusion or misunderstanding here. The gene list we used was not based on the findings of the fetal brain study (Polioudakis et al. 2019), nor on any expression trajectory. To clarify this point, in the revision, we have changed the citation (Line 457–458) to reference the SFARI database (syndromic and level 1,2 high confidence genes), which is a central resource for the field that curates evidence-based ASD risk genes from the literature, as well as Gordon et al. *Nature Neuroscience* 2021, where these lists were used as an example. We note incidentally that the Polioudakis et al. paper used the same lists for their analysis of fetal brains.

To acknowledge that we only performed an associational analysis, which was meant to highlight potential differential cellular contributions to a variety of neurological disorders when considering a pool of risk genes across marmoset brain cell types, we clarified the following:

Lines 461–464:

“Genes associated with autism spectrum disorder or intellectual disability were enriched in both excitatory and inhibitory neurons^{62,63}, and there was a remarkably similar profile for seizures and schizophrenia (Supplementary Fig. 43).”

REVIEWER COMMENTS

Reviewer #3 (Remarks to the Author):

I believe the authors have now addressed all our comments as sufficiently as possible. I am happy with the new data and the additional figures, and how the paper now reads.

REVIEWERS' COMMENTS

Reviewer #3 (Remarks to the Author):

I believe the authors have now addressed all our comments as sufficiently as possible. I am happy with the new data and the additional figures, and how the paper now reads.

We thank the reviewer for comments and suggestions.